# GATITOS:
# Using a New Multilingual Lexicon for Low-resource Machine Translation

**Alex Jones**[*, †]     **Isaac Caswell**[*]     **Ishank Saxena**     **Orhan Firat**

Google Research

## Abstract

Modern machine translation models and language models are able to translate without having been trained on parallel data, greatly expanding the set of languages that they can serve. However, these models still struggle in a variety of predictable ways, a problem that cannot be overcome without at least some trusted bilingual data. This work expands on a cheap and abundant resource to combat this problem: bilingual lexica (BILEXs). We test the efficacy of bilingual lexica in a real-world setup, on 200-language translation models trained on web-crawled text. We present several findings: (1) using lexical data augmentation, we demonstrate sizable performance gains for unsupervised translation; (2) we compare several families of data augmentation, demonstrating that they yield similar improvements, and can be combined for even greater improvements; (3) we demonstrate the importance of carefully curated lexica over larger, noisier ones, especially with larger models; and (4) we compare the efficacy of multilingual lexicon data versus human-translated parallel data. Based on results from (3), we develop and open-source GATITOS, a high-quality, curated dataset covering 170 mostly low-resource languages at the time of this submission, one of the first human-translated resources to support many of these languages[1].

## 1 Introduction

Neural machine translation (NMT) has emerged as the dominant way of training machine translation models (Bahdanau et al., 2015), where translation is modeled as a sequence-to-sequence task to be learned by neural networks (Sutskever et al., 2014). Massively multilingual machine translation (MMMT) refers to the concept of training a single machine translation model on many languages and language pairs using a shared set of parameters, and has also seen success in recent years (Firat et al., 2016; Wu et al., 2016; Johnson et al., 2017; Aharoni et al., 2019; Fan et al., 2022; NLLBTeam et al., 2022; Bapna et al., 2022; Siddhant et al., 2022). Training these models typically relies on large-scale parallel corpora mined from the web (Resnik and Smith, 2003; Uszkoreit et al., 2010; Esplà-Gomis, 2009; Bañón et al., 2020).

However, beyond the traditional technique of training NMT models with human-translated parallel texts, a number of other strategies have shown success recently, especially on lower-resource languages. One of these techniques is self-supervised training using monolingual corpora (Siddhant et al., 2020; Cheng et al., 2021). With this approach, NMT models are pretrained or jointly trained on a self-supervised task with monolingual data, such as the MASS (Song et al., 2019b) or BART (Lewis et al., 2020; Liu et al., 2020) tasks, as well as the usual neural machine translation task. This training regime can aid the model in performing zero-shot translation (Bapna et al., 2022; Siddhant et al., 2022), in cases where a language has monolingual data but no parallel data. Moreover, both the self-supervised task and the supervised MT task can be modeled as neural sequence-to-sequence (Seq2Seq) problems, meaning a single Seq2Seq model can be used for training on both tasks.

Other techniques that have proven useful for low-resource MT include back-translation (Sennrich et al., 2016; Caswell et al., 2019; Feldman and Coto-Solano, 2020) and the incorporation of language models into MT training (Gulcehre et al., 2017; Baziotis et al., 2020; Freitag et al., 2022b). There has also been extensive work on training completely unsupervised MT systems using monolingual corpora only (Artetxe et al., 2017, 2019). For example, Artetxe et al. (2017) uses a

---

[*]Equal contribution. Correspondence to icaswell@google.com, alexjones1925@gmail.com.

[†]Work done while interning on the Translate team at Google.

[1]https://github.com/google-research/url-nlp/tree/main/gatitos

combination of denoising autoencoding with pre-trained cross-lingual embeddings and on-the-fly back-translation to achieve reasonable MT performance with zero parallel data.

In our work, we supplement the approach that combines supervised and self-supervised training with multilingual lexica. The motivation for using this resource is as follows. Despite the successes of the approach combining supervised and self-supervised training, cross-lingual vocabulary alignment is still highly imperfect in these models, especially for low-resource and unsupervised languages (see Bapna et al. (2022) for examples of some common failure modes). That is, training on all languages using a shared set of parameters is insufficient to induce perfect cross-lingual vocabulary alignment. Of course, we are not the first to experiment with multilingual lexica to improve NMT performance, or multilingual NLP applications more generally; Section 2 gives more details.

Using the publicly available massively multilingual lexicon Panlex (Kamholz et al., 2014), we demonstrate that this added lexical data leads to small but significant gains over a baseline model on average, even for high-resource languages; and with smaller but carefully curated bilingual lexica, the gains are substantially larger. In both cases, the gains are most significant for unsupervised and low-resource languages. Our contributions are as follows:

1. We provide a thorough comparison of several lexicon-based data augmentation variants for MT, all of which are simple, generalizable, and easy to implement;
2. We test these approaches "in the wild", i.e. on in a highly multilingual, web-mined data regime such as production systems tend to use, with hundreds of languages and billions of monolingual and parallel sentences;
3. We explore the effects of lexical data quality *and* quantity;
4. We demonstrate the efficacy of bilingual lexicon-based approaches as models scale;
5. We open-source the high-quality multilingual GATITOS lexicon for low-resource languages.

The **tl;dr** of this paper is that bilingual lexica help low-resource and zero-shot NMT in almost all cases, and that most training-time augmentation methods have similar efficacy, and can be combined to be more effective. When scaling up to larger and more expressive models, these methods

retain their efficacy, but the quality of the translated bilingual lexica becomes more important than the sheer quantity of lexical data points used for data augmentation. For instance, small, high-quality lexica like GATITOS show about 5x larger CHRF improvement than larger, noisier lexica like Panlex.

Throughout this paper, experiments are done on only 24 GATITOS languages; based on the success of these, we expand the dataset to 170 languages, all low-resource, and open-source it.

## 2 Related Work

A number of works have looked at using multilingual lexicon data augmentation for NMT and other NLP tasks. The first class of augmentations that we experiment with is "codeswitching," where words in the source sentence are swapped out for their dictionary translations to create mixed-language sentences. This approach has been used for a range of multilingual NLP tasks, including MT (Reid and Artetxe, 2022; Yang et al., 2020; Liu et al., 2021; Lin et al., 2020, 2021; Pan et al., 2021; Yang et al., 2021; Kumar et al., 2022; Khatri et al., 2021; Kuwanto et al., 2021; Xia et al., 2019). Many of these, however, only look at codeswitching between the source and target languages, e.g. substituting source words with dictionary translations into the target language, or word-for-word BiLex translations of the target to make synthetic back-translated data (Nag et al., 2020). Qin et al. (2020) experiment with codeswitching on NLI, sentiment classification, document classification, dialogue state tracking, and spoken language understanding, Malon (2021) looks at codeswitching embeddings for language modeling, and Wang et al. (2022) experiment on NER, POS tagging, and dependency parsing. Another similar work is Chaudhary et al. (2020), in which the MLM task is modified such that instead of predicting masked source tokens in the *source* language, the authors provided language embeddings to cue the model to predict the masked tokens in a *different* language instead. Codeswitching augmentations go by a variety of different names, e.g. "dictionary denoising" (Reid and Artetxe, 2022), "Random Aligned Substitution" (Lin et al., 2020), or "code-mixing". In our paper, we will stick to the term "codeswitching," though we will try to point out where an identical or similar approach has been tried under a different name.

The second class of augmentations we experiment with involves prepending lexicon translations

to source sentences as additional cross-lingual signal, as instead of swapping out words in the source sentence. This approach has been tried as well for enhancing MT performance, e.g. in Song et al. (2019a); Maheshwari et al. (2022); Niehues (2021); Michon et al. (2020); Zhong and Chiang (2020); Susanto et al. (2020), and for similar tasks like language modeling (Yu et al., 2021). One potential advantage this approach has over the codeswitching method is that it can be applied at inference time as well: multilingual lexicon entries can be prepended to sentence queries to steer the model toward more accurate word translations. Outside of NMT models, this lexical prompting approach has also been applied to translation with LLMs: Ghazvininejad et al. (2023) provide LLMs with dictionary translations of some of the source sentence words, which the model can use to cover gaps in its vocabulary coverage (although the authors do not experiment with truly low-resource languages). With the rise in popularity of LLMs for MT and other tasks, this is an exciting area for further research.

## 3 Training data

Models are trained on sentence-level web text.

### 3.1 Monolingual data

The monolingual training data is from a clean, sentence-level web-mine following the approach set forth in Caswell et al. (2020). To make rapid training and development possible, we subsampled the monolingual data for the 100 highest-resource languages to 10% of its original size. In sum, this totaled about 4B sentences, or about 80B tokens. For the large model experiments in section 8.1, we used the full data, totaling 27B sentences (540B tokens).

### 3.2 Parallel data

We use an in-house mine of parallel data. It tends to be much noisier than the monolingual data. All parallel data are sampled to 10% of their original size, resulting in 9B parallel sentences (162M tokens) into English and the same number out of English, as well as 700K non-English-centric sentence-pairs. The large models in Section 8.1 use the full dataset.

### 3.3 Multilingual lexica

### 3.4 GATITOS

The GATITOS dataset is a new dataset open-sourced in this paper. It consists of 4000 short English seg-

ments translated into 170 very low-resource languages; however, at the time of the experiments in the paper, it covered only 24. The English source text is a mixture of words from a variety of sources, including frequent tokens in the English language, words for numbers, months, days of the week, Swadesh words, names of the languages themselves (including the endonym), and a few short sentences. The tokens were manually reviewed by the authors of this paper to ensure they looked reasonable. As the name implies, this dataset is mostly very short, consisting of 93% single tokens. There are also some short sentences, though only 23 entries have over 5 tokens. We hope this dataset will complement existing publicly available multilingual lexicons like MUSE (Conneau et al., 2017; Lample et al., 2017)

### 3.5 Panlex

Panlex (Kamholz et al., 2014) is a free, open-access massive online database consisting of word and phrase translations for 5000+ languages, sourced from 2500 individual dictionaries. Panlex contains $\approx 1.3B$ translations across all language pairs. For our experiments, we use a subset of the Panlex database covering 177 languages and containing 66M word pairs. Languages were chosen largely by availability of eval sets; details in Appendix J.1.

## 4 Evaluation

We use two translation evaluation sets: FLORES-200 (NLLBTeam et al., 2022; Goyal et al., 2021; Guzmán et al., 2019), an open-sourced evaluation set consisting of 2009 English Wikipedia sentences translated by humans into 200 languages, and GATONES, an in-house evaluation set of 1,200 English sentences translated into various languages. We use the SacreBLEU (Post, 2018) implementation of CHRF [2] for our evaluation metric. Higher-quality, embedding-based metrics like BLEURT (Sellam et al., 2020) are not available for these languages, and CHRF seems to be the one of best of the surface-level metrics for low-resource languages (Bapna et al., 2022; Kocmi et al., 2021; Freitag et al., 2022a).

For this study, we only evaluate on English-centric directions. The reason is that, although both evaluation sets are multi-way parallel, they are both also English-original, and thus lose the same infor-

---

[2]signature nrefs:1|case:mixed|eff:yes|nc:6|nw:0|space:no|version:2.3.1

mation that pivot translations do. Furthermore, this study places more weight on the en→xx direction than the xx→en direction. The main reason for this is that the xx→en direction is generally an easier direction for models to learn (since they see so much more English text), so the en→xx direction is usually the limiting reagent when it comes to model quality; as a result we care more about improving this direction.

# 5 Model

For our experiments we use a Transformer Big encoder-decoder model (Vaswani et al., 2017) with approximately 475M parameters. We train each model for 400K steps on 64 TPU v2 chips. Our models assign a 40% weight to the translation task, and a 60% weight to the MASS task. For models augmented with a monolingual data augmentation, we split the 60% weight on the MASS task into a 30% weight on the augmented data and a 30% weight on the non-augmented data. Parallel data augmentations were done in an analogous way. For the raw-token-pair augmentation, we add in this task with a 5% weight and shrink the other weights accordingly. We use a task-specific token for each of the tasks the models may see, namely translation, MASS, GlowupMono, GlowupParallel, CodeswitchMono, and CodeswitchParallel.

# 6 Methods

In this paper, we divide our augmentation approaches into two classes: "codeswitching" approaches, which involve substituting source sentence words for their dictionary translations, and "GLOWUP," which entails prepending dictionary translations of source words to source sentences. The main difference between these approaches is whether dictionary translations are substituted for source text (in the case of codeswitching) or added to the sentence (in the case of GLOWUP). As a third augmentation, we experiment with training on raw lexicon token pairs directly, treating them like any other parallel data.

The novelty of our contribution lies not so much in any one of our methods, but rather in (1) the application of these methods to unsupervised machine translation; (2) the number of methods we compare in controlled experiments; (3) the scale of our experiments, in terms of number of languages, data quantity, and model capacity; and (4) the application of these methods to "in the wild" web-crawled

data. While a variety of papers (e.g. Reid and Artetxe (2022); Yang et al. (2020)) have explored specific augmentations on particular language pairs, we believe our paper is the first to undertake a rigorous comparison of different augmentation strategies across hundreds of languages in a real-world setting.

## 6.1 Codeswitching

In our "codeswitching" augmentation strategy, words in the source sentence are swapped out for their dictionary translations to create mixed-language sentences. We experiment with this augmentation on both monolingual and parallel data. The details of this method are described below.

### 6.1.1 Codeswitched Autoencoding

Our multilingual codeswitching autoencoding (MCA) approach is similar to the "dictionary denoising" objective in Reid and Artetxe (2022). Let $D$ represent a multilingual lexicon containing word or phrase translation pairs for many languages. Given a source sentence $x = (x_1, x_2, ..., x_n)$ from monolingual corpus $X_{mono}$, we substitute each token in $x$ for its dictionary translation with probability $p_{tr} = 0.4$. (More implementation details in Appendix Section H). Note that Reid and Artetxe (2022) also apply additional noise to $x$ on top of codeswitching, along the lines of (m)BART (Lewis et al., 2020; Liu et al., 2020). For simplicity and so we can better examine the effects of lexicon information in isolation, we do not do this.

### 6.1.2 Codeswitching MT

Our codeswitching MT task is essentially the same approach as described in Section 6.1.1, except it applies to parallel rather than monolingual data. Given a source sentence $x$ from parallel corpus $X_{parallel}$, we perform the identical procedure described in section Section 6.1.1 to obtain multilingual codeswitched sentence $x'$. We then train the model on the translation task using sentence pairs $(x', y)$, where $(x, y)$ is a sentence pair in $X_{parallel}$. This method is effectively identical to the Random Aligned Substitution method proposed in Lin et al. (2020). As with MCA, we use $p_{tr} = 0.4$ and apply the augmentation on half the available parallel data.

## 6.2 Lexical prompting (GLOWUP)

The second class of lexical augmentations we experiment with is lexical prompting, which we

call GLOWUP (Guiding Lexical Outputs With Understandable Prompts). This method prepends $(src, transl)$ pairs to the beginning of a source sentence for some uniform random fraction of the words in that sentence that are in out bilex. These hints can then be used to help the model guess the translation or the denoised sentence. The GLOWUP task has the advantage that it can be used at inference time, without retraining a model, and may be simpler to implement. However, it does result in longer and less balanced sequence lengths, which can pose problems for decoding.

### 6.2.1 MASS with monolingual GLOWUP

To apply GLOWUP to monolingual text, we simply first sample the lexical prompts, and after those translations are prepended to the source, we apply MASS in the standard way to mask random subspans. This may mean that the GLOWUP prompt itself is masked.

### 6.2.2 GLOWUP-MT

The extension of GLOWUP to parallel data (GLOWUP-MT), is effectively the same as the monolingual variant of the task, but without the MASS element. For a given sentence pair $(x, y)$ in the training corpus, the prompting is performed on the source sentence $x$, essentially to give it hints about how to produce $y$. The model is then trained on the translation task using $(x', y)$, with the task token `<2glowup>` instead of `<2translation>`. Like GlowupMono, this can be applied in an inference-only way.

## 7 Experiments

### 7.1 Training regimes

In our experiments, we train models with various combinations of the augmentations outlined above, as well as a baseline (with no data augmentation of any kind) and a model where we simply provide word pairs from the lexicon as additional parallel data. The details of our training regimes are discussed below.

### 7.1.1 Baseline

We first train a baseline model with no data augmentation, using the monolingual and parallel data described in Sections 3.1 and 3.2, respectively. This model is comparable to the model trained in Bapna et al. (2022), but smaller, and without iterative back-translation; larger models with more data, are explored in Section 8.1.

### 7.1.2 Token-pair-only model

In addition to the baseline model, we also experiment with the extremely simple approach of providing raw word pairs from multilingual lexica to the model as additional parallel data. That is, given a dictionary entry $s$ and its translation $t$, we provide the model with a "sentence" pair of the form (`<2translation>` `<2lang>` `<2script>` $s, t$). We call this token-pair baseline GatiPanlexTokenPairs.

### 7.1.3 Single augmentation models

We also train models on each of the augmentations described in Section 6. As noted in that section, we only augment half the relevant data (monolingual, parallel, or both) before training each of these models, leaving the other half to be trained identically to the baseline (i.e. joint training on the MT task and MASS). The models with a single augmentation are named after their augmentation, viz. **CodeswitchMono**, **CodeswitchParallel**, **GlowupMono**, and **GlowupParallel**; those with two or more augmentations include them all in the name, viz. **CodeswitchMonoParallel**, **GlowupMonoParallel**, **CodeswitchMonoParallelGatiPanlex**, and **GlowupMonoParallelGatiPanlex**. We leave experimentation with a hybrid codeswitch-GLOWUP approach (e.g. CodeswitchMonoGlowupMono) for future work.

## 8 Results

We evaluate all our models on the FLORES-200 dataset (NLLBTeam et al., 2022; Goyal et al., 2021; Guzmán et al., 2019), which contains English-aligned parallel sentences for 200 languages. In Appendix section B we also report scores on the in-house GATONES eval set.

We use the following resourcedness classifications for our analysis:

1. High-Resource Languages (HRLs): $> 2B$ total training tokens in parallel data
2. Medium-Resource Languages (MRLs): 360M to 2B training tokens in parallel data
3. Low-Resource Languages (LRLs): 1 to 360M training tokens in parallel data
4. Unsupervised Languages (URLs): no parallel data

The results relative to the baseline, in $\Delta$CHRF, are summarized in Tables 1 (en→xx) and 2 (xx→en). A few trends jump out. Firstly, all models trained with only monolingual data augmentations see consistent performance gains over

| Model | $\mu$ | HRL | MRL | LRL | URL | $\text{LRL}_{GAT}$ | $\text{URL}_{GAT}$ |
|---|---|---|---|---|---|---|---|
| Baseline | 39.7 | 50.5 | 46.6 | 34.4 | 28.7 | 35.4 | 26.6 |
| GatiPanlexTokenPairs | +0.4 | -0.3 | -0.1 | +0.5 | +1.4 | +1.8 | +5.1 |
| CodeswitchMono | +0.8 | +0.8 | +0.9 | +0.8 | +0.4 | +1.9 | +4.8 |
| CodeswitchParallel | -1.4 | -1.9 | -1.7 | -1.2 | -0.9 | +0.2 | +2.1 |
| CodeswitchMonoParallel | -0.2 | -1.0 | -0.7 | +0.0 | +1.1 | +1.5 | +5.2 |
| CodeswitchMonoGatiPanlex | +1.0 | +0.4 | +0.6 | +1.3 | **+1.5** | **+2.8** | **+7.0** |
| GlowupMono | +1.1 | **+1.5** | **+1.4** | +1.0 | +0.5 | +2.2 | +5.6 |
| GlowupParallel | -1.1 | -2.0 | -1.8 | -1.1 | +0.4 | +0.8 | +3.0 |
| GlowupMonoParallel | +0.3 | +0.1 | +0.2 | +0.1 | +1.0 | +1.3 | +3.4 |
| GlowupMonoGatiPanlex | **+1.2** | +1.3 | +1.2 | **+1.4** | +0.8 | +2.5 | +5.6 |

Table 1: en→xx performance on the FLORES-200 test set, measured in $\Delta$CHRF over the baseline. Gains are particularly strong on the languages with GATITOS (last columns), reaching +7 CHRF for unsupervised language pairs.

| Model | $\mu$ | HRL | MRL | LRL | URL | $\text{LRL}_{GAT}$ | $\text{URL}_{GAT}$ |
|---|---|---|---|---|---|---|---|
| Baseline | 47.2 | 57.2 | 52.1 | 43.6 | 37.0 | 49.1 | 36.1 |
| GatiPanlexTokenPairs | -0.0 | -0.3 | -0.3 | +0.1 | **+0.5** | +1.0 | +2.4 |
| CodeswitchMono | +0.2 | +0.5 | +0.4 | +0.1 | -0.1 | +0.8 | +1.5 |
| CodeswitchParallel | -1.2 | -1.6 | -1.5 | -1.2 | -0.3 | +0.0 | +1.2 |
| CodeswitchMonoParallel | -0.8 | -0.7 | -0.9 | -0.8 | -0.7 | -0.1 | +1.1 |
| CodeswitchMonoGatiPanlex | +0.1 | +0.3 | +0.1 | +0.3 | -0.1 | +0.9 | **+2.9** |
| GlowupMono | **+0.7** | **+1.2** | **+1.1** | +0.6 | +0.0 | **+1.5** | +2.0 |
| GlowupParallel | -1.3 | -1.7 | -1.6 | -1.2 | -0.6 | -0.1 | +1.2 |
| GlowupMonoParallel | +0.1 | +0.0 | -0.0 | -0.0 | **+0.5** | +0.6 | +1.7 |
| GlowupMonoGatiPanlex | +0.6 | +1.0 | +0.8 | **+0.6** | -0.1 | +1.3 | +1.8 |

Table 2: xx→en performance on the FLORES-200 test set, measured in $\Delta$CHRF over the baseline, showing a weaker version of the same trends from en→xx.

the baseline. Conversely, models with only parallel data augmentations show performance degradations. Models mixing monolingual and parallel data augmentations fare in-between those poles.

These general trends are the same between en→xx and xx→en directions, though the gains are generally lower in the xx→en direction. As noted in Section 4, this is expected, and this direction is less of a priority for translation improvements. Results on GATONES, in Appendix B, show the same trends, though the performance gains tend to be larger for all augmentations, with CodeSwitchMonoGatiPanlex gaining +2.3 CHRF for URLs.

Despite these trends seeming robust across models, the effect sizes are relatively small, maxing out at about +1.5 average $\Delta$CHRF gain. However, the picture changes dramatically when only looking at the subset of languages that has the higher-quality GATITOS training lexica. For these 26 languages, every augmentation, even the parallel ones, have large performance gains. The winning augmentations for URLs remain CodeSwitchMonoGatiPanlex and GlowupMonoGatiPanlex, the former having an average gain of +7.0CHRF on FLORES-200and +8.0CHRF on GATONES, and the latter having +5.6CHRF on FLORES-200 and +9.5CHRF on GATONES.

Finally, although it was not the first place model in any category, the GatiPanlexTokenPairs has large gains in all directions over baseline, and only falls short of the more complex augmentations by a small margin. Furthermore, when used in conjunction with either Glowup or Codeswitch, it further improves performance across all categories. This may be one of the most useful long-term findings of this study: raw token pairs perform roughly on par with all the other fancier augmentations!

## 8.1 Scaling up: bigger models, more data

Sometimes, results on smaller models do not transfer to larger models. To this end, we train larger transformer models with 1.6B parameters using

$10\times$ the parallel data and $10\times$ the high-resource monolingual data.

We trained three large models: a baseline, a token-pair model, and a token-pairs + Codeswitch-Mono model. The CHRF scores for these models can be seen in Table 3. There are a number of obvious differences between these results and the results on the smaller (475M parameter) models trained with $\approx \frac{1}{10}$th the data. First, the positive impact of the data augmentations is smaller in all categories, indicating that the gains previously seen from augmentations are partially washed out in the larger-data, larger model regime. On FLORES-200 en→xx, both models are very close to baseline—within the realm of noise. On GATONES en→xx, they see consistent small gains of around $+0.5$ CHRF, much smaller than previously. For both eval sets in the xx→en direction, there are consistent small losses.

Is this the Bitter Lesson (Sutton, 2019) getting us again? Perhaps—but the picture is less bleak than it first appears. When we look at the subset of the languages where we have a higher assurance that the bilingual lexica are higher-quality—namely, those that use GATITOS bilingual lexica—we still see consistent wins.For these 26 languages, all models see consistent gains, and the gains are biggest on unsupervised languages.

Overall, the takeaway from these experiments is that one has to ensure that the data is of high quality when applying lexical data augmentation at such a large scale. While we saw substantial improvements for many languages, these were balanced out by losses for other languages (especially those with only Panlex, but not GATITOS, data).

## 8.2 Glowup Decoding

In principle, one of the advantages of the Glowup-Parallel approach is that lexica can be used at inference time. Therefore, we experimented with decoding the eval sets not with the translation task ID, but with the Glowup task ID, along with the relevant lookups from the lexica. Unfortunately, these decodes failed impressively, with performance degrading the more prompts that were included. Model decodes often had long sequences of control tokens. Further work should not disregard this direction; indeed, a variant of this likely has particular promise in the world of foundation models. The current approach likely just needs some tweaks to eliminate this sort of out-of-domain

decoding errors we were seeing, but we leave an investigation of this hypothesis for future work.

## 8.3 Oracles: trusted parallel text

Parallel text is much more costly to produce than bilingual lexica, but also contains many more useful signals, including examples of word usage in context. But how much more helpful is it, really, than bilingual lexica? The answer seems to be "much more helpful".

To measure this, we trained a model with a mixture of thirteen public parallel datasources (Appendix A) covering our lowest-resource languages. These are high-quality, trusted datasets, prepared by community members – a very different resource than the web-mined parallel data that the model is otherwise trained on.

Table 4 reports on the 24 GATITOS languages, comparing four models: the standard baseline and GatiPanlexTokenPairs model, as well as the "Parallel" model (which adds the external parallel translation task with a 5% weight) and the "Parallel + GatiPanlexTokenPairs" model, which uses both the token-pairs and the parallel data, with a combined weight of 5%. On both eval sets, we see that using Bilex yields a gain of around +3.5 CHRF, but using true parallel yields a much larger gain of about +10 CHRF. Using the bilingual lexica on top of the parallel data yields a further gain of about +0.5 CHRF, demonstrating that, though many of the gains have been washed out by the true parallel data, there are still modest gains to be had from bilex training. Full results are in Appendix Table 9.

## 8.4 How many token pairs do I really need?

We examine the relationship between the number of lexical token pairs provided during training and CHRF. First, we perform regressions using $\Delta$CHRF over baseline as the outcome variable and three predictor variables: (1) number of Panlex entries, (2) number of GATITOS entries, and (3) number of monolingual sentences. We include monolingual sentences in the regression to control for it as a confound. To eliminate parallel data quantity as a confound, we only perform this analysis on URLs.

As expected, both the number of Panlex word pairs for a given language and the number of GATITOS word pairs have a positive $\beta$ coefficient. However, note that the $\beta$ for GATITOS is $\approx 3\times$ larger than $\beta$ for Panlex on FLORES-200, and $\approx 39\times$ larger for GATONES. We conclude that GATITOS is

| | HRL | MRL | LRL | URL | LRL$_{GAT}$ | URL$_{GAT}$ |
|---|---|---|---|---|---|---|
| FLORES-200 en→xx | | | | | | |
| BaselineBig | 56.1 | 51.2 | 38.5 | 35.5 | 24.4 | 36.0 |
| GatiPanlexTokenPairsBig | -0.5 | -0.3 | +0.1 | -0.3 | +1.2 | +2.2 |
| CodeswitchMonoGatiPanlexBig | +0.1 | +0.1 | +0.3 | -0.3 | +0.9 | +3.4 |
| GATONES en→xx | | | | | | |
| BaselineBig | - | 34.1 | 26.9 | 25.5 | 23.9 | 27.1 |
| GatiPanlexTokenPairsBig | - | +0.2 | +0.6 | +0.7 | +1.6 | +3.5 |
| CodeswitchMonoGatiPanlexBig | - | +0.4 | +0.7 | +0.2 | +1.4 | +4.2 |

**Table 3:** Average CHRF scores by resource category for the larger models, reported in delta relative to baseline. These models are trained with 10x the data and 3x the parameters. The gains as a whole are washed out somewhat, but for those low-resource and unsupervised language pairs with the more trusted GATITOS data (last two columns), there is still a noticeable gain.

| | Baseline | GatiPanlexTokenPairs | Parallel | Parallel + GatiPanlex |
|---|---|---|---|---|
| FLORES-200 | 21.1 | 24.4 | 33.6 | **34.2** |
| GATONES | 20.2 | 23.9 | 29.5 | **30.1** |

**Table 4:** Comparing en→xx improvements from token-pair data to the oracle: training on trusted parallel data. Parallel data is much more effective than GATITOS alone, but the combination of the two is the most effective overall.

more efficient for improving MT than Panlex, probably due to higher quality. Full regression results are given in Appendix Table 6.

The most practical question we can seek to answer is, **If I can spend \$X on translating tokens, how much quality increase can I expect?** To investigate this "bang for buck" question in a more controlled way, we observe the effects of the GATITOS dataset in isolation, without Panlex. We train a "GatiTokenPairs" model, which is identical to the "GatiPanlexTokenPairs" model, except the token-pair task has only GATITOS data. Thus, this tells us specifically what gains we can expect if we are to get 4,500 tokens' worth per language.

The result of this experiment is a gain of +4.9 CHRF on FLORES-200 and +5.2 CHRF on GATONES, respectively, for en→xx URLs; the improvement for languages with some parallel data is reduced but significant, at +1.6/2.3 CHRF resp. Full results reported in Appendix Table 10.

## 9 Conclusions

In this paper we explore the ways that that augmenting training data with bilingual lexicon information can improve the performance of machine translation models on low-resource and unsupervised languages, and open-source the GATITOS dataset, which leads to average gains of about +7 CHRF on unsupervised languages. We perform extensive experimentation with three main types of lexical augmentation: codeswitching, lexical

prompting, and raw token-pair training. The results show that applying any of these augmentations to monolingual data yields substantial improvements, and that they can be combined for greater effect. The leader (by a small margin) is the combination of CodeswitchMono and raw token-pair training. These results hold when scaling up model and data size, but in the settings with more data and larger models, the quality of the bilingual lexica plays a relatively bigger role, and augmentation with the noisier Panlex begins to lag in quality behind the much smaller, yet higher-quality, GATITOS dataset.

Future work will likely want to focus on prompting foundation models with bilingual lexica. Large Language Models show promise on machine translation for high-resource languages (Jiao et al., 2023), but their capabilities on low-resource languages have yet to be thoroughly explored. Additionally, a more thorough investigation of the trade-off between cost and quality for tiny datasets can be explored: with a fixed budget of time or money, should one spend their time translating text, making monolingual text, or making bilingual lexica?

## 10 Limitations

There are several limitations with the present work. For one, we rely on the automated metric CHRF, which is less reliable than human ratings. Similarly, though we perform some qualitative evaluation of error types (Appendix G), a detailed human eval-

uation might reveal the precise ways in which our models are failing.

A second limitation is that, while we open-source GATITOS, the base training data from our models is not opensourced (to protect copyright), and thus our experiments are not replicable.

Finally, we leave open two important questions: 1) how much data do I need translated before I can reach X quality? and 2) if I can only translate X tokens' worth of data, what is the best way to select those X tokens? The present work only partially answers the first question, and does not address the second at all.

## 11  Ethics Statement

Improving the state of technology for under-served communities is usually considered a positive contribution. There are various nuances to this, however, including questions of consent of and involvement of the affected community, "helicopter NLP," and data sovereignty. By open-sourcing GATITOS to be used by all affected communities, we hope to respect their data sovereignty and not keep this resource from them. Throughout this project we have also sought advice from community members, which we hope will alleviate the other concerns.

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

Xing Jie Zhong and David Chiang. 2020. Look it up: Bilingual and monolingual dictionaries improve neural machine translation. *CoRR*, abs/2010.05997.

**GlowupMono** w/ masking (multilingual lexically-prompted MASS)

```
Source    <2glowup> <2en> <2Latn> <src> we <tgt> nous <src> see <tgt> ka
          ye <2en> <MASK> Saturday, we drove <MASK> <MASK> see <MASK> countryside.

Target    Last Saturday, we drove out to see the countryside.
```

**GlowupParallel** (target-language lexically-prompted translation)

```
Source    <2glowup> <2fr> <2Latn> <src> came <tgt> venu <src> dramatic
          <tgt> dramatique <src> fashion <tgt> façon <2fr> The performance came to
          a close in dramatic fashion.

Target    La performance s'est conclue de façon dramatique.
```

**Figure 1:** Examples of the monolingual (top) and parallel (bottom) GLOWUP augmentation strategies. In both cases, random tokens in the source sentence are prepended to the source sentence, along with their translations in a random language from a multilingual lexicon. As in Figure 2, differential color coding is done to draw attention to dictionary translations in different languages.

**CodeswitchMono** (multilingual codeswitching autoencoding)

```
Source    <2codeswitch> <2en> <2Latn> Dernier Saturday, noi yatwaye out to
          हेनुहोस् the panpa.

Target    Last Saturday, we drove out to see the countryside.
```

**CodeswitchParallel** (codeswitching translation)

```
Source    <2codeswitch> <2fr> <2Latn> The പ്രകടനം came til a близкий in
          dramatisk fashion.

Target    La performance s'est conclue de façon dramatique.
```

**Figure 2:** Examples of the monolingual (top) and parallel (bottom) codeswitching augmentation strategies. In both cases, random tokens in the source sentence are replaced with their translations in a random language from a multilingual lexicon. Color coding is used to indicate which source words have been swapped for their dictionary translation. The different colors are used simply to point out the fact that the words are from codeswitched words in each sentence come from different languages.

## A  Public parallel data used

The datasets we used were HornMT (Hadgu et al., 2022), SALT (Sunbird AI Language Translation) dataset (Akera et al., 2022), FFR: Fon-French Neural Machine Translation (Emezue and Dossou, 2020), Tatoeba (Tiedemann, 2020), The Makerere MT Corpus: English to Luganda parallel corpus (Mukiibi et al., 2021), Commonvoice (Ardila et al., 2020), Kencorpus: Kenyan Languages Corpus (Wanjawa et al., 2022), Chuvash-Russian parallel corpus( Antonov, Alexander, 2022), Abkhaz Corpus (Tlisha, Nart, 2022), Bashkir Corpus(Shakirov and Kunafin, 2023), Sprotin Faroese Corpus(Andersen, Jógvan, 2021), Jojajovai Guarani-Spanish Parallel Corpus (Chiruzzo et al., 2022), and the NLLB Seed data (NLLBTeam et al., 2022). We were not able to use AmericasNLP (Mager et al., 2021) because of the license.

| Source | Khawvelah hian tawng chi hrang 5000 chuang a awma, heng zinga tawng 20 ai tam hi chuan tawng hmangtu maktaduai 50 aia tam a nei a ni. |
|---|---|
| Reference | The world has over 5,000 different languages, including more than twenty with 50 million or more speakers. |
| Baseline | There are 5000 houses in the city, 5000 houses in the city, and 5000 houses in the city. |
| Augment | There are 5000 languages in the world, 20 of which are 50 million languages. |
| Source | Maani isumaqarpunga inuunerma sinnerani pilluartussanngorlunga. |
| Reference | Here I think I'll remain happy throughout my life. |
| Baseline | I believe I am the only one who has ever died. |
| Augment | I mean, I'm blessed with a lifetime of joy. |
| Source | нагахь вы ж хьашто хилар в дӀадахар къамелан ма расширяемости, дазар с нами хула uservoice |
| Reference | If you are interested in continuing the conversation around extensibility, connect with us via uservoice |
| Baseline | If you want to use the extension, see uservoice |
| Augment | If you want to learn more about extensibility, contact us at uservoice |

**Table 5:** Examples of translations where our lexical data augmentation methods appear to have helped the model choose the correct vocabulary, in Mizo (lus), Kalaallisut (kl), and Chechen (ce). Words appearing in the lexical data are colored green.

| | No. PanLex | No. GAT. | No. mono | Intercept | Adjusted $R^2$ |
|---|---|---|---|---|---|
| $\Delta$FLORES-200 | 6.55e-5 | 1.96e-4 | $-6.71$e-7 | 1.93 | 0.40 |
| $\Delta$GATONES | 6.59e-6 | 2.58e-4 | $-7.78$e-8 | 2.91 | 0.23 |

**Table 6:** Linear regression results in the en→xx direction. Here, the dependent variable is $\Delta$CHRF over the baseline, and the independent variables are: number of PanLex word pairs for a given language (No. PanLex), number of GATITOS word pairs (No. GATITOS), and number of monolingual sentences (No. mono).

## B  Results on GATONES

The main paper reports the scores on the more widely-used FLORES-200 eval set; this section reports on the other dataset that we evaluate our models on. This is an in-house eval set, which we call GATONES (Google AuTOmatic NTL Eval Set). The dataset contains 63 languages, most of which are unsupervised or low-resource. Here are some notes, by translaiton direction:

**en→xx** The evaluation results on GATONES for the en→xx direction are summarized in Table 7. The trends are mostly the same as what we saw for the FLORES-200 en→xx evaluation. Codeswitch-MonoGatiPanlex emerges, even more definitively than on FLORES-200, as the best model, winning all categories except MRLs.

**xx→en** The evaluation results for the xx→en direction are given in Table 8. The single most important takeaway from this part of the analysis is the same as it was for the FLORES-200 evaluation: the plain GatiPanlexTokenPairs model helps URLs the most in this direction, with a $\Delta$CHRF of $+1.1$ over the baseline. Yet again, the improvements are smaller in this direction than for en→xx. The only other thing that stands out about this part of the evaluation is that the GlowupMono augmentation doesn't seem to be as helpful according to this test set as for the FLORES-200 set. Although GlowupMonoParallel and GlowupMonoGatiPanlex do reasonably well, their improvements are sig-

nificantly smaller than the improvement from using GatiPanlexTokenPairs alone, and the GlowupMono augmentation by itself actually results in losses on URLs. So taking the GATONES and FLORES-200 results together, it seems that adding raw token pairs as additional parallel data is the best way, out of the techniques we tried, to improve performance in the xx→en direction for very low-resource languages.

## C  Bilexica vs true parallel data

In the main paper in Section 8.3, we evaluate on the 24 GATITOS languages, comparing four models with different combinations of trusted parallel data and GATITOS bilingual lexica. The full results are given here in Table 9, on both FLORES-200 and GATONES, sorted from highest-resource to lowest-resource.

| Model | $\mu$ | MRL | LRL | URL | $\text{LRL}_{GAT}$ | $\text{URL}_{GAT}$ |
|---|---|---|---|---|---|---|
| Baseline | 21.4 | 28.7 | 22.9 | 19.1 | 17.1 | 20.0 |
| GatiPanlexTokenPairs | +1.7 | +0.6 | +1.6 | +1.8 | +3.6 | +8.4 |
| CodeswitchMono | +1.4 | **+1.6** | +0.8 | +1.9 | +3.2 | +8.3 |
| CodeswitchParallel | -0.5 | -1.8 | -0.9 | -0.0 | +0.9 | +3.4 |
| CodeswitchMonoParallel | +1.4 | -0.1 | +0.7 | +2.2 | +2.5 | +6.1 |
| CodeswitchMonoGatiPanlex | **+2.2** | +1.1 | **+2.1** | **+2.3** | **+3.9** | +8.0 |
| GlowupMono | +0.3 | +1.0 | +0.6 | -0.2 | +2.0 | +7.6 |
| GlowupParallel | -0.2 | -1.6 | -0.7 | +0.6 | +1.0 | +3.0 |
| GlowupMonoParallel | +0.2 | -0.3 | -0.7 | +1.1 | +2.1 | +7.6 |
| GlowupMonoGatiPanlex | +1.3 | +1.4 | +1.9 | +0.7 | +3.5 | **+9.5** |

**Table 7:** en→xx performance on the GATONES test set, measured in $\Delta$CHRF over the baseline.

| Model | $\mu$ | MRL | LRL | URL | $\text{LRL}_{GAT}$ | $\text{URL}_{GAT}$ |
|---|---|---|---|---|---|---|
| Baseline | 29.4 | 43.2 | 30.2 | 27.0 | 24.7 | 22.1 |
| GatiPanlexTokenPairs | **+0.8** | +0.0 | **+0.5** | **+1.1** | **+1.1** | +3.2 |
| CodeswitchMono | +0.1 | +0.1 | +0.0 | +0.2 | +0.8 | +2.2 |
| CodeswitchParallel | -0.4 | -1.2 | -0.9 | +0.2 | +0.3 | +1.8 |
| CodeswitchMonoParallel | -0.5 | -0.8 | -0.7 | -0.2 | +0.2 | +1.6 |
| CodeswitchMonoGatiPanlex | +0.5 | +0.0 | **+0.5** | +0.6 | **+1.1** | **+3.3** |
| GlowupMono | -0.0 | **+0.9** | +0.2 | -0.4 | +0.5 | +1.7 |
| GlowupParallel | -0.6 | -1.5 | -0.9 | -0.3 | +0.0 | +1.3 |
| GlowupMonoParallel | +0.0 | -0.2 | -0.2 | +0.2 | +0.2 | +1.6 |
| GlowupMonoGatiPanlex | +0.2 | +0.8 | +0.3 | +0.1 | +0.8 | **+3.3** |

**Table 8:** xx→en performance on the GATONES test set, measured in $\Delta$CHRF over the baseline.

| | mean | ts | nso | lg | ee | bho | bm | ff | gn | ti | om |
|---|---|---|---|---|---|---|---|---|---|---|---|
| en→xx Parallel toks | 375169 | 2296670 | 606853 | 315905 | 275417 | 154023 | 148549 | 130953 | 112958 | 43511 | 42018 |
| en→xx Bilex toks | 16324 | 8015 | 4500 | 8298 | 6755 | 24665 | 24665 | 70854 | 4500 | 6484 | 4500 |
| FLORES-200 en→xx | | | | | | | | | | | |
| Baseline | 21.1 | 33.2 | 31.9 | 30.1 | 26.9 | 14.7 | 15 | 19.1 | 19.1 | 5.9 | 15.5 |
| GatiPanlexTokenPairs | 24.4 | 37.3 | 31.9 | 32.7 | 30.5 | 25.5 | 22.6 | 19.6 | 19.4 | 8.1 | 16.1 |
| Parallel | 33.6 | 45.6 | 48 | 39.4 | 28 | 40.6 | 30.7 | 25 | 39.2 | 15.7 | 24.1 |
| Parallel + GatiPanlex | 34.2 | 45.3 | 45.3 | 38.2 | 31.1 | 40.8 | 30.1 | 24.9 | 39.9 | 18.1 | 28.8 |
| GATONES en→xx | | | | | | | | | | | |
| Baseline | 20.2 | 32.1 | 28.3 | 27.6 | 23.5 | 15.8 | 13.5 | 23.9 | 15.6 | 6.9 | 14.6 |
| GatiPanlexTokenPairs | 23.9 | 37.3 | 28.7 | 31.3 | 28.1 | 26.2 | 22.5 | 23.5 | 16.1 | 9.3 | 15.6 |
| Parallel | 29.5 | 42.9 | 39.9 | 35.5 | 26.9 | 38.5 | 30.3 | 23.6 | 28.2 | 11.1 | 18 |
| Parallel + GatiPanlexTokenPairs | 30.1 | 42.6 | 37.9 | 35 | 29.6 | 38.7 | 30.4 | 22.6 | 30.1 | 13.1 | 20.8 |

**Table 9:** Comparing improvements from token-pair data to the oracle: training on trusted parallel data.

# D Training only on GATITOS

The main paper (Section 8.4) describes the results of training models only on GATITOS and not on PanLex. Full results are reported here, in Table 10, reported in delta versus the baseline model for both FLORES-200 and GATONES in the en→xx direction. The improvement for unsupervised languages is around +5.0 CHRF for both eval sets; the improvement for languages with some parallel data is less but still noticeable, hovering around +2 CHRF. The largest improvement is in Goan Konkani (+11.0 CHRF), with Mizo, Ilocano, and Bambara close on its heels with gains of around +8 CHRF. Only Maithili, which has interesting properties a a close dialect of Hindustani, sees a loss on both eval sets. The gains are not obviously related to the total number of tokens per language.

As an aside, it is heartening that FLORES-200 and GATONES seem to agree very nicely, despite their different domains (Wikipedia versus web + question-answers).

| Total Tokens | | | 1.8M | 2.1M | 2.6M | 3.6M | 3.7M | 5.4M | 6.2M | 14.7M | 14.8M | 16M | 16.9M | 26.1M |
|---|---|---|---|---|---|---|---|---|---|---|---|---|---|---|
| | $\mu_{lrl}$ | $\mu_{url}$ | ff | mni-Mtei | kri | doi | bm | ay | gom | bho | kl | ee | qu | ts |
| Δ Flores-200 | **+1.6** | **+4.9** | +0.1 | - | - | - | +5.0 | +0.8 | - | +4.5 | - | +2.9 | - | +4.2 |
| Δ Gatones | **+2.3** | **+5.2** | +0.0 | +0.7 | +5.1 | +1.5 | +7.6 | -0.1 | +11.0 | +5.1 | +4.8 | +3.9 | +1.7 | +5.3 |

| Total Tokens | 26.4M | 27M | 28M | 40M | 41M | 52M | 52M | 80M | 115M | 124M | 157M | 167M | 204M | 505M |
|---|---|---|---|---|---|---|---|---|---|---|---|---|---|---|
| | ak | mai | ln | lg | gn | nso | ilo | ti | om | sa | dv | lus | as | ckb |
| Δ Flores-200 | - | -1.2 | +1.3 | +1.6 | -0.4 | -0.5 | +7.0 | +0.9 | -0.2 | +0.2 | - | +8.6 | +3.0 | +3.3 |
| Δ Gatones | +2.0 | -1.2 | 1.2 | +3.0 | +0.5 | +0.2 | +8.0 | +1.7 | +0.1 | +0.4 | +6.1 | +9.3 | +3.4 | +2.7 |

**Table 10:** Improvements on languages with GATITOS data when training ONLY on GATITOS data, in the en→xx direction. Sorted by total training tokens (mono, parallel, and bilex), in millions.

# E   Effects on distributionally similar noun mistranslation

Part of the motivation for using bilingual lexicons for unsupervised translation was to see whether we could repair the common error mode of mistranslating distributionally similar nouns. Bapna et al. (2022) note that this error mode is particularly common for two categories of nouns: animals and colors.

To measure improvement on this phenomenon, we define the *token hit-rate* as the following: for some set of desired tokens $D$, let $R_D$ be the subset of the eval set such that each reference contains at least one token in $D$. The hit-rate is then the percentage of times in $R_D$ that the model correctly generated one of the desired tokens in $D$. For instance, if the desired tokens are "kitten" and "puma", $R_D$ is the set of references containing one of these words, e.g. "The **kitten** lies" and "A **Puma** eats hot chip". If the model produces "**kitten** lie on floor" and "**Crocodile** charge they phone" from the corresponding sources, it has a hit-rate of 50%, since it correctly got "kitten", but not "puma".

Table 11 looks at the token hit-rate for the models **BaselineBig** and **CodeswitchMonoGatiPanlexBig**, for the categories of animals occurring in GATITOS (*bear, bee, bird, butterfly, cat, chicken, deer, dog, elephant, fish, frog, goat, horse, insect, lion, monkey, parrot, pig, rabbit, sheep, snail, snake, tiger, turkey, turtle*), animals NOT appearing in GATITOS (*ant, antelope, buffalo, cheetah, crocodile, dolphin, dormouse, gorilla, jellyfish, koala, leopard, moose, mosquito, newt, ocelot, otter, reindeer, robin, scorpion, shark, sloth, spider, springbok, tortoise, velociraptor*), colors (*black, white, red, blue, yellow, green, purple, orange, grey*), and numbers (*one, two, three, four, five, six, seven, eight, nine, ten, hundred, million*). All numbers and colors appear in GATITOS. Numbers are included as a weak control, since the model will

tend to make fewer mistakes on them, though such UNMT-style mistakes do occur.

As expected, the GATITOS-augmented model performs better on these tokens. Two things are worth noting. First, the model improves noticeably on the complementary distribution—words that do not appear in the lexicon training data—but unsurprisingly improves more on the words that are present in GATITOS. Second, the improvements are not as large as expected: why is it not now getting 100% accuracy? Digging into the outputs, it seems that this is mainly due to the high baseline of (a) undertranslation; (b) hallucination; and (c) copying, as we expect from a model trained without various other tricks like back-translation (see Section G for an analysis of common error types). This point is underscored by the models' imperfect performance on the "easy" class of numbers.

# F   Biggest winners

We also look at the top 5 languages that were the biggest gainers over the baseline for each model. In some cases these may represent remarkable successes of a particular approach—though in other cases they may represent noisy outliers, as is to be expected when evaluating 200 languages.

## F.1   FLORES-200

**en→xx**   The biggest winners for each model in the en→xx direction for the FLORES-200 evaluation set are given in Table 12.

There are seven languages that gained at least 5 CHRF over the baseline on at least one model trained with data augmentation. These languages are:

1. Bhojpuri (bho): up to +14.5 CHRF
2. Ilocano (ilo): up to +9.1 CHRF
3. Serbian (sr): up to +8.3 CHRF
4. Bambara (bm): up to +8.1 CHRF
5. Tibetan (bo): up to +8.0 CHRF

|  | FLORES-200 | | | | GATONES | | | |
| --- | --- | --- | --- | --- | --- | --- | --- | --- |
| cat. | $A \in$ GAT. | $A \notin$ GAT. | colors | #s | $A \in$ GAT. | $A \notin$ GAT. | colors | #s |
| BaselineBig | 36.6 | 36.6 | 55.3 | 63.3 | 33.4 | 25.8 | 35.4 | 53.5 |
| CodeswitchMonoGatiPanlexBig | 45.5 | 40.7 | 66.5 | 66.8 | 44.9 | 32.5 | 47.8 | 58.0 |
| $\Delta$ | +8.9 | +4.0 | +11.1 | +3.6 | +11.5 | +6.7 | +12.4 | 4.5 |

**Table 11:** Comparing token hit-rate on classes of nouns known to have issues for UNMT models, along with a weak control of numbers. There are large improvements for animals in the GATITOS training lexicon ($A \in GAT$), as well as their complementary distribution, animals not in GATITOS ($A \notin GAT$), and colors. Number hit-rate has a minor bump.

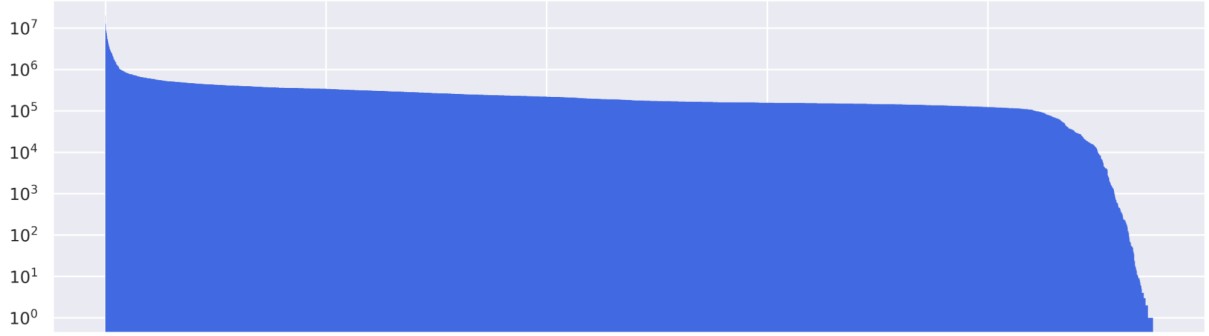

**Figure 3:** Number of word pairs available in Panlex for each of 4750 BCP-47 languages (log scale).

6. Nuer (nus): up to +6.8 CHRF
7. Mizo (lus): up to +6.2 CHRF

Unsurprisingly, most of these languages are unsupervised or low-resource, except for Serbian which is medium-resource in our dataset. Of the seven languages listed above, we use Panlex data for Ilocano, Serbian, Bambara, Tibetan, Nuer, and Mizo, and there is GATITOS data for Bhojpuri, Ilocano, Bambara, and Mizo. As will be discussed in Section 8.4, the GATITOS bilingual lexica are clearly a very useful resource for MT, although evidently Panlex alone can help as well. Another interesting finding is that Nuer, which has *no* English-aligned entries in Panlex but ≈ 20K non-English-aligned entries, still sees large improvements when translating from English. This is evidence that lexicon data can improve performance even in the zero-shot case, where e.g. the model learns better vocabulary alignment between English and Nuer despite not receiving explicit alignment information during training. In Section 8.4, we look at the relationship between the number of lexical data points for a language and the CHRF improvement, which provides some insight (albeit not perfect clarity) into why these particular languages did well.

**xx→en** Table 13 shows the top 5 biggest winners per model for the xx→en direction. Clearly there is a lot of overlap with the en→xx direction, although there are some differences. Also, note

again that the magnitude of improvement in this direction is smaller, likely because the baseline performance is higher and there is less improvement to be made simply by better aligning vocabulary cross-linguistically. Some of the biggest winners in this direction that weren't already discussed for the en→xx direction are:

1. Tsonga (ts): up to +3.1 CHRF
2. Guarani (gn): up to +2.8 CHRF
3. Bashkir (ba): up to +2.5 CHRF
4. Minangkabau (min): up to +2.5 CHRF

### F.2 GATONES

**en→xx** The biggest winners on GATONES in the en→xx direction are given in Table 14. Though there is some overlap with the biggest winners on the FLORES-200 dataset (e.g. Ilocano, Bambara, Mizo, Bhojpuri), a number of different languages perform well too, some of which simply aren't included in the FLORES-200 set. The languages which gain > 5.0 CHRF on this part of the evaluation are:

1. Adyghe (ady): up to +14.1 CHRF
2. Kedah Malay (meo): up to +12.6 CHRF
3. Goan Konkani (gom): up to +11.6 CHRF
4. Bhojpuri (bho): up to +10.4 CHRF
5. Ilocano (ilo): up to +9.8 CHRF
6. Avar (av): up to +9.5 CHRF
7. Bambara (bm): up to +9.0 CHRF
8. Mizo (lus): up to +8.9 CHRF

| Model | | | | | |
|---|---|---|---|---|---|
| GatiPanlexTokenPairs | ilo (+7.5) | nus (+6.8) | lus (+6.2) | bm (+5.5) | ts (+4.4) |
| CodeswitchMono | bho (+8.6) | bo (+5.6) | lus (+5.2) | ilo (+5.0) | quy (+4.9) |
| CodeswitchParallel | ilo (+3.7) | bho (+3.5) | as (+2.4) | lus (+2.3) | min (+1.9) |
| CodeswitchMonoParallel | bho (+9.4) | ilo (+6.4) | bo (+6.2) | lus (+6.0) | ln (+4.9) |
| CodeswitchMonoGatiPanlex | ilo (+9.1) | sr (+8.3) | bm (+8.1) | bho (+7.9) | bo (+6.8) |
| GlowupMono | sr (+6.2) | acq (+4.5) | bo (+3.8) | aeb (+3.1) | bho (+3.1) |
| GlowupParallel | bho (+4.4) | shn (+3.8) | sg (+3.7) | kac (+3.6) | kpb (+3.5) |
| GlowupMonoParallel | bho (+12.3) | sr (+7.1) | sg (+4.5) | bo (+4.1) | nus (+3.3) |
| GlowupMonoGatiPanlex | bho (+14.5) | bo (+8.0) | sr (+7.3) | bm (+5.7) | acq (4.2) |

**Table 12:** Top 5 biggest winners per model (en→xx) on the FLORES-200 test set, measured in $\Delta$CHRF over the baseline.

| Model | | | | | |
|---|---|---|---|---|---|
| GatiPanlexTokenPairs | bm (+3.5) | ts (+3.1) | lus (+2.7) | gn (+2.6) | tum (+2.1) |
| CodeswitchMono | bo (+2.5) | lus (+2.4) | ti (+1.4) | ko (+1.3) | ay (+1.3) |
| CodeswitchParallel | mni (+2.2) | min (+1.9) | kg (+1.7) | lus (+1.4) | kbp (+1.4) |
| CodeswitchMonoParallel | bo (+2.2) | lus (+1.9) | ay (+1.4) | bm (+1.1) | ff (+1.0) |
| CodeswitchMonoGatiPanlex | lus (+5.0) | bo (+3.5) | bm (+3.2) | gn (+2.8) | ba (+2.5) |
| GlowupMono | bo (+2.4) | ti (+2.2) | ks (+2.2) | am (+2.0) | mai (+2.0) |
| GlowupParallel | min (+2.1) | ee (+1.9) | kg (+1.6) | kac (+1.5) | kbp (+1.5) |
| GlowupMonoParallel | lus (+2.9) | min (+2.5) | bo (+1.8) | ace (+1.8) | bug (+1.8) |
| GlowupMonoGatiPanlex | lus (+2.2) | gn (+2.2) | ko (+1.9) | sa (+1.8) | ckb (+1.7) |

**Table 13:** Top 5 biggest winners per model (xx→en) on the FLORES-200 test set, measured in $\Delta$CHRF over the baseline.

9. Madurese (mad): up to +6.6 CHRF
10. Assamese (as): up to +6.3 CHRF
11. Pattani Malay (mfa): up to +5.7 CHRF
12. Kalaallisut (kl): up to +5.4 CHRF
13. Tibetan (bo): up to +5.2 CHRF

**xx→en** The biggest winners in the xx→en direction are given in Table 15. Many of the biggest winners overlap with the en→xx direction, but some of the languages that haven't yet been mentioned are:

1. Manipuri (mni{-Mtei}): up to +7.7 CHRF
2. Dogri (doi): up to +5.4 CHRF
3. Dhivehi (dv): up to +3.4 CHRF
4. Tigrinya (ti): up to +3.0 CHRF

### F.3 Big model winners

The biggest winners for the 1.6B parameter models are given in Tables 16, 17, 18, and 19.

### F.4 Empirical study of the quality of Panlex

One way to judge the quality of a dataset is to review it manually, as in (Kreutzer et al., 2022); another is to see the empirical effects on model quality of training on it. As a byproduct of using Panlex for this project, can we judge the quality of Panlex for different languages?

To reduce noise, we average the scores on the three main uses of the bilexes, namely the TokenPairs model, the Glowup model, and the Codeswitch model. We average the FLORES-200 and GATONES scores. We then compare those scores to the baseline model for both en→xx and xx→en. For the purposes of this analysis, we treat any absolute delta of under 0.3 CHRF to be noise. The result is displayed in Tables 20 and 21.

One would like to say that the upper left-hand corner represents languages with unequivocally high-quality lexical data, and the lower right-hand corner represents languages with poor quality lexical data. Alas, however, this picture is rather muddied when we scale up to larger models, as we see that many languages jump from one bucket to another. Nonetheless, we do see the trend that GATITOS languages tend to cluster to the upper left-hand corner in both cases, and that Shan ('shn') and Latin ('la') do poorly in all cases, and should likely be avoided by practitioners.

**Teasing out the confound of the mixed GATI-TOS and Panlex data:** For the 26 GATITOS languages, it is harder to trust the previous analysis. However, we can compare the scores of these languages between the GatiPanlexTokenPairs model and the GatiTokenPairs model. The second of these models is trained on a strict subset of the data that the first is. If a language performs better with this subset of the data, we can presume that the Panlex data was on average lower quality; if a language performs better on the superset, the Panlex data might still be lower quality, but its quantity at least makes up for performance to some degree. The languages that do over +0.3 CHRF better on the subset data are ts, dv, bm, lus, ff, and ckb, suggesting that those may have poorer-quality Panlex data, with the largest difference being lus at +2.7 CHRF; those that do better on the superset are gom, mni-Mtei, kri, ln, doi, ay, sa, ti, mai, and as, suggesting that Panlex still adds useful signal there.

The picture that begins to come together is that Panlex often has some useful signal, but also contains considerable amounts of noise. For a less expressive model that is already not able to reach very high quality, some noise in the lexicons does not hurt, and Panlex can help the model get off the ground for the lowest-resource languages. But for a stronger baseline model that produces higher-quality translations on average, this noise can actively harm performance. Therefore, more carefully curated bilingual lexica, like GATITOS, will tend to will yield higher quality results when used for model training with bigger models, as evinced in Table 21.

| Model | | | | | |
|---|---|---|---|---|---|
| GatiPanlexTokenPairs | gom (+11.6) | ilo (+8.1) | meo (+7.7) | bm (+6.8) | lus (+6.6) |
| CodeswitchMono | ady (+14.1) | av (+9.5) | meo (+6.8) | mad (+6.3) | gom (+5.3) |
| CodeswitchParallel | tiv (+3.6) | min (+3.4) | as (+3.3) | iso (+3.2) | mfa (+2.9) |
| CodeswitchMonoParallel | lus (+7.7) | ady (+6.8) | mad (+6.6) | bm (+6.0) | mfa (+5.7) |
| CodeswitchMonoGatiPanlex | bho (+10.4) | gom (+9.9) | ilo (+9.8) | bm (+9.0) | lus (+8.9) |
| GlowupMono | meo (+10.4) | bho (+6.6) | bo (+5.2) | gom (+5.0) | mfa (+4.1) |
| GlowupParallel | bal (+4.1) | yua (+3.1) | meo (+3.0) | tiv (+2.9) | mni (+2.6) |
| GlowupMonoParallel | meo (+7.1) | gom (+5.9) | mad (+3.9) | za (+3.7) | mni (+3.5) |
| GlowupMonoGatiPanlex | meo (+12.6) | gom (+11.6) | as (+6.3) | ilo (+6.2) | kl (+5.4) |

Table 14: Top 5 biggest winners per model (en→xx) on the GATONES test set, measured in ΔCHRF over the baseline.

| Model | | | | | |
|---|---|---|---|---|---|
| GatiPanlexTokenPairs | bm (+4.7) | mni-Mtei (+4.7) | gn (+3.8) | lus (+3.6) | ilo (+3.5) |
| CodeswitchMono | av (+3.5) | mni-Mtei (+2.9) | yua (+2.5) | dv (+2.3) | lus (+2.2) |
| CodeswitchParallel | mni (+3.0) | cv (+2.5) | av (+2.1) | lus (+1.9) | ee (+1.8) |
| CodeswitchMonoParallel | mni-Mtei (+7.7) | av (+3.8) | lus (+2.4) | bm (+1.5) | chr (+1.5) |
| CodeswitchMonoGatiPanlex | mni-Mtei (+5.4) | lus (+5.3) | gn (+3.5) | bm (+3.5) | dv (+3.4) |
| GlowupMono | ti (+3.0) | bo (+1.8) | or (+1.6) | quc (+1.4) | dv (+1.4) |
| GlowupParallel | ee (+2.2) | mad (+1.6) | yua (+1.6) | av (+1.6) | bm (+1.5) |
| GlowupMonoParallel | mad (+2.8) | min (+2.2) | lus (+2.0) | quc (+1.6) | gom (+1.3) |
| GlowupMonoGatiPanlex | mni-Mtei (+6.0) | doi (+5.4) | gom (+3.5) | dv (+3.3) | bm (+2.7) |

Table 15: Top 5 biggest winners per model (xx→en) on the GATONES test set, measured in ΔCHRF over the baseline.

| Model | | | | | |
|---|---|---|---|---|---|
| GatiPanlexTokenPairsBig | ts (+7.5) | din (+6.0) | ln (+ 5.8) | ilo (+5.3) | ay (+4.1) |
| CodeswitchMonoGatiPanlexBig | ts (+6.9) | bm (+5.6) | ilo (+5.2) | ln (+4.8) | mag (+3.8) |

Table 16: Top 5 biggest winners per model (en→xx) on the FLORES-200 test set for the 1.6B parameter models, measured in ΔCHRF over the baseline.

| Model | | | | | |
|---|---|---|---|---|---|
| GatiPanlexTokenPairsBig | tpi (+9.1) | mni (+6.2) | bm (+5.5) | ts (+3.5) | ay (+3.1) |
| CodeswitchMonoGatiPanlexBig | tpi (+5.1) | ay (+3.0) | mni (+2.2) | bm (+1.9) | ltg (+1.2) |

Table 17: Top 5 biggest winners per model (xx→en) on the FLORES-200 test set for the 1.6B parameter models, measured in ΔCHRF over the baseline.

| Model | | | | | |
|---|---|---|---|---|---|
| GatiPanlexTokenPairsBig | ts (+7.7) | gom (+6.1) | ilo (+6.0) | dv (+5.8) | bm (+5.1) |
| CodeswitchMonoGatiPanlexBig | ts (+7.3) | bm (+7.0) | ilo (+6.8) | mni-Mtei (+5.5) | gom (+5.3) |

Table 18: Top 5 biggest winners per model (en→xx) on the GATONES test set for the 1.6B parameter models, measured in ΔCHRF over the baseline.

| Model | | | | |
|---|---|---|---|---|
| GatiPanlexTokenPairsBig | cv (+8.7) | mni (+8.4) | bm (+6.6) | kl (+5.8) | ee (+5.0) |
| CodeswitchMonoGatiPanlexBig | cv (+6.9) | kl (+6.2) | ce (+3.4) | av (+2.8) | chr (+2.7) |

**Table 19:** Top 5 biggest winners per model (xx→en) on the GATONES test set for the 1.6B parameter models, measured in ΔCHRF over the baseline.

| | Win xx→en | Neut. xx→en | Loss xx→en |
|---|---|---|---|
| Win en→xx | aa ace av **ay** bci **bm** bo cv **doi dv** dyu dz **ee gn gom ilo** kbp **kl kri lg lus** mad min **mni-Mtei nso** nus **om** quc quy sg **ts** yua | cjk **ckb** ny **ti** tn | acm acq aeb af am apc ar ar-MA arz **as** awa az ba bbc be bg **bho** bn bs ca ce ceb cs cy da de el eo et eu fa-AF fi fil fj fo fr ga gl gu hr hu hy id is iso it iw ja jv ka kk km kn ko ks ku lb lo lt ltg lv mag mfa mg mk ml mn mni mr ms mt nl no pa pt rn ro ru rw scn si sk sl sn so sq sr su sv sw ta te tg th tr tt uk ur uz vi war xh yue zh zu |
| Neut. en→xx | **ak** bug pag **qu** tpi | bew kmb or pcm sat-Beng sm | ban brx-Beng es fa gd ha hi ht ig ky **ln mai** mi my ne oc pap pl ps sd skr st tk ug vec wo yi yo zza |
| Loss en→xx | din **ff** fon kg **sa** tum | ady ber kac | ahr ber-Latn hne la shn |

**Table 20:** Languages sorted by whether it helps or hurts to include PanLex and GATITOS, for the smaller models (Transformer Big, 475M). GATITOS languages bolded.

| | Win xx→en | Neut. xx→en | Loss xx→en |
|---|---|---|---|
| Win en→xx | **ay bm** cv **dv ee gom ilo kl kri** ltg **mni-Mtei sa** | awa bci din **doi** fo ga **lus** mag quy tt | aeb ahr **ak** ba **ckb** dyu kg **lg ln** mfa pag pap rn sat-Beng skr su **ts** yua |
| Neut. en→xx | ce km mg tpi zza | af am ar az ban be bg **bho** bn bs ca cs cy da de el eo es et eu fa fa-AF **ff** fi fil fr gd gl gu ha hi hne hr ht hu hy id is it iw ja ka kk kn ko ky lb lo lt lv mad mi mk ml mn mr ms mt ne nl no ny pa pcm pl pt **qu** ro ru si sk sl so sq sr sv sw ta te tg th tk tr uk ur uz vi yi yo zh zu | ace acm acq apc ar-MA arz bew brx-Beng bug ceb dz fj ig jv ks **nso** oc **om** or ps scn sd sm sn st ug vec xh yue |
| Loss en→xx | ady av ber-Latn min mni | aa bbc cjk **gn** kmb my sg tn tum | **as** ber bo fon iso kac kbp ku la **mai** nus quc rw shn **ti** war wo |

**Table 21:** Languages sorted by whether it helps or hurts to include PanLex and GATITOS, for the bigger models (Transformer 1.6B). GATITOS languages bolded.

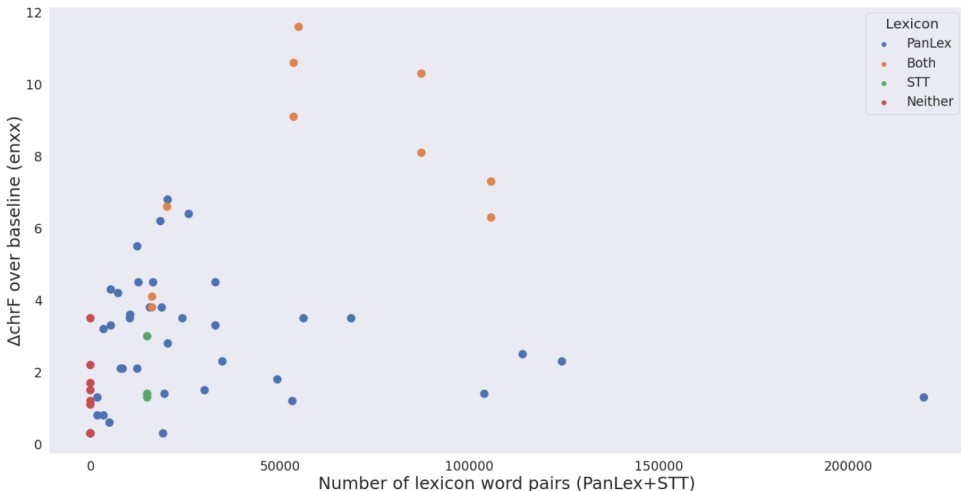

**Figure 4:** Number of lexicon word pairs in augmented data vs. $\Delta$CHRF over baseline for unsupervised languages in the en→xx direction. Results for FLORES-200 and GATONES are combined here.

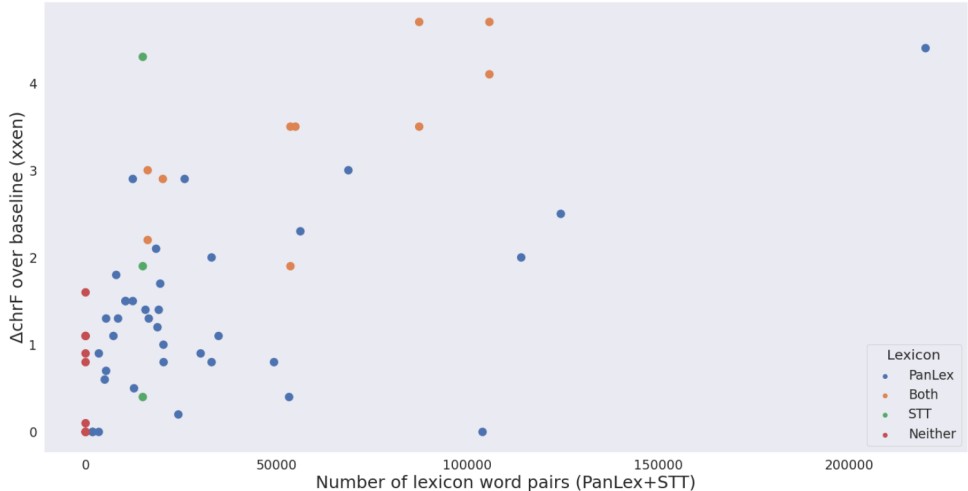

**Figure 5:** Number of lexicon word pairs in augmented data vs. $\Delta$CHRF over baseline for unsupervised languages in the xx→en direction. Results for FLORES-200 and GATONES are combined here.

## G   Does lexical augmentation fix common MT mistakes?

In evaluating the "big" models with 1.6B parameters, we wished to see whether our preferred lexical data augmentation methods (GatiPanlexTokenPairs or CodeswitchMonoGatiPanlex) reduced several common types of MT errors. The errors we looked at were (1) null output, or the "question mark phenomenon," where the model simply outputs some unrelated symbol (such as question marks) instead of actual text; (2) copying, where the model copies some or all of the source sentence in its prediction; and (3) repetition, where the model erroneously repeats the same word or phrase many times. There are other error types we could look at, like hallucination, but we stick with these three basic types for this paper. More precise definitions of these errors are given below.

The results of this analysis are given in Table 22. For each error type, we computed the percentage of sentences that were affected by dividing the number of affected sentences by the total number of sentences in the eval set. FLORES-200 has 806248 sentence pairs across all languages and GATONES has 309887.

**Null output (question mark phenomenon)**   The first error type occurs when the model outputs only "??", or some other arbitary character, as its prediction. Instances of this likely indicate catastrophic effects of out-of-domain phenomena for 0-shot translation.

**Copying**   Another common error is copying, where the model's prediction is close or identi-

| | % question marks | % near copy | % repetition |
|---|---|---|---|
| | FLORES-200 | | |
| BaselineBig | 0.7 | 4.5 | 3.4 |
| GatiPanlexTokenPairsBig | 0.9 | **3.6** | **2.7** |
| CodeswitchMonoGatiPanlexBig | **0.4** | 4.7 | 3.2 |
| | GATONES | | |
| BaselineBig | 0.6 | 2.2 | 3.1 |
| GatiPanlexTokenPairsBig | 0.6 | **1.8** | **2.2** |
| CodeswitchMonoGatiPanlexBig | **0.4** | 2.3 | 2.6 |

**Table 22:** The frequencies of three common error types in MT in each of the eval sets, as a percentage of the total sentences in the set that had each issue (lower is better). Exact descriptions of the error types are given in Section G.

cal to the source sentence. In measuring this phenomenon, we said that any prediction with $> 85\%$ character-level similarity to the source sentence was considered a copy. To measure character-level similarity, we took the multiset intersection of the character frequencies in the source and the character frequencies in the prediction, and then divided the size of the intersection by the number of characters in the source.

**Repetition** The last common error type we examined was repetition. To count these mistakes, we divided the total number of tokens in a sentence by the number of *unique* tokens. If the ratio was $> 3$, we counted the prediction as an instance of erroneous repetition.

## H Comparing sampling strategies for translating tokens

As one recalls from Section 6.1.1, the Codeswitch augmentation works as follows: Let $D$ represent a multilingual lexicon containing word or phrase translation pairs for many languages. Given a source sentence $x = (x_1, x_2, ..., x_n)$ from monolingual corpus $X_{mono}$, we substitute each token in $x$ for its dictionary translation with probability $p_{tr}$.

However, there is an issue with this formulation. Because the lexica we use do not have exhaustive coverage across languages, it is often the case that simply looping over $x$ and attempting to translate each token with probability $p_{tr}$ would result in translating a fraction of $x$ that is significantly less than $p_{tr}$. So in order to approximate this desired fraction $p_{tr}$ as closely as possible, we first count how many tokens in $x$ have dictionary translations. Let this number be $k$. We then compute the adjusted probability $\tilde{p}_{tr} = \max(\frac{np_{tr}}{k}, 1)$, and sample from amongst the words in $x$ with translations

with probability $\tilde{p}_{tr}$, to obtain the codeswitched sentence $x'$. When substituting a source word for its translation, we choose a translation uniformly at random from all available translations in all languages. Because of this, it is usually the case that $x$ is codeswitched into many languages. Finally, we train the model to reconstruct the monolingual sentence $x$ from $x'$ using the same sequence-to-sequence model and loss function as for the MT task.

In our experiments we use use $p_{tr} = 0.4$. We apply MCA to all 208 languages in our corpus, but augment only half the available monolingual data and train the remaining half with MASS (Song et al., 2019b), as done in the baseline training regime (Bapna et al., 2022). We prepend a task token, `<2codeswitch>`, to the codeswitched sentences to cue the model to perform the MCA task, as well as language (`<2lang>`) and script (`<2script>`) tokens. The `<2lang>` and script `<2script>` tags are used in all models, including the baseline.

Since this augmentation samples each token with some probability, the number of tokens translated in a given sentence follows a binomial distribution. The Glowup augmentation, however, samples a number of tokens to translate uniformly at random from all possible translatable tokens. So one has a binomial distribution over N tokens sampled, and the other has a uniform distribution—does this make a difference?

To test this we trained a version of the CodeswitchMono model using uniform sampling. The average CHRF of the CodeswitchMonoUniform model was 0.1 to 0.2 higher on all four of the (en→xx xx→en) x (FLORES-200 GATONES) directions. We conclude that this may have a slight benefit, but the difference is within the realm of

noise, and does not affect the conclusions elsewhere in this paper.

## I  Relationship between number of tokens and MT performance

We also graph the relationship between number of lexical word pairs and $\Delta$CHRF in Figures 4 (en→xx) and 5 (xx→en) for URLs only. The results for FLORES-200 and GATONES are combined in these plots. In both directions, we observe a moderate positive relationship between the number of lexical word pairs for a given language in the augmented data and the $\Delta$CHRF over the baseline.

## J  Languages

### J.1  Rationale for Language Choice

Although this project is aligned with the 1000-language initiative from Bapna et al. (2022), we wanted to use smaller models for more rapid iteration, and as a result, commensurately smaller data and number of languages to fit comfortably in the model. Therefore, we chose to work with about 200 languages.

With this in mind, we also wanted to choose specifically those languages whose performance we could measure. Therefore, our approach was as follows:

- Include all languages with supervised (parallel) data, for maximal cross-lingual transfer
- Include all languages that have non-zero data and a FLORES-200 eval set
- Include all languages that have non-zero data and a GATONES eval set

### J.2  Complete Language data

The following table gives a list of the languages used in our experiments, along with some linguistic and resource-related statistics. The numbers for data resources (i.e. Mono, Parallel, Panlex, and GATITOS) refer to the amount of data actually used in our experiments, *not* necessarily the total amount of data available. For example, we subsampled the parallel and high-resource monolingual data we had available by a factor of 10.

| BCP-47 | Language | Cat. | Mono | Parallel | PanLex | GATITOS | Speak. | Script | Cont. | Family |
|--------|----------|------|------|----------|--------|---------|--------|--------|-------|--------|
| en | English | HRL | 738.8M | 4726.4M | 3.6M | 0 | 984M | Latn | Europe | Indo-European |
| es | Spanish | HRL | 175.1M | 585.6M | 2.4M | 4K | 528M | Latn | Europe | Indo-European |
| de | German | HRL | 169.3M | 389.3M | 2.8M | 4K | 130M | Latn | Europe | Indo-European |
| id | Indonesian | HRL | 95M | 97.4M | 1.1M | 4K | 198M | Latn | Asia | Austronesian |
| yue | Cantonese | MRL | 83.1M | 405K | 180K | 0 | 84M | Hant | Asia | Sino-Tibetan |
| hu | Hungarian | HRL | 72.4M | 50.9M | 1.3M | 0 | 13M | Latn | Europe | Indo-European |
| pl | Polish | HRL | 68.7M | 152.8M | 1.4M | 4K | 41M | Latn | Europe | Indo-European |
| zh | Mandarin | HRL | 67.6M | 215.7M | 6K | 4K | 1092M | Hans | Asia | Sino-Tibetan |
| mi | Maori | MRL | 67.4M | 1.3M | 474K | 0 | 50K | Latn | Oceania | Austronesian |
| ko | Korean | HRL | 67.2M | 128.1M | 1.3M | 5K | 77M | Kore | Asia | Koreanic |
| ja | Japanese | HRL | 65.2M | 307.7M | 2.2M | 4K | 128M | Jpan | Asia | Japonic |
| lo | Lao | MRL | 59.4M | 817K | 184K | 0 | 30M | Laoo | Asia | Kra-Dai |
| ru | Russian | HRL | 57M | 294.7M | 2.8M | 4K | 268M | Cyrl | Europe | Indo-European |
| gd | Scottish Gaelic | MRL | 56.9M | 4M | 324K | 0 | 57K | Latn | Europe | Indo-European |
| tr | Turkish | HRL | 56.5M | 159.2M | 1.3M | 4K | 71M | Latn | Asia | Turkic |
| so | Somali | MRL | 56.4M | 1.3M | 112K | 0 | 16M | Latn | Africa | Afro-Asiatic |
| th | Thai | HRL | 53.1M | 69.1M | 1.6M | 4K | 61M | Thai | Asia | Kra-Dai |
| ha | Hausa | MRL | 50.6M | 1.8M | 202K | 0 | 80M | Latn | Africa | Afro-Asiatic |
| it | Italian | HRL | 49.1M | 245.5M | 2M | 4K | 66M | Latn | Europe | Indo-European |
| pt | Portuguese | HRL | 48.5M | 240.7M | 1.7M | 4K | 230M | Latn | Europe | Indo-European |
| vi | Vietnamese | HRL | 47.7M | 94.2M | 825K | 4K | 68M | Latn | Asia | Austroasiatic |
| fr | French | HRL | 45M | 481.6M | 2.5M | 4K | 230M | Latn | Europe | Indo-European |
| ceb | Cebuano | MRL | 43.9M | 9.2M | 62K | 0 | 20M | Latn | Asia | Austronesian |
| yo | Yoruba | MRL | 43.4M | 847K | 244K | 0 | 50M | Latn | Africa | Niger-Congo |
| sd | Sindhi | MRL | 42.3M | 1.6M | 43K | 0 | 26M | Arab | Asia | Indo-European |
| co | Corsican | MRL | 41.2M | 1.5M | 148K | 0 | 150K | Latn | Europe | Indo-European |
| mg | Malagasy | MRL | 40.2M | 3.6M | 116K | 0 | 25M | Latn | Africa | Austronesian |
| ms | Malay | HRL | 39.4M | 53.8M | 0 | 0 | 77M | Latn | Asia | Austronesian |
| ar-MA | Mor. Arabic | URL | 35.5M | 0 | 0 | 0 | 52M | Arab | Africa | Afro-Asiatic |
| bew | Betawi | URL | 33.3M | 0 | 5K | 0 | 5M | Latn | Asia | Malay Creole |
| ny | Nyanja | MRL | 32.9M | 1.2M | 26K | 0 | 12M | Latn | Africa | Niger-Congo |
| nl | Dutch | HRL | 32M | 258M | 1.6M | 4K | 22M | Latn | Europe | Indo-European |
| uk | Ukrainian | HRL | 32M | 75.3M | 892K | 0 | 35M | Cyrl | Europe | Indo-European |
| sv | Swedish | HRL | 31.1M | 122.6M | 1.3M | 0 | 12M | Latn | Europe | Indo-European |
| haw | Hawaiian | MRL | 30.4M | 698K | 156K | 0 | 24K | Latn | Americas | Austronesian |
| ro | Romanian | HRL | 30.1M | 45M | 841K | 0 | 24M | Latn | Europe | Indo-European |
| cs | Czech | HRL | 30M | 106.9M | 1.6M | 0 | 13M | Latn | Europe | Indo-European |
| hmn | Hmong | MRL | 27.7M | 4.9M | 4K | 0 | 4M | Latn | Asia | Hmong-Mien |
| yi | Yiddish | MRL | 27.6M | 760K | 0 | 0 | 2M | Hebr | Europe | Indo-European |
| fa | Persian | HRL | 27.5M | 45.2M | 0 | 0 | 53M | Arab | Asia | Indo-European |
| ig | Igbo | MRL | 27.4M | 647K | 48K | 0 | 27M | Latn | Africa | Niger-Congo |
| lv | Latvian | MRL | 26M | 22.4M | 0 | 0 | 2M | Latn | Europe | Indo-European |
| ar | Arabic | HRL | 25.8M | 116.7M | 0 | 4K | 310M | Arab | Asia | Afro-Asiatic |
| ckb | Sorani | MRL | 25.1M | 155K | 53K | 4K | 7M | Arab | Asia | Indo-European |
| tt | Tatar | MRL | 25M | 557K | 128K | 0 | 5M | Cyrl | Europe | Turkic |
| sm | Samoan | MRL | 23.9M | 502K | 58K | 0 | 510K | Latn | Oceania | Austronesian |
| zu | Zulu | MRL | 23.4M | 2.3M | 144K | 0 | 12M | Latn | Africa | Niger-Congo |
| no | Norwegian | HRL | 23.1M | 85.8M | 4K | 0 | 5M | Latn | Europe | Indo-European |
| st | Sesotho | MRL | 22.6M | 1.2M | 35K | 0 | 6M | Latn | Africa | Niger-Congo |
| ta | Tamil | MRL | 22M | 11.1M | 247K | 0 | 76M | Taml | Asia | Dravidian |
| or | Odia (Oriya) | MRL | 21.2M | 169K | 16K | 0 | 35M | Orya | Asia | Indo-European |
| sn | Shona | MRL | 18.5M | 958K | 102K | 0 | 8M | Latn | Africa | Niger-Congo |
| bo | Tibetan | LRL | 17.8M | 282K | 56K | 0 | 1M | Tibt | Asia | Sino-Tibetan |
| el | Greek | HRL | 17.3M | 54M | 1.2M | 0 | 13M | Grek | Europe | Indo-European |
| fi | Finnish | HRL | 17.1M | 48.6M | 2.1M | 0 | 6M | Latn | Europe | Uralic |
| hi | Hindi | HRL | 16.5M | 75.7M | 449K | 4K | 381M | Deva | Asia | Indo-European |
| xh | Xhosa | LRL | 15.8M | 697K | 57K | 0 | 8M | Latn | Africa | Niger-Congo |
| mr | Marathi | MRL | 15.6M | 8.1M | 154K | 0 | 75M | Deva | Asia | Indo-European |
| sk | Slovak | HRL | 15.6M | 63.9M | 1.1M | 0 | 7M | Latn | Europe | Indo-European |
| hy | Armenian | MRL | 15.4M | 6.9M | 751K | 0 | 5M | Armn | Asia | Indo-European |
| kk | Kazakh | MRL | 15.3M | 6.6M | 241K | 0 | 13M | Cyrl | Asia | Turkic |
| da | Danish | HRL | 15.2M | 78.9M | 545K | 0 | 6M | Latn | Europe | Indo-European |
| mk | Macedonian | MRL | 15M | 6.6M | 363K | 0 | 2M | Cyrl | Europe | Indo-European |
| bg | Bulgarian | MRL | 14.9M | 37.4M | 693K | 0 | 8M | Cyrl | Europe | Indo-European |
| sr | Serbian | MRL | 14.1M | 30.6M | 206K | 0 | 8M | Cyrl | Europe | Indo-European |
| ml | Malayalam | MRL | 13.9M | 6.4M | 163K | 0 | 34M | Mlym | Asia | Dravidian |
| az | Azerbaijani | MRL | 13.5M | 19.6M | 0 | 0 | 23M | Latn | Asia | Turkic |
| is | Icelandic | MRL | 13.4M | 15.8M | 620K | 0 | 310K | Latn | Europe | Indo-European |
| te | Telugu | MRL | 12.7M | 8M | 311K | 0 | 79M | Telu | Asia | Dravidian |
| ne | Nepali | MRL | 11.6M | 9.7M | 0 | 0 | 16M | Deva | Asia | Indo-European |
| mzn | Mazanderani | URL | 11.6M | 0 | 29K | 0 | 6M | Arab | Asia | Indo-European |
| meo | Kedah Malay | URL | 11.3M | 0 | 0 | 0 | 3M | Latn | Asia | Austronesian |
| et | Estonian | MRL | 11.2M | 30.1M | 0 | 0 | 1M | Latn | Europe | Uralic |
| iw | Hebrew | HRL | 10.8M | 57.1M | 707K | 0 | 5M | Hebr | Asia | Afro-Asiatic |
| rw | Kinyarwanda | LRL | 10.1M | 803K | 56K | 0 | 10M | Latn | Africa | Niger-Congo |
| mn | Mongolian | LRL | 10M | 4.1M | 0 | 0 | 5M | Cyrl | Asia | Mongolic |
| ur | Urdu | MRL | 9.9M | 15.2M | 400K | 0 | 163M | Arab | Asia | Indo-European |
| apc | N. Lev. Arabic | URL | 9.7M | 0 | 0 | 0 | 15M | Arab | Asia | Afro-Asiatic |
| hr | Croatian | MRL | 9.7M | 17.4M | 738K | 0 | 7M | Latn | Europe | Indo-European |
| fil | Filipino | MRL | 9.5M | 25.3M | 61K | 0 | 45M | Latn | Asia | Austronesian |
| as | Assamese | LRL | 9.3M | 575K | 78K | 4K | 15M | Beng | Asia | Indo-European |
| arz | Egyptian Arabic | URL | 9.2M | 0 | 101K | 0 | 58M | Arab | Africa | Afro-Asiatic |
| fo | Faroese | LRL | 9.2M | 26K | 254K | 0 | 66K | Latn | Europe | Indo-European |
| pap | Papiamento | URL | 9.1M | 0 | 99K | 0 | 341K | Latn | Americas | Portuguese Creole |
| fa-AF | Dari | LRL | 8.7M | 10K | 0 | 0 | 21M | Arab | Asia | Indo-European |
| acm | Mesop. Arabic | URL | 8.7M | 0 | 11K | 0 | 15M | Arab | Asia | Afro-Asiatic |

| BCP-47 | Language | Cat. | Mono | Parallel | PanLex | GATITOS | Speak. | Script | Cont. | Family |
|--------|----------|------|------|----------|--------|---------|--------|--------|-------|--------|
| lt | Lithuanian | MRL | 8.6M | 30.9M | 611K | 0 | 3M | Latn | Europe | Indo-European |
| lus | Mizo | URL | 8.3M | 0 | 85K | 4K | 688K | Latn | Asia | Sino-Tibetan |
| be | Belarusian | LRL | 8.3M | 6.5M | 508K | 0 | 3M | Cyrl | Europe | Indo-European |
| kn | Kannada | LRL | 8M | 5.8M | 116K | 0 | 47M | Knda | Asia | Dravidian |
| dv | Dhivehi | LRL | 7.9M | 1K | 34K | 4K | 300K | Thaa | Asia | Indo-European |
| my | Burmese | LRL | 7.8M | 5.5M | 121K | 0 | 43M | Mymr | Asia | Sino-Tibetan |
| oc | Occitan | LRL | 7.5M | 6K | 2.4M | 0 | 500K | Latn | Europe | Indo-European |
| bn | Bengali | MRL | 7.3M | 21.7M | 259K | 0 | 262M | Beng | Asia | Indo-European |
| af | Afrikaans | MRL | 7.1M | 12.7M | 258K | 0 | 18M | Latn | Africa | Indo-European |
| eu | Basque | LRL | 7M | 6.4M | 792K | 0 | 540K | Latn | Europe | Language isolate |
| gu | Gujarati | LRL | 6.9M | 5.8M | 198K | 0 | 47M | Gujr | Asia | Indo-European |
| gl | Galician | MRL | 6.4M | 13.1M | 383K | 0 | 2M | Latn | Europe | Indo-European |
| sa | Sanskrit | LRL | 6.2M | 11K | 168K | 4K | 100K | Deva | Asia | Indo-European |
| sl | Slovenian | MRL | 5.9M | 27M | 672K | 0 | 2M | Latn | Europe | Indo-European |
| ug | Uyghur | LRL | 5.7M | 526K | 116K | 0 | 10M | Arab | Asia | Turkic |
| ba | Bashkir | LRL | 5.6M | 303K | 112K | 0 | 1M | Cyrl | Europe | Turkic |
| si | Sinhala | LRL | 5.6M | 6.5M | 88K | 0 | 16M | Sinh | Asia | Indo-European |
| om | Oromo | LRL | 5.6M | 203K | 10K | 4K | 24M | Latn | Africa | Afro-Asiatic |
| zza | Zaza | URL | 5.3M | 0 | 0 | 0 | 2M | Latn | Asia | Indo-European |
| uz | Uzbek | MRL | 5.3M | 11.2M | 0 | 0 | 34M | Latn | Asia | Turkic |
| sw | Swahili | MRL | 5.2M | 10.5M | 0 | 0 | 150M | Latn | Africa | Niger-Congo |
| km | Khmer | MRL | 5.1M | 8M | 188K | 0 | 17M | Khmr | Asia | Austroasiatic |
| ky | Kyrgyz | LRL | 4.9M | 2.9M | 174K | 0 | 5M | Cyrl | Asia | Turkic |
| am | Amharic | LRL | 4.7M | 2.8M | 71K | 0 | 26M | Ethi | Africa | Afro-Asiatic |
| vec | Venetian | URL | 4.4M | 0 | 202K | 0 | 4M | Latn | Europe | Indo-European |
| ca | Catalan | MRL | 4.4M | 32.9M | 803K | 0 | 9M | Latn | Europe | Indo-European |
| tk | Turkmen | LRL | 4M | 416K | 177K | 0 | 7M | Latn | Asia | Turkic |
| ti | Tigrinya | LRL | 3.9M | 67K | 45K | 4K | 8M | Ethi | Africa | Afro-Asiatic |
| pa | Punjabi | LRL | 3.7M | 3.3M | 89K | 0 | 29M | Guru | Asia | Indo-European |
| sq | Albanian | MRL | 3.2M | 10.6M | 269K | 0 | 13M | Latn | Europe | Indo-European |
| ka | Georgian | MRL | 3M | 11.7M | 482K | 0 | 4M | Geor | Asia | Kartvelian |
| cv | Chuvash | URL | 2.8M | 0 | 121K | 0 | 1M | Cyrl | Europe | Turkic |
| ilo | Ilocano | URL | 2.6M | 0 | 41K | 4K | 9M | Latn | Asia | Austronesian |
| bal | Baluchi | URL | 2.5M | 0 | 16K | 0 | 8M | Arab | Asia | Indo-European |
| eo | Esperanto | LRL | 2.4M | 7.5M | 1.3M | 0 | 2M | Latn | Europe | Constructed |
| cy | Welsh | LRL | 2.2M | 6.4M | 448K | 0 | 590K | Latn | Europe | Indo-European |
| la | Latin | LRL | 2.2M | 2.2M | 740K | 0 | 0 | Latn | Europe | Indo-European |
| dz | Dzongkha | LRL | 2.1M | 260K | 31K | 0 | 200K | Tibt | Asia | Sino-Tibetan |
| mt | Maltese | LRL | 2.1M | 7.3M | 247K | 0 | 470K | Latn | Europe | Afro-Asiatic |
| tn | Tswana | LRL | 2M | 66K | 100K | 0 | 8M | Latn | Africa | Niger-Congo |
| lg | Luganda | LRL | 2M | 3K | 31K | 4K | 4M | Latn | Africa | Niger-Congo |
| ht | Haitian | LRL | 2M | 3.4M | 177K | 0 | 8M | Latn | Americas | French Creole |
| nso | Sepedi | LRL | 1.9M | 798K | 13K | 4K | 5M | Latn | Africa | Niger-Congo |
| ps | Pashto | LRL | 1.8M | 2.1M | 0 | 0 | 50M | Arab | Asia | Indo-European |
| za | Zhuang | URL | 1.7M | 0 | 0 | 0 | 15M | Latn | Asia | Kra-Dai |
| ga | Irish | LRL | 1.7M | 4M | 428K | 0 | 1M | Latn | Europe | Indo-European |
| tpi | Tok Pisin | LRL | 1.7M | 3.3M | 62K | 0 | 120K | Latn | Oceania | English Creole |
| pcm | Nigerian Pidgin | LRL | 1.6M | 24K | 4K | 0 | 40M | Latn | Africa | English Creole |
| lb | Luxembourgish | LRL | 1.5M | 4.6M | 183K | 0 | 420K | Latn | Europe | Indo-European |
| ku | Kurmanji | LRL | 1.5M | 2.1M | 0 | 0 | 15M | Latn | Asia | Indo-European |
| tg | Tajik | LRL | 1.4M | 1.5M | 194K | 0 | 8M | Cyrl | Asia | Indo-European |
| ln | Lingala | LRL | 1.4M | 5K | 97K | 4K | 58M | Latn | Africa | Niger-Congo |
| ce | Chechen | URL | 1.4M | 0 | 113K | 0 | 1M | Cyrl | Europe | NE Caucasian |
| mai | Maithili | URL | 1.3M | 0 | 0 | 4K | 65M | Deva | Asia | Indo-European |
| jv | Javanese | LRL | 1.3M | 6.2M | 128K | 0 | 84M | Latn | Asia | Austronesian |
| ts | Tsonga | LRL | 1.3M | 2K | 9K | 4K | 13M | Latn | Africa | Niger-Congo |
| fj | Fijian | LRL | 1.3M | 6K | 44K | 0 | 339K | Latn | Oceania | Austronesian |
| ak | Twi | LRL | 1.3M | 38K | 115K | 4K | 11M | Latn | Africa | Niger-Congo |
| ber-Latn | Tamazight | URL | 1.2M | 0 | 2K | 0 | 30M | Latn | Africa | Afro-Asiatic |
| su | Sundanese | LRL | 1.2M | 2.7M | 90K | 0 | 34M | Latn | Asia | Austronesian |
| fy | Western Frisian | LRL | 1.1M | 4.8M | 90K | 0 | 850K | Latn | Europe | Indo-European |
| skr | Saraiki | URL | 974K | 0 | 0 | 0 | 20M | Arab | Asia | Indo-European |
| bbc | Batak Toba | URL | 932K | 0 | 23K | 0 | 2M | Latn | Asia | Austronesian |
| war | Waray (PHs) | URL | 902K | 0 | 48K | 0 | 3M | Latn | Asia | Austronesian |
| gn | Guarani | LRL | 861K | 1.3M | 6K | 4K | 5M | Latn | Americas | Tupian |
| qu | Quechua | LRL | 842K | 2K | 46K | 4K | 9M | Latn | Americas | Quechuan |
| bug | Buginese | URL | 797K | 0 | 19K | 0 | 6M | Latn | Asia | Austronesian |
| ee | Ewe | LRL | 796K | 4K | 90K | 4K | 4M | Latn | Africa | Niger-Congo |
| ltg | Latgalian | URL | 796K | 0 | 34K | 0 | 170K | Latn | Europe | Indo-European |
| kl | Kalaallisut | LRL | 741K | 500 | 48K | 4K | 56K | Latn | Americas | Eskimo-Aleut |
| bho | Bhojpuri | LRL | 734K | 4K | 0 | 4K | 60M | Deva | Asia | Indo-European |
| ar-Latn | Arabic | URL | 634K | 0 | 4K | 0 | 3M | Latn | Asia | Afro-Asiatic |
| pag | Pangasinan | URL | 594K | 0 | 19K | 0 | 1M | Latn | Asia | Austronesian |
| shn | Shan | URL | 566K | 0 | 16K | 0 | 3M | Mymr | Asia | Kra-Dai |
| min | Minangkabau | URL | 533K | 0 | 12K | 0 | 6M | Latn | Asia | Austronesian |
| cjk | Chokwe | URL | 494K | 0 | 8K | 0 | 983K | Latn | Africa | Niger-Congo |
| yua | Yucateco | URL | 419K | 0 | 67K | 0 | 766K | Latn | Americas | Mayan |
| sg | Sango | URL | 410K | 0 | 32K | 0 | 400K | Latn | Africa | Ngbandi Creole |
| iso | Isoko | URL | 409K | 0 | 3K | 0 | 420K | Latn | Africa | Niger-Congo |
| kac | Kachin | URL | 402K | 0 | 10K | 0 | 940K | Latn | Asia | Sino-Tibetan |
| kg | Kongo | LRL | 376K | 5K | 15K | 0 | 7M | Latn | Africa | Niger-Congo |
| gom | Goan Konkani | URL | 311K | 0 | 37K | 4K | 2M | Deva | Asia | Indo-European |
| bs | Bosnian | MRL | 311K | 22.6M | 112K | 0 | 2M | Cyrl | Europe | Indo-European |
| av | Avaric | URL | 301K | 0 | 216K | 0 | 760K | Cyrl | Europe | Northeast Caucasian |
| tiv | Tiv | URL | 297K | 0 | 13K | 0 | 2M | Latn | Africa | Niger-Congo |
| ady | Adyghe | URL | 296K | 0 | 25K | 0 | 575K | Cyrl | Europe | Northwest Caucasian |

| BCP-47 | Language | Cat. | Mono | Parallel | PanLex | GATITOS | Speak. | Script | Cont. | Family |
|--------|----------|------|------|----------|--------|---------|--------|--------|-------|--------|
| wo | Wolof | LRL | 289K | 290K | 94K | 0 | 4M | Latn | Africa | Niger-Congo |
| hne | Chhattisgarhi | URL | 269K | 0 | 0 | 0 | 18M | Deva | Asia | Indo-European |
| ay | Aymara | LRL | 267K | 600 | 91K | 4K | 3M | Latn | Americas | Aymaran |
| quc | K'iche' | URL | 250K | 0 | 54K | 0 | 2M | Latn | Americas | Mayan |
| ace | Achinese | URL | 226K | 0 | 18K | 0 | 4M | Latn | Asia | Austronesian |
| acq | Mesop. Arabic | URL | 216K | 0 | 3K | 0 | 7M | Arab | Asia | Afro-Asiatic |
| fon | Fon | URL | 197K | 0 | 8K | 0 | 1M | Latn | Africa | Niger-Congo |
| ban | Balinese | LRL | 188K | 9K | 30K | 0 | 3M | Latn | Asia | Austronesian |
| bm | Bambara | URL | 187K | 0 | 73K | 4K | 14M | Latn | Africa | Mande |
| doi | Dogri | URL | 179K | 0 | 0 | 4K | 2M | Deva | Asia | Indo-European |
| tum | Tumbuka | LRL | 171K | 4K | 1K | 0 | 2M | Latn | Africa | Niger-Congo |
| bci | Baoulé | URL | 152K | 0 | 20K | 0 | 2M | Latn | Africa | Niger-Congo |
| quy | Ayacucho Quechua | URL | 140K | 0 | 94K | 0 | 900K | Latn | Americas | Quechuan |
| mad | Madurese | URL | 138K | 0 | 18K | 0 | 7M | Latn | Asia | Austronesian |
| awa | Awadhi | URL | 136K | 0 | 0 | 0 | 38M | Deva | Asia | Indo-European |
| dyu | Dyula | URL | 130K | 0 | 5K | 4K | 3M | Latn | Africa | Mande |
| kbp | Kabiyè | URL | 129K | 0 | 10K | 0 | 1M | Latn | Africa | Niger-Congo |
| kri | Krio | URL | 129K | 0 | 10K | 4K | 496K | Latn | Africa | English Creole |
| rn | Rundi | LRL | 125K | 2K | 27K | 0 | 9M | Latn | Africa | Niger-Congo |
| mni | Manipuri | URL | 106K | 0 | 2K | 0 | 1M | Beng | Asia | Sino-Tibetan |
| mni-Mtei | Manipuri | URL | 103K | 0 | 1K | 4K | 1M | Mtei | Asia | Sino-Tibetan |
| ber | Tamazight | URL | 96K | 0 | 0 | 0 | 30M | Tfng | Africa | Afro-Asiatic |
| kmb | Kimbundu | URL | 94K | 0 | 7K | 0 | 4M | Latn | Africa | Niger-Congo |
| scn | Sicilian | URL | 92K | 0 | 149K | 0 | 5M | Latn | Europe | Indo-European |
| ff | Fulah | LRL | 86K | 4K | 30K | 4K | 50M | Latn | Africa | Niger-Congo |
| aa | Afar | URL | 82K | 0 | 52K | 0 | 4M | Latn | Africa | Afro-Asiatic |
| ks | Kashmiri | LRL | 71K | 1K | 24K | 0 | 6M | Arab | Asia | Indo-European |
| mag | Magahi | URL | 66K | 0 | 0 | 0 | 14M | Deva | Asia | Indo-European |
| chr | Cherokee | LRL | 63K | 76K | 61K | 0 | 13K | Cher | Americas | Iroquoian |
| din | Dinka | URL | 62K | 0 | 3K | 0 | 1M | Latn | Africa | Nilo-Saharan |
| aeb | Tunisian Arabic | URL | 48K | 0 | 13K | 0 | 11M | Arab | Africa | Afro-Asiatic |
| ahr | Ahirani | URL | 24K | 0 | 0 | 0 | 2M | Deva | Asia | Indo-European |
| nus | Nuer | URL | 24K | 0 | 20K | 0 | 890K | Latn | Africa | Nilo-Saharan |
| mfa | Pattani Malay | URL | 7K | 0 | 0 | 0 | 1000K | Arab | Asia | Austronesian |
| sat-Beng | Santali | URL | 7K | 0 | 0 | 0 | 6M | Beng | Asia | Austroasiatic |
| brx-Beng | Bodo (India) | URL | 4K | 0 | 0 | 0 | 1M | Beng | Asia | Sino-Tibetan |

## K Full results

The full results on FLORES-200 for the various models we trained are available in Tables 24 and 25. Model abbreviations are clarified in Table 23. Scores from the NLLB model are included as reference, though keep in mind that the smaller research models in this paper will naturally have lower quality; even the "Big" models are $30\times$ smaller, and not optimized with back-translation and so on.

| Model Name | Abbr. | Description |
|---|---|---|
| Baseline | B | Trained on MASS + translation |
| GatiPanlexTokenPairs | T | Baseline + 5% token pairs |
| GatiPanlexTokenPairsSamp75 | $T_{75}$ | token-pairs sampled to 75% |
| GatiPanlexTokenPairsSamp50 | $T_{50}$ | token-pairs sampled to 50% |
| GatiPanlexTokenPairsSamp25 | $T_{25}$ | token-pairs sampled to 25% |
| GatiTokenPairs | $T_{GAT}$ | token pairs with only GATITOS |
| CodeswitchMono | $C_M$ | See §6 |
| CodeswitchParallel | $C_P$ | See §6 |
| CodeswitchMonoParallel | $C_{MP}$ | See §6 |
| CodeswitchMonoGatiPanlex | $C_M T$ | See §6 |
| GlowupMono | $G_M$ | See §6 |
| GlowupParallel | $G_P$ | See §6 |
| GlowupMonoParallel | $G_{MP}$ | See §6 |
| GlowupMonoGatiPanlex | $G_M T$ | See §6 |
| BaselineBig | $B_{BIG}$ | Big version of the Baseline (§8.1) |
| GatiPanlexTokenPairsBig | $T_{BIG}$ | Big version of the TP (§8.1) |
| CodeswitchMonoGatiPanlexBig | $(C_M{+}T)_{BIG}$ | Big version of $C_M T$ (§8.1) |
| NLLB | NL | NLLB 54B model |

**Table 23:** Names, abbreviations, and descriptions of the full model results in Tables 25 and 24.

| lang | cat. | B | T | $T_{75}$ | $T_{50}$ | $T_{25}$ | $T_{Gatitos}$ | $C_M$ | $C_P$ | $C_{MP}$ | $C_M$ T | $G_M$ | $G_P$ | $G_{MP}$ | $G_M$ T | $B_{BIG}$ | $T_{BIG}$ | $(C_M$ T$)_{BIG}$ | NLLB |
|---|---|---|---|---|---|---|---|---|---|---|---|---|---|---|---|---|---|---|---|
| $\mu$ | mean | 39.7 | 40.0 | 40.0 | 39.6 | 39.8 | 39.7 | 40.4 | 38.3 | 39.5 | 40.7 | 40.8 | 38.6 | 40.0 | 40.7 | 44.6 | 44.4 | 44.7 | 50.7 |
| HRL | mean | 50.5 | 50.1 | 50.5 | 50.1 | 50.1 | 50.1 | 51.3 | 48.5 | 49.4 | 50.9 | 52.0 | 48.4 | 50.6 | 51.6 | 56.0 | 55.5 | 56.1 | 58.5 |
| MRL | mean | 46.6 | 46.4 | 46.5 | 46.4 | 46.4 | 46.4 | 47.5 | 44.9 | 45.8 | 47.2 | 48.0 | 44.8 | 46.8 | 47.6 | 51.2 | 50.9 | 51.3 | 53.8 |
| LRL | mean | 34.4 | 34.9 | 34.9 | 34.6 | 34.6 | 34.6 | 35.2 | 33.3 | 34.4 | 35.7 | 35.4 | 33.4 | 34.5 | 35.6 | 38.5 | 38.6 | 38.8 | 50.3 |
| URL | mean | 28.7 | 30.2 | 29.5 | 28.8 | 29.5 | 29.1 | 29.2 | 27.8 | 29.9 | 30.2 | 29.2 | 29.2 | 29.7 | 29.5 | 34.8 | 34.5 | 34.5 | 40.1 |
| OOM | mean | 6.2 | 12.6 | 10.9 | 13.2 | 12.3 | 12.5 | 13.7 | 9.7 | 12.2 | 12.2 | 14.2 | 8.8 | 14.8 | 10.6 | 11.0 | 12.9 | 13.0 | - |
| $\Delta_{UR}$ | mean | 30.4 | 31.6 | 31.1 | 30.6 | 31.0 | 30.8 | 31.0 | 29.4 | 31.3 | 31.9 | 31.1 | 30.4 | 31.2 | 31.3 | 35.9 | 35.7 | 35.8 | 43.2 |
| — | — | — | — | — | — | — | — | — | — | — | — | — | — | — | — | — | — | — | — |
| ace | URL | 27.4 | 29.9 | 25.8 | 25.9 | 29.8 | 28.4 | 31.3 | 28.4 | 30.3 | 31.0 | 28.5 | 28.2 | 29.6 | 29.6 | 33.2 | 33.2 | 32.7 | 31.5 |
| acm | URL | 42.0 | 41.5 | 42.5 | 41.3 | 41.3 | 41.7 | 43.4 | 39.8 | 41.3 | 42.9 | 44.4 | 39.3 | 42.5 | 43.4 | 49.0 | 48.2 | 48.8 | 42.1 |
| acq | URL | 39.5 | 39.9 | 40.4 | 39.6 | 40.3 | 40.8 | 44.1 | 38.5 | 41.1 | 43.5 | 45.3 | 38.2 | 42.7 | 42.5 | 49.0 | 46.7 | 49.2 | 26.8 |
| aeb | URL | 36.4 | 36.4 | 35.9 | 35.1 | 36.1 | 37.6 | 39.4 | 35.4 | 37.2 | 39.1 | 40.5 | 34.0 | 38.4 | 38.5 | 42.4 | 41.2 | 43.6 | 61.2 |
| apc | URL | 40.8 | 40.1 | 40.2 | 39.8 | 39.5 | 39.8 | 43.7 | 35.3 | 41.5 | 42.8 | 45.2 | 37.4 | 41.8 | 42.2 | 50.0 | 47.8 | 50.2 | 38.9 |
| ar-MA | URL | 36.0 | 35.7 | 36.2 | 35.6 | 35.7 | 35.5 | 36.8 | 34.5 | 35.2 | 36.3 | 37.3 | 34.0 | 36.0 | 36.5 | 39.9 | 39.8 | 40.2 | 28.3 |
| arz | URL | 36.4 | 33.5 | 32.8 | 33.3 | 35.3 | 32.5 | 38.4 | 27.5 | 35.7 | 37.5 | 38.6 | 30.7 | 33.9 | 37.2 | 43.7 | 42.5 | 43.5 | 50.5 |
| awa | URL | 43.0 | 43.5 | 42.9 | 42.2 | 42.1 | 41.4 | 43.5 | 37.0 | 42.4 | 44.0 | 44.8 | 41.0 | 43.1 | 44.4 | 46.0 | 43.4 | 46.7 | 27.1 |
| ber | URL | 9.2 | 9.9 | 9.0 | 5.7 | 8.8 | 8.1 | 6.7 | 9.7 | 8.9 | 7.9 | 9.6 | 9.8 | 10.2 | 9.7 | 13.6 | 13.5 | 12.1 | 34.3 |
| bm | URL | 15.0 | 20.5 | 21.0 | 20.4 | 18.1 | 20.0 | 14.4 | 16.6 | 20.3 | 22.6 | 13.8 | 16.9 | 14.3 | 18.4 | 23.1 | 26.5 | 28.7 | 44.3 |
| bug | URL | 29.3 | 29.1 | 29.4 | 29.2 | 29.2 | 29.4 | 29.6 | 29.0 | 29.3 | 29.5 | 29.7 | 29.1 | 29.4 | 29.6 | 30.1 | 29.8 | 30.1 | 30.4 |
| cjk | URL | 14.1 | 15.9 | 15.9 | 15.6 | 15.8 | 15.0 | 12.9 | 15.3 | 15.5 | 14.7 | 15.6 | 16.2 | 17.3 | 14.4 | 17.3 | 17.1 | 16.1 | 47.3 |
| din | URL | 12.8 | 9.9 | 11.2 | 9.8 | 9.4 | 9.7 | 11.6 | 8.1 | 13.3 | 11.5 | 8.3 | 11.0 | 11.2 | 8.3 | 12.6 | 18.6 | 15.5 | 19.8 |
| dyu | URL | 12.8 | 15.6 | 14.4 | 14.3 | 15.0 | 13.8 | 11.7 | 14.5 | 15.9 | 16.2 | 12.3 | 14.8 | 14.4 | 15.1 | 17.7 | 19.6 | 20.6 | 50.7 |
| fon | URL | 12.3 | 13.7 | 12.3 | 11.1 | 13.0 | 11.8 | 10.7 | 12.7 | 12.5 | 10.6 | 11.5 | 14.3 | 13.4 | 10.0 | 18.7 | 17.0 | 15.8 | 31.3 |
| hne | URL | 45.8 | 46.1 | 46.1 | 45.6 | 46.7 | 44.0 | 46.2 | 40.2 | 44.6 | 45.4 | 43.1 | 45.0 | 45.3 | 47.6 | 49.1 | 50.0 | 49.3 | 38.5 |
| ilo | URL | 35.5 | 43.0 | 41.5 | 42.4 | 40.9 | 42.6 | 39.5 | 39.2 | 41.3 | 43.5 | 36.3 | 37.7 | 37.9 | 41.9 | 42.7 | 48.0 | 47.9 | 33.0 |
| kac | URL | 16.5 | 17.2 | 16.3 | 13.5 | 17.6 | 15.7 | 12.9 | 15.1 | 16.1 | 16.4 | 15.3 | 20.1 | 18.0 | 12.0 | 26.5 | 23.8 | 22.5 | 46.8 |
| kbp | URL | 12.6 | 13.4 | 13.6 | 10.2 | 11.4 | 14.1 | 13.0 | 12.4 | 14.2 | 12.4 | 13.0 | 16.2 | 14.6 | 13.7 | 21.2 | 18.3 | 16.2 | 48.4 |
| kmb | URL | 19.2 | 23.4 | 19.4 | 19.2 | 20.6 | 17.6 | 15.7 | 19.8 | 22.2 | 19.0 | 19.1 | 22.4 | 21.8 | 17.5 | 25.0 | 24.7 | 20.5 | 41.4 |
| ltg | URL | 29.1 | 29.2 | 29.1 | 29.8 | 30.9 | 30.4 | 31.7 | 30.1 | 30.6 | 31.7 | 30.5 | 31.0 | 30.6 | 31.4 | 38.1 | 38.2 | 38.7 | 34.5 |
| lus | URL | 18.0 | 24.2 | 24.2 | 18.1 | 21.8 | 26.6 | 21.6 | 20.3 | 26.2 | 25.7 | 19.5 | 19.0 | 20.0 | 22.4 | 29.3 | 30.5 | 32.8 | 57.5 |
| mag | URL | 47.3 | 48.0 | 46.4 | 46.0 | 47.1 | 45.0 | 48.2 | 38.5 | 45.1 | 48.4 | 47.9 | 45.8 | 47.5 | 49.1 | 47.5 | 50.2 | 51.3 | 23.5 |
| mai | URL | 43.2 | 44.4 | 44.0 | 43.9 | 44.1 | 42.0 | 42.9 | 39.6 | 41.5 | 43.8 | 41.3 | 41.4 | 41.9 | 44.4 | 48.8 | 47.7 | 47.9 | 48.2 |
| min | URL | 38.2 | 39.4 | 39.5 | 39.8 | 39.7 | 39.4 | 40.0 | 40.0 | 40.0 | 40.1 | 39.8 | 39.7 | 39.5 | 39.4 | 45.7 | 45.5 | 43.6 | 39.1 |
| mni | URL | 13.2 | 12.8 | 15.7 | 14.2 | 14.6 | 14.1 | 12.8 | 15.1 | 14.5 | 12.2 | 16.2 | 17.2 | 14.6 | 12.9 | 22.9 | 20.3 | 15.3 | 38.0 |
| nus | URL | 9.6 | 16.4 | 12.0 | 12.0 | 11.1 | 10.9 | 10.8 | 10.1 | 13.3 | 11.3 | 11.4 | 12.1 | 11.6 | 9.4 | 17.1 | 17.6 | 16.3 | 28.3 |
| pag | URL | 35.9 | 36.9 | 36.3 | 35.7 | 36.1 | 36.3 | 35.1 | 35.8 | 37.3 | 36.4 | 34.9 | 36.8 | 37.0 | 35.6 | 39.9 | 40.2 | 40.9 | 55.6 |
| pap | URL | 40.1 | 41.0 | 39.6 | 40.9 | 41.1 | 39.1 | 40.5 | 38.9 | 40.6 | 42.1 | 38.5 | 39.4 | 39.5 | 39.8 | 44.4 | 44.2 | 45.3 | 42.0 |
| quy | URL | 26.6 | 30.5 | 27.5 | 29.6 | 28.5 | 31.1 | 28.8 | 28.3 | 30.7 | 31.2 | 27.6 | 28.1 | 28.6 | 26.3 | 31.5 | 31.6 | 34.5 | 49.6 |
| scn | URL | 38.2 | 39.4 | 39.3 | 39.0 | 38.1 | 38.9 | 38.1 | 37.8 | 39.7 | 39.5 | 38.4 | 38.3 | 38.6 | 39.1 | 43.7 | 44.7 | 43.3 | 45.2 |
| sg | URL | 21.9 | 25.1 | 23.1 | 21.9 | 22.4 | 21.4 | 19.0 | 22.3 | 25.9 | 21.5 | 23.4 | 25.6 | 25.9 | 19.8 | 31.2 | 29.8 | 28.7 | 56.8 |
| shn | URL | 14.2 | 15.6 | 14.7 | 13.5 | 15.6 | 10.6 | 11.9 | 12.2 | 8.5 | 12.6 | 8.1 | 18.1 | 12.2 | 13.9 | 25.1 | 16.2 | 21.3 | 38.3 |
| vec | URL | 42.3 | 42.2 | 42.2 | 42.1 | 42.0 | 42.1 | 42.8 | 42.0 | 42.3 | 43.0 | 42.5 | 41.5 | 42.6 | 42.8 | 44.4 | 44.1 | 44.3 | 41.9 |
| war | URL | 51.5 | 53.2 | 53.0 | 52.9 | 53.1 | 52.3 | 51.7 | 53.2 | 51.2 | 51.2 | 52.3 | 52.0 | 52.3 | 51.9 | 57.1 | 56.1 | 53.1 | 31.8 |
| am | LRL | 30.6 | 30.5 | 31.6 | 30.3 | 30.9 | 30.6 | 32.5 | 28.3 | 29.4 | 32.0 | 33.2 | 28.2 | 31.3 | 32.6 | 38.4 | 37.7 | 38.0 | 59.3 |
| as | LRL | 13.1 | 15.3 | 17.0 | 12.9 | 16.0 | 16.0 | 11.1 | 15.5 | 14.5 | 18.6 | 15.9 | 12.8 | 15.1 | 15.9 | 20.7 | 18.2 | 17.2 | 48.9 |
| ay | LRL | 22.5 | 25.9 | 23.1 | 24.0 | 22.9 | 23.3 | 22.5 | 23.2 | 26.0 | 24.2 | 24.4 | 22.7 | 24.6 | 22.9 | 27.0 | 23.5 | 50.7 |  |
| ba | LRL | 14.6 | 16.8 | 13.0 | 14.3 | 15.5 | 15.9 | 13.6 | 15.2 | 13.2 | 17.3 | 16.3 | 17.2 | 12.9 | 15.8 | 16.9 | 15.6 | 18.3 | 50.8 |
| ban | LRL | 41.0 | 40.6 | 40.6 | 40.3 | 40.8 | 40.7 | 41.6 | 39.8 | 40.2 | 41.3 | 41.0 | 39.0 | 40.6 | 41.3 | 42.4 | 43.1 | 42.2 | 53.2 |
| be | LRL | 37.7 | 37.3 | 38.0 | 37.5 | 37.5 | 37.2 | 38.4 | 36.2 | 37.1 | 38.0 | 38.9 | 36.1 | 37.8 | 38.4 | 42.0 | 41.8 | 42.3 | 44.7 |
| bho | LRL | 14.7 | 17.9 | 20.0 | 21.9 | 24.7 | 19.2 | 23.2 | 18.2 | 23.1 | 25.5 | 25.4 | 19.1 | 16.0 | 21.6 | 21.0 | 15.4 | 22.1 | 51.0 |
| bo | LRL | 13.2 | 16.3 | 16.1 | 15.7 | 14.7 | 15.1 | 18.2 | 11.1 | 13.3 | 18.0 | 17.9 | 12.7 | 13.5 | 21.3 | 24.8 | 23.3 | 24.4 | 55.8 |
| cy | LRL | 64.4 | 63.9 | 64.7 | 64.1 | 63.8 | 64.0 | 65.8 | 61.4 | 63.1 | 65.5 | 66.8 | 61.2 | 64.3 | 66.1 | 72.2 | 71.9 | 72.6 | 45.4 |
| dz | LRL | 5.6 | 5.1 | 4.9 | 5.0 | 4.5 | 5.5 | 6.6 | 6.0 | 4.9 | 4.9 | 4.9 | 6.7 | 5.1 | 4.8 | 5.7 | 5.2 | 6.4 | 53.3 |
| ee | LRL | 26.9 | 29.5 | 28.9 | 28.3 | 27.2 | 29.3 | 28.4 | 25.2 | 28.5 | 30.5 | 26.8 | 26.3 | 26.8 | 28.6 | 28.6 | 32.0 | 31.3 | 40.4 |
| eo | LRL | 57.6 | 57.5 | 57.7 | 57.4 | 57.3 | 57.4 | 58.1 | 56.7 | 57.0 | 58.0 | 58.2 | 56.5 | 57.6 | 58.0 | 59.0 | 58.8 | 59.1 | 27.9 |
| eu | LRL | 47.6 | 47.1 | 47.8 | 47.0 | 46.8 | 47.0 | 48.4 | 45.4 | 46.8 | 48.1 | 48.8 | 45.4 | 47.7 | 48.4 | 52.6 | 52.4 | 52.4 | 54.8 |
| fa-AF | LRL | 47.4 | 47.1 | 47.5 | 46.7 | 47.0 | 46.6 | 48.0 | 45.9 | 46.1 | 47.0 | 49.4 | 45.0 | 48.3 | 48.6 | 53.3 | 53.1 | 52.9 | 54.0 |
| ff | LRL | 19.1 | 18.9 | 19.0 | 19.4 | 18.3 | 19.3 | 18.8 | 18.8 | 18.8 | 18.8 | 19.0 | 18.6 | 19.6 | 19.2 | 19.8 | 20.1 | 20.2 | 46.2 |
| fj | LRL | 33.7 | 34.8 | 35.6 | 33.7 | 35.6 | 34.0 | 34.0 | 33.7 | 35.5 | 34.8 | 34.0 | 34.2 | 35.7 | 35.2 | 39.7 | 39.6 | 39.7 | 42.3 |
| fo | LRL | 38.5 | 38.2 | 38.4 | 38.0 | 38.3 | 38.2 | 40.1 | 37.7 | 39.0 | 39.6 | 38.7 | 37.5 | 37.0 | 38.5 | 44.4 | 44.7 | 45.1 | 43.3 |
| ga | LRL | 47.9 | 47.6 | 48.6 | 47.9 | 47.6 | 47.9 | 49.5 | 46.1 | 47.3 | 49.1 | 50.2 | 45.7 | 48.0 | 49.7 | 55.0 | 54.8 | 55.7 | 40.8 |
| gn | LRL | 19.1 | 19.6 | 19.2 | 18.5 | 19.4 | 18.7 | 20.3 | 18.4 | 19.2 | 19.4 | 19.0 | 19.3 | 19.3 | 18.0 | 20.8 | 20.7 | 20.1 | 31.6 |
| gu | LRL | 49.8 | 49.5 | 49.9 | 49.1 | 49.4 | 49.4 | 50.7 | 47.7 | 48.6 | 50.3 | 52.1 | 48.0 | 49.9 | 51.0 | 54.1 | 53.6 | 54.5 | 51.1 |
| ht | LRL | 49.6 | 49.1 | 49.6 | 49.3 | 49.4 | 49.4 | 49.9 | 48.6 | 48.9 | 49.7 | 50.2 | 48.4 | 49.6 | 49.9 | 51.2 | 51.1 | 51.0 | 48.3 |
| jv | LRL | 51.7 | 51.8 | 52.7 | 52.1 | 51.5 | 51.7 | 52.5 | 50.7 | 51.1 | 52.3 | 53.1 | 50.7 | 51.6 | 52.8 | 54.5 | 54.3 | 54.4 | 50.6 |
| kg | LRL | 33.1 | 31.9 | 33.0 | 32.9 | 31.0 | 32.9 | 32.3 | 31.9 | 32.9 | 32.8 | 32.9 | 32.9 | 33.1 | 32.1 | 32.8 | 35.2 | 35.1 | 61.1 |
| kn | LRL | 50.5 | 50.4 | 51.0 | 50.0 | 49.9 | 49.9 | 51.4 | 48.4 | 49.2 | 51.2 | 52.2 | 48.4 | 50.6 | 51.7 | 55.5 | 55.0 | 55.7 | 42.8 |
| ks | LRL | 0.8 | 0.8 | 0.8 | 0.8 | 0.8 | 0.8 | 0.8 | 0.7 | 0.8 | 0.9 | 0.8 | 0.8 | 0.8 | 0.8 | 0.8 | 0.8 | 1.0 | 40.9 |
| ku | LRL | 33.2 | 33.1 | 32.8 | 32.5 | 32.9 | 32.4 | 33.8 | 31.5 | 31.7 | 33.4 | 34.1 | 30.9 | 33.4 | 33.9 | 37.9 | 37.1 | 37.0 | 35.4 |
| ky | LRL | 41.8 | 41.1 | 41.5 | 40.9 | 41.2 | 41.2 | 42.4 | 39.6 | 40.5 | 41.8 | 42.8 | 39.7 | 41.7 | 42.7 | 45.7 | 45.4 | 45.7 | 53.9 |
| lb | LRL | 50.0 | 49.8 | 49.9 | 49.8 | 49.9 | 49.6 | 50.7 | 48.4 | 49.4 | 50.5 | 51.4 | 48.3 | 50.1 | 50.4 | 53.9 | 53.5 | 54.3 | 55.4 |
| lg | LRL | 30.1 | 31.9 | 31.4 | 31.5 | 30.5 | 31.7 | 32.0 | 29.4 | 32.3 | 32.7 | 26.6 | 29.0 | 29.9 | 31.3 | 30.9 | 34.4 | 32.0 | 61.5 |
| ln | LRL | 31.9 | 34.8 | 35.2 | 34.9 | 32.3 | 33.1 | 28.9 | 30.9 | 36.0 | 38.6 | 32.3 | 29.0 | 30.7 | 33.0 | 33.8 | 39.6 | 38.5 | 51.8 |
| mn | LRL | 42.4 | 41.9 | 41.5 | 41.3 | 41.9 | 41.8 | 43.0 | 39.9 | 40.7 | 42.7 | 43.7 | 39.9 | 42.4 | 43.2 | 47.8 | 47.5 | 48.0 | 50.9 |
| mt | LRL | 62.6 | 62.7 | 62.4 | 62.5 | 62.0 | 62.7 | 63.5 | 60.7 | 61.6 | 63.1 | 64.0 | 60.3 | 62.4 | 63.8 | 68.2 | 68.0 | 68.2 | 60.4 |
| my | LRL | 35.3 | 34.0 | 34.9 | 34.4 | 34.6 | 34.0 | 36.0 | 32.7 | 34.0 | 37.0 | 36.8 | 32.2 | 35.1 | 36.5 | 40.8 | 39.2 | 40.0 | 54.9 |
| nso | LRL | 31.9 | 32.0 | 32.1 | 31.5 | 31.7 | 31.5 | 31.7 | 31.2 | 32.0 | 31.9 | 31.7 | 31.4 | 31.8 | 31.9 | 33.4 | 33.6 | 33.3 | 49.1 |
| oc | LRL | 47.6 | 47.6 | 47.6 | 49.5 | 46.5 | 49.2 | 47.0 | 48.7 | 51.6 | 51.1 | 47.9 | 49.3 | 48.6 | 49.3 | 52.6 | 52.6 | 52.6 | 51.0 |
| om | LRL | 15.5 | 15.5 | 15.5 | 15.7 | 15.5 | 15.5 | 16.0 | 15.3 | 16.0 | 15.9 | 16.6 | 15.3 | 15.2 | 15.6 | 17.1 | 16.8 | 16.8 | 42.2 |
| pa | LRL | 44.7 | 44.6 | 44.3 | 44.0 | 44.3 | 44.3 | 45.3 | 42.5 | 43.4 | 45.0 | 45.8 | 42.7 | 44.6 | 45.5 | 49.7 | 49.2 | 49.2 | 40.3 |
| ps | LRL | 33.7 | 33.3 | 33.2 | 32.8 | 33.5 | 32.9 | 33.9 | 32.0 | 32.9 | 33.9 | 34.7 | 32.5 | 33.8 | 34.2 | 36.5 | 36.3 | 36.3 | 59.1 |
| rn | LRL | 31.1 | 31.8 | 31.5 | 31.6 | 31.4 | 32.0 | 32.5 | 30.4 | 31.9 | 32.5 | 32.2 | 30.9 | 32.2 | 32.9 | 35.8 | 36.6 | 37.0 | 53.8 |
| rw | LRL | 34.4 | 35.5 | 34.7 | 34.8 | 33.9 | 34.5 | 35.7 | 33.2 | 34.0 | 34.9 | 35.1 | 33.1 | 34.9 | 35.4 | 44.1 | 42.5 | 43.5 | 57.2 |
| sa | LRL | 23.0 | 23.7 | 22.4 | 23.0 | 23.9 | 22.3 | 22.7 | 21.1 | 22.5 | 22.3 | 22.1 | 21.9 | 23.3 | 24.9 | 25.2 | 26.0 | 72.6 |  |
| si | LRL | 42.1 | 42.2 | 42.1 | 41.9 | 42.1 | 42.0 | 43.8 | 39.5 | 40.7 | 43.5 | 44.7 | 39.2 | 42.9 | 44.4 | 49.8 | 49.4 | 50.2 | 70.8 |
| su | LRL | 46.5 | 46.6 | 47.1 | 46.8 | 46.6 | 46.8 | 47.2 | 45.9 | 46.8 | 47.4 | 47.5 | 45.8 | 46.8 | 47.3 | 46.2 | 46.2 | 47.4 | 54.7 |

| lang | cat. | B | T | $T_{75}$ | $T_{50}$ | $T_{25}$ | $T_{Gatitos}$ | $C_M$ | $C_P$ | $C_{MP}$ | $C_M$ T | $G_M$ | $G_P$ | $G_{MP}$ | $G_M$ T | $B_{BIG}$ | $T_{BIG}$ | $(C_M$ T$)_{BIG}$ | NLLB |
|---|---|---|---|---|---|---|---|---|---|---|---|---|---|---|---|---|---|---|---|
| tg | LRL | 43.3 | 42.7 | 42.5 | 42.5 | 43.1 | 42.1 | 44.4 | 40.6 | 41.4 | 44.0 | 45.0 | 41.0 | 43.4 | 44.7 | 48.3 | 47.9 | 48.3 | 64.0 |
| ti | LRL | 5.9 | 8.0 | 8.1 | 6.6 | 5.2 | 6.8 | 7.7 | 5.5 | 7.0 | 8.1 | 6.8 | 4.0 | 4.5 | 6.3 | 9.6 | 9.0 | 7.6 | 47.7 |
| tk | LRL | 42.0 | 42.7 | 42.1 | 40.3 | 41.6 | 41.9 | 43.7 | 39.2 | 42.4 | 42.9 | 43.9 | 40.1 | 41.7 | 43.7 | 50.8 | 50.5 | 51.1 | 38.6 |
| tn | LRL | 35.6 | 35.5 | 35.1 | 36.5 | 35.3 | 36.1 | 36.1 | 34.7 | 34.8 | 36.2 | 36.7 | 35.0 | 36.2 | 36.7 | 41.1 | 40.7 | 40.5 | 48.0 |
| tpi | LRL | 25.2 | 25.2 | 25.2 | 25.2 | 25.2 | 25.0 | 25.2 | 25.2 | 25.2 | 25.1 | 25.0 | 24.6 | 25.0 | 25.0 | 25.3 | 25.3 | 25.2 | 56.6 |
| ts | LRL | 33.2 | 37.6 | 35.9 | 36.1 | 35.1 | 37.4 | 35.3 | 31.9 | 34.8 | 37.3 | 31.9 | 32.2 | 32.9 | 34.6 | 33.9 | 41.3 | 40.7 | 40.1 |
| tum | LRL | 32.1 | 32.0 | 31.6 | 32.0 | 32.3 | 32.1 | 31.4 | 31.0 | 31.0 | 30.1 | 31.3 | 30.8 | 30.8 | 31.9 | 32.4 | 32.5 | 31.3 | 45.7 |
| ug | LRL | 36.2 | 37.0 | 36.9 | 35.9 | 36.9 | 35.5 | 38.6 | 32.2 | 35.3 | 39.2 | 38.1 | 32.8 | 37.3 | 38.6 | 47.9 | 47.7 | 47.9 | 58.3 |
| wo | LRL | 13.9 | 15.2 | 14.5 | 14.8 | 14.4 | 13.0 | 11.6 | 12.1 | 14.8 | 14.2 | 14.0 | 14.7 | 14.3 | 14.8 | 21.4 | 20.4 | 20.9 | 54.9 |
| xh | LRL | 47.6 | 47.1 | 47.2 | 47.3 | 47.6 | 47.3 | 48.1 | 46.3 | 46.4 | 47.9 | 48.8 | 46.0 | 47.8 | 48.1 | 52.2 | 51.6 | 52.1 | 48.8 |
| af | MRL | 64.5 | 64.1 | 64.5 | 64.2 | 64.2 | 64.2 | 64.9 | 63.0 | 63.5 | 64.3 | 65.5 | 62.9 | 64.5 | 65.1 | 67.9 | 67.5 | 67.8 | 45.0 |
| az | MRL | 42.1 | 42.0 | 42.1 | 41.7 | 41.9 | 41.8 | 42.6 | 41.1 | 41.3 | 42.2 | 43.2 | 40.8 | 42.1 | 42.6 | 45.2 | 45.2 | 45.1 | 50.4 |
| bg | MRL | 59.3 | 58.8 | 59.3 | 58.8 | 58.7 | 58.6 | 60.0 | 57.1 | 58.4 | 59.6 | 61.2 | 57.0 | 59.5 | 60.6 | 65.6 | 65.2 | 65.8 | 60.6 |
| bn | MRL | 47.1 | 46.7 | 47.2 | 46.5 | 47.0 | 46.5 | 48.0 | 45.1 | 46.1 | 47.8 | 48.8 | 44.7 | 47.1 | 48.2 | 51.9 | 51.3 | 51.8 | 50.9 |
| bs | MRL | 52.9 | 52.7 | 53.3 | 52.8 | 52.6 | 52.7 | 53.9 | 51.3 | 52.4 | 53.4 | 54.8 | 51.4 | 53.1 | 54.4 | 58.4 | 57.8 | 58.5 | 49.2 |
| ca | MRL | 60.7 | 60.4 | 60.8 | 60.4 | 60.4 | 60.5 | 61.4 | 59.4 | 60.3 | 61.2 | 62.0 | 59.1 | 60.8 | 61.6 | 64.8 | 64.7 | 65.3 | 52.5 |
| ceb | MRL | 58.6 | 58.6 | 58.7 | 58.6 | 58.4 | 58.1 | 59.5 | 57.6 | 57.9 | 58.7 | 59.4 | 57.5 | 58.9 | 59.0 | 61.2 | 60.5 | 60.9 | 50.6 |
| ckb | MRL | 31.9 | 35.0 | 34.9 | 33.5 | 31.8 | 35.2 | 35.9 | 28.9 | 33.8 | 37.1 | 31.9 | 30.3 | 31.2 | 36.5 | 43.2 | 44.7 | 44.6 | 61.2 |
| et | MRL | 52.3 | 51.7 | 52.4 | 51.8 | 51.9 | 51.7 | 52.9 | 50.0 | 51.0 | 52.5 | 53.9 | 49.8 | 52.2 | 53.2 | 58.2 | 57.8 | 58.3 | 52.6 |
| fil | MRL | 58.1 | 58.1 | 58.6 | 58.0 | 58.0 | 58.0 | 58.8 | 57.2 | 57.7 | 58.4 | 59.1 | 57.1 | 58.3 | 58.7 | 60.8 | 60.7 | 61.0 | 55.7 |
| gd | MRL | 48.9 | 48.4 | 48.8 | 48.6 | 48.6 | 47.9 | 49.1 | 47.4 | 48.1 | 48.9 | 49.8 | 47.0 | 48.6 | 49.7 | 53.0 | 52.6 | 52.9 | 57.3 |
| gl | MRL | 56.8 | 56.6 | 57.0 | 56.9 | 56.6 | 56.5 | 57.3 | 56.0 | 56.3 | 57.0 | 57.7 | 55.8 | 57.0 | 57.5 | 60.0 | 59.7 | 60.1 | 63.2 |
| ha | MRL | 43.6 | 42.7 | 43.5 | 42.8 | 43.2 | 42.4 | 44.3 | 41.7 | 41.4 | 44.0 | 44.3 | 41.7 | 44.4 | 43.5 | 49.1 | 48.3 | 49.0 | 56.0 |
| hr | MRL | 51.9 | 51.7 | 52.1 | 51.5 | 51.6 | 51.5 | 52.8 | 50.3 | 51.3 | 52.6 | 53.5 | 50.3 | 52.1 | 53.1 | 57.0 | 56.6 | 57.0 | 58.8 |
| hy | MRL | 47.7 | 47.9 | 47.9 | 47.3 | 47.4 | 47.7 | 49.3 | 45.2 | 47.2 | 48.5 | 49.9 | 45.0 | 48.0 | 48.9 | 53.3 | 53.3 | 53.6 | 43.6 |
| ig | MRL | 38.2 | 37.7 | 38.2 | 37.6 | 37.9 | 37.5 | 38.5 | 36.8 | 37.1 | 38.8 | 39.2 | 36.7 | 38.0 | 38.8 | 41.3 | 41.0 | 41.4 | 60.1 |
| is | MRL | 43.8 | 43.6 | 44.2 | 43.4 | 43.6 | 43.7 | 44.7 | 41.8 | 43.0 | 44.5 | 45.7 | 41.7 | 43.8 | 45.3 | 51.0 | 50.4 | 51.1 | 41.8 |
| ka | MRL | 45.2 | 45.3 | 45.4 | 45.0 | 44.9 | 45.0 | 46.3 | 43.5 | 44.3 | 45.6 | 46.5 | 43.0 | 45.2 | 46.0 | 49.5 | 49.3 | 49.6 | 57.8 |
| kk | MRL | 49.6 | 49.5 | 49.7 | 49.5 | 49.4 | 49.4 | 51.0 | 46.9 | 49.1 | 50.3 | 51.6 | 47.0 | 49.9 | 51.1 | 55.6 | 55.2 | 55.5 | 49.7 |
| km | MRL | 38.3 | 38.0 | 38.7 | 38.2 | 38.2 | 37.6 | 39.5 | 36.6 | 37.4 | 39.0 | 40.2 | 36.8 | 38.8 | 39.5 | 43.0 | 43.0 | 43.0 | 66.6 |
| lo | MRL | 44.0 | 44.1 | 44.3 | 44.0 | 44.0 | 43.7 | 44.8 | 42.4 | 43.1 | 44.6 | 45.2 | 41.8 | 44.0 | 44.6 | 47.3 | 47.8 | 47.7 | 61.9 |
| lt | MRL | 49.4 | 48.9 | 49.2 | 49.1 | 48.8 | 49.0 | 50.5 | 47.2 | 48.5 | 49.8 | 51.0 | 47.0 | 49.5 | 50.6 | 56.2 | 55.6 | 56.4 | 52.1 |
| lv | MRL | 51.8 | 51.4 | 51.3 | 51.6 | 51.5 | 51.6 | 52.9 | 49.6 | 50.4 | 52.3 | 53.7 | 49.7 | 52.2 | 53.0 | 58.7 | 57.9 | 58.7 | 48.7 |
| mg | MRL | 44.1 | 44.3 | 44.3 | 44.3 | 44.8 | 43.9 | 45.2 | 42.7 | 43.3 | 45.2 | 45.9 | 42.5 | 44.9 | 45.5 | 49.3 | 48.8 | 49.0 | 43.5 |
| mi | MRL | 40.6 | 40.5 | 40.3 | 40.5 | 40.7 | 40.2 | 41.0 | 40.2 | 40.6 | 40.9 | 41.1 | 39.8 | 40.9 | 40.6 | 41.8 | 41.7 | 41.8 | 60.1 |
| mk | MRL | 57.4 | 57.2 | 57.1 | 57.1 | 57.1 | 57.2 | 58.4 | 55.5 | 56.7 | 57.9 | 58.8 | 55.4 | 57.5 | 58.5 | 61.8 | 61.5 | 61.7 | 51.1 |
| ml | MRL | 48.3 | 47.6 | 47.8 | 47.7 | 47.9 | 47.6 | 49.5 | 45.3 | 46.8 | 49.0 | 50.3 | 44.9 | 48.5 | 50.0 | 55.2 | 53.9 | 55.4 | 59.4 |
| mr | MRL | 45.3 | 44.8 | 44.7 | 44.8 | 44.9 | 44.9 | 46.1 | 43.3 | 44.2 | 46.2 | 46.5 | 43.2 | 45.6 | 46.2 | 50.2 | 49.8 | 50.5 | 52.8 |
| ne | MRL | 49.5 | 49.0 | 48.4 | 48.6 | 48.4 | 48.7 | 49.7 | 47.0 | 48.1 | 49.4 | 50.7 | 47.5 | 49.4 | 50.0 | 53.5 | 53.0 | 53.1 | 27.4 |
| ny | MRL | 44.6 | 44.1 | 44.7 | 44.5 | 44.3 | 44.5 | 45.3 | 43.1 | 43.8 | 44.8 | 45.8 | 43.2 | 44.6 | 45.3 | 48.0 | 47.7 | 48.1 | 50.1 |
| or | MRL | 40.6 | 39.5 | 39.2 | 39.2 | 39.3 | 39.5 | 41.0 | 37.4 | 38.5 | 40.6 | 41.6 | 37.4 | 40.1 | 41.4 | 47.3 | 46.1 | 47.1 | 61.4 |
| sd | MRL | 44.0 | 43.5 | 43.3 | 43.2 | 43.7 | 43.1 | 43.9 | 42.8 | 42.6 | 43.6 | 44.4 | 42.5 | 43.6 | 44.0 | 46.1 | 46.1 | 46.2 | 54.2 |
| sl | MRL | 50.7 | 49.9 | 50.6 | 50.1 | 50.1 | 50.3 | 51.5 | 48.8 | 49.7 | 51.0 | 52.1 | 48.4 | 50.7 | 51.7 | 56.1 | 56.1 | 56.6 | 47.1 |
| sm | MRL | 49.2 | 48.7 | 48.9 | 48.9 | 48.8 | 49.1 | 49.4 | 48.2 | 48.5 | 49.4 | 49.7 | 47.5 | 49.3 | 49.6 | 52.0 | 51.3 | 51.6 | 54.2 |
| sn | MRL | 43.4 | 43.3 | 43.2 | 43.3 | 43.4 | 43.2 | 44.0 | 42.3 | 42.9 | 43.5 | 44.2 | 42.0 | 43.5 | 43.8 | 46.4 | 46.1 | 46.1 | 56.0 |
| so | MRL | 42.3 | 42.1 | 42.5 | 42.0 | 42.3 | 42.1 | 42.8 | 41.1 | 41.3 | 42.6 | 43.4 | 41.1 | 42.6 | 42.6 | 45.7 | 45.3 | 45.6 | 66.8 |
| sq | MRL | 54.1 | 54.1 | 53.9 | 54.0 | 53.8 | 53.9 | 54.9 | 52.8 | 53.5 | 54.8 | 55.4 | 52.6 | 54.3 | 55.0 | 58.1 | 57.9 | 58.2 | 47.4 |
| sr | MRL | 31.2 | 34.8 | 31.7 | 34.8 | 34.4 | 36.8 | 35.8 | 28.1 | 33.1 | 38.9 | 39.4 | 31.5 | 36.6 | 38.2 | 41.9 | 43.7 | 42.5 | 62.3 |
| st | MRL | 44.6 | 44.5 | 44.3 | 44.6 | 44.4 | 44.3 | 44.9 | 43.5 | 44.1 | 44.7 | 45.3 | 43.4 | 44.8 | 45.2 | 47.2 | 47.1 | 47.1 | 58.8 |
| sw | MRL | 58.3 | 58.1 | 58.2 | 58.3 | 58.0 | 58.1 | 59.2 | 56.6 | 57.4 | 58.6 | 59.8 | 56.3 | 58.4 | 59.5 | 62.4 | 62.2 | 62.2 | 53.5 |
| ta | MRL | 52.9 | 52.5 | 52.0 | 52.4 | 52.4 | 52.4 | 53.9 | 50.3 | 51.8 | 53.5 | 54.4 | 50.0 | 52.9 | 54.2 | 58.5 | 58.2 | 58.5 | 59.6 |
| te | MRL | 52.6 | 52.5 | 52.2 | 52.2 | 52.5 | 51.9 | 53.8 | 49.9 | 51.6 | 53.5 | 54.4 | 49.9 | 52.8 | 53.8 | 58.4 | 58.1 | 58.5 | 53.6 |
| tt | MRL | 34.7 | 34.9 | 34.4 | 34.4 | 34.2 | 35.8 | 38.3 | 33.1 | 36.0 | 37.6 | 37.0 | 33.3 | 35.9 | 37.6 | 39.9 | 40.2 | 42.0 | 59.9 |
| ur | MRL | 46.2 | 46.0 | 45.9 | 45.5 | 45.5 | 45.6 | 46.5 | 44.4 | 45.3 | 46.2 | 47.0 | 44.5 | 46.4 | 46.9 | 49.2 | 48.9 | 49.3 | 62.1 |
| uz | MRL | 51.7 | 51.4 | 51.3 | 51.2 | 51.3 | 51.3 | 52.4 | 49.8 | 50.5 | 51.8 | 53.5 | 49.8 | 51.8 | 52.6 | 56.2 | 55.5 | 55.9 | 45.9 |
| yi | MRL | 36.6 | 36.0 | 36.4 | 36.5 | 36.6 | 35.9 | 36.6 | 36.2 | 36.2 | 36.5 | 36.5 | 36.2 | 36.4 | 36.3 | 37.2 | 36.9 | 37.1 | 58.1 |
| yo | MRL | 20.9 | 20.9 | 20.7 | 21.1 | 20.9 | 20.7 | 21.0 | 20.8 | 20.9 | 21.0 | 21.0 | 20.5 | 20.9 | 20.9 | 21.3 | 21.1 | 21.4 | 57.9 |
| yue | MRL | 12.2 | 12.9 | 12.7 | 12.2 | 13.3 | 12.2 | 13.2 | 10.8 | 10.9 | 12.5 | 13.5 | 11.6 | 13.2 | 13.0 | 17.8 | 17.7 | 17.1 | 20.8 |
| zu | MRL | 49.7 | 49.1 | 49.4 | 49.6 | 49.3 | 49.4 | 50.7 | 47.7 | 48.5 | 50.1 | 51.3 | 47.5 | 49.7 | 50.9 | 55.2 | 54.4 | 54.9 | 67.1 |
| ar | HRL | 47.7 | 47.4 | 48.0 | 47.4 | 47.4 | 47.1 | 49.0 | 45.1 | 46.2 | 48.4 | 50.2 | 44.6 | 49.1 | 49.1 | 56.1 | 55.2 | 56.0 | 59.4 |
| cs | HRL | 50.6 | 50.1 | 50.6 | 50.1 | 50.2 | 49.9 | 51.4 | 48.3 | 49.5 | 51.0 | 52.1 | 48.2 | 50.6 | 51.7 | 56.4 | 55.9 | 56.7 | 53.5 |
| da | HRL | 62.8 | 62.2 | 62.7 | 62.4 | 62.4 | 62.3 | 63.2 | 61.2 | 61.7 | 62.8 | 64.0 | 60.8 | 62.7 | 63.4 | 67.3 | 66.8 | 67.3 | 63.6 |
| de | HRL | 55.9 | 55.2 | 55.7 | 56.2 | 55.5 | 55.4 | 56.7 | 53.8 | 55.0 | 56.2 | 57.1 | 61.4 | 60.9 | 61.6 | 62.6 | | | |
| el | HRL | 45.1 | 44.8 | 45.5 | 45.0 | 44.9 | 44.8 | 46.4 | 43.1 | 44.1 | 45.9 | 46.9 | 43.0 | 45.5 | 46.3 | 50.9 | 50.4 | 51.2 | 54.3 |
| es | HRL | 51.6 | 51.4 | 51.7 | 51.5 | 51.4 | 51.4 | 51.9 | 50.7 | 51.1 | 51.8 | 52.3 | 50.5 | 51.6 | 52.1 | 54.5 | 54.2 | 54.6 | 61.6 |
| fa | HRL | 46.0 | 45.2 | 46.1 | 45.0 | 45.3 | 45.0 | 46.7 | 44.0 | 44.6 | 46.0 | 47.0 | 43.5 | 45.9 | 47.1 | 50.7 | 49.9 | 50.7 | 59.1 |
| fi | HRL | 48.5 | 48.2 | 48.6 | 48.0 | 48.1 | 48.0 | 49.6 | 46.4 | 47.5 | 48.9 | 49.8 | 45.1 | 47.8 | 49.1 | 55.0 | 54.6 | 55.5 | 68.5 |
| fr | HRL | 63.3 | 62.7 | 63.4 | 63.0 | 62.8 | 62.7 | 64.0 | 61.7 | 62.3 | 63.5 | 64.7 | 61.5 | 63.2 | 64.1 | 68.6 | 68.3 | 68.7 | 59.3 |
| hi | HRL | 54.4 | 53.9 | 54.3 | 53.8 | 53.8 | 53.9 | 54.8 | 52.4 | 53.0 | 54.5 | 55.3 | 52.5 | 54.4 | 55.1 | 58.2 | 57.8 | 58.4 | 68.4 |
| hu | HRL | 47.5 | 47.2 | 47.9 | 47.2 | 47.2 | 47.2 | 49.0 | 45.1 | 46.6 | 48.4 | 49.8 | 45.1 | 47.8 | 49.1 | 54.9 | 54.2 | 55.2 | 58.7 |
| id | HRL | 63.8 | 63.7 | 64.2 | 63.8 | 63.8 | 64.0 | 64.7 | 62.5 | 63.3 | 64.0 | 65.3 | 62.6 | 64.2 | 65.0 | 68.4 | 68.0 | 68.2 | 51.4 |
| it | HRL | 52.8 | 52.7 | 53.1 | 52.6 | 52.5 | 52.6 | 53.4 | 51.7 | 52.2 | 53.2 | 53.8 | 51.5 | 52.8 | 53.7 | 57.0 | 56.6 | 57.2 | 61.2 |
| iw | HRL | 46.3 | 45.4 | 46.7 | 45.9 | 45.7 | 46.0 | 48.0 | 43.6 | 44.9 | 47.1 | 48.9 | 43.3 | 46.8 | 47.9 | 55.2 | 54.2 | 55.4 | 59.6 |
| ja | HRL | 29.3 | 29.0 | 29.4 | 28.7 | 28.9 | 29.0 | 30.2 | 26.9 | 28.2 | 29.6 | 31.0 | 27.1 | 29.6 | 30.8 | 36.4 | 35.7 | 36.9 | 59.8 |
| ko | HRL | 26.6 | 26.4 | 26.9 | 26.5 | 26.5 | 26.3 | 27.6 | 24.3 | 25.7 | 27.2 | 28.7 | 24.4 | 27.2 | 28.3 | 33.4 | 32.3 | 33.4 | 60.7 |
| ms | HRL | 64.1 | 63.9 | 64.1 | 63.8 | 63.9 | 63.9 | 64.7 | 62.4 | 63.2 | 64.2 | 65.4 | 62.5 | 64.1 | 64.9 | 67.7 | 67.4 | 67.6 | 70.6 |
| nl | HRL | 52.5 | 52.4 | 52.4 | 52.3 | 52.3 | 52.3 | 53.1 | 51.2 | 51.8 | 52.8 | 53.5 | 51.1 | 52.6 | 53.2 | 55.8 | 55.7 | 56.2 | 68.1 |
| no | HRL | 57.6 | 57.2 | 57.5 | 57.5 | 57.6 | 57.5 | 57.9 | 56.3 | 56.6 | 57.7 | 58.7 | 56.1 | 57.6 | 58.2 | 60.8 | 60.6 | 60.8 | 38.0 |
| pl | HRL | 43.8 | 43.3 | 43.4 | 43.1 | 43.3 | 43.5 | 44.2 | 42.0 | 42.7 | 44.1 | 44.8 | 42.0 | 43.5 | 44.7 | 48.4 | 48.3 | 49.0 | 61.8 |
| pt | HRL | 64.0 | 63.9 | 64.0 | 64.0 | 63.6 | 63.7 | 64.7 | 62.6 | 63.4 | 64.2 | 65.1 | 62.6 | 64.4 | 65.1 | 68.4 | 67.9 | 68.4 | 52.1 |
| ro | HRL | 57.0 | 57.2 | 56.9 | 56.9 | 56.7 | 56.8 | 58.0 | 55.2 | 56.4 | 57.6 | 58.8 | 55.3 | 57.5 | 58.1 | 62.6 | 62.4 | 62.7 | 29.3 |
| ru | HRL | 49.5 | 49.2 | 49.3 | 49.3 | 49.0 | 48.9 | 50.0 | 47.7 | 48.4 | 49.8 | 51.1 | 47.5 | 49.7 | 50.5 | 55.3 | 54.9 | 55.4 | 69.6 |
| sk | HRL | 51.6 | 51.3 | 51.3 | 51.2 | 51.2 | 51.0 | 52.5 | 49.7 | 50.4 | 52.1 | 53.6 | 49.3 | 51.7 | 52.9 | 58.6 | 58.2 | 58.7 | 35.0 |
| sv | HRL | 62.0 | 61.5 | 61.8 | 61.8 | 61.7 | 61.6 | 62.4 | 60.0 | 60.9 | 62.4 | 63.4 | 59.9 | 61.8 | 63.1 | 67.2 | 66.7 | 67.0 | 59.9 |
| th | HRL | 44.2 | 44.0 | 43.4 | 43.5 | 43.4 | 44.0 | 44.7 | 41.6 | 42.8 | 45.0 | 45.6 | 41.3 | 44.1 | 45.6 | 50.1 | 50.0 | 50.2 | 57.8 |
| tr | HRL | 53.4 | 52.8 | 52.7 | 52.9 | 52.8 | 53.0 | 54.3 | 51.0 | 52.0 | 53.6 | 55.1 | 50.9 | 53.2 | 54.5 | 59.6 | 59.2 | 59.8 | 59.0 |

| lang | cat. | B | T | $T_{75}$ | $T_{50}$ | $T_{25}$ | $T_{Gatitos}$ | $C_M$ | $C_P$ | $C_{MP}$ | $C_M$ T | $G_M$ | $G_P$ | $G_{MP}$ | $G_M$ T | $B_{BIG}$ | $T_{BIG}$ | $(C_M$ T$)_{BIG}$ | NLLB |
|---|---|---|---|---|---|---|---|---|---|---|---|---|---|---|---|---|---|---|---|
| uk | HRL | 49.5 | 49.2 | 49.5 | 49.2 | 49.3 | 49.0 | 50.5 | 47.4 | 48.6 | 49.9 | 51.2 | 47.5 | 49.9 | 50.6 | 55.4 | 55.0 | 55.5 | 65.2 |
| vi | HRL | 49.9 | 50.0 | 50.0 | 49.7 | 49.9 | 49.8 | 50.7 | 47.9 | 48.8 | 51.0 | 51.8 | 47.9 | 50.4 | 51.2 | 55.9 | 55.3 | 56.1 | 71.4 |
| zh | HRL | 22.3 | 22.1 | 22.1 | 22.2 | 22.0 | 22.0 | 23.0 | 19.9 | 21.2 | 22.7 | 24.1 | 20.2 | 22.6 | 23.4 | 29.6 | 29.3 | 29.9 | 56.0 |
| ar-Latn | OOM | 3.0 | 3.0 | 3.0 | 3.6 | 3.5 | 2.9 | 2.3 | 2.2 | 2.9 | 3.8 | 2.9 | 3.0 | 2.8 | 3.6 | 1.6 | 2.1 | 2.1 | - |
| kr | OOM | 11.1 | 6.3 | 5.3 | 4.3 | 4.8 | 17.5 | 17.4 | 4.2 | 15.9 | 6.7 | 17.0 | 12.8 | 17.9 | 3.9 | 15.3 | 6.0 | 6.7 | 31.5 |
| ki | OOM | 8.7 | 14.6 | 13.1 | 15.5 | 15.3 | 15.9 | 14.8 | 13.6 | 15.0 | 14.5 | 16.0 | 10.5 | 16.9 | 12.3 | 14.9 | 16.1 | 12.3 | 40.1 |
| taq | OOM | 9.8 | 16.4 | 15.1 | 18.0 | 18.2 | 18.0 | 18.9 | 14.8 | 15.6 | 16.0 | 17.6 | 12.6 | 18.5 | 14.9 | 16.0 | 18.0 | 16.3 | 25.9 |
| nn | OOM | 11.7 | 20.4 | 19.8 | 22.3 | 21.8 | 16.0 | 21.9 | 18.8 | 20.7 | 20.2 | 21.7 | 16.5 | 23.4 | 18.5 | 19.4 | 22.4 | 21.1 | 56.1 |
| luo | OOM | 2.4 | 15.2 | 6.5 | 14.8 | 6.8 | 7.9 | 10.3 | 3.9 | 7.2 | 10.9 | 13.7 | 3.6 | 14.8 | 6.6 | 11.4 | 16.2 | 19.8 | 41.7 |
| lmo | OOM | 11.6 | 19.6 | 18.4 | 21.6 | 21.1 | 21.0 | 22.5 | 17.9 | 19.6 | 20.1 | 21.5 | 15.8 | 22.4 | 18.1 | 18.8 | 21.4 | 20.5 | 38.7 |
| li | OOM | 11.3 | 20.7 | 19.2 | 23.0 | 22.7 | 20.9 | 22.4 | 19.3 | 20.5 | 20.6 | 22.4 | 16.3 | 23.9 | 18.7 | 17.5 | 22.9 | 19.2 | 51.6 |
| fur | OOM | 11.8 | 20.3 | 19.3 | 22.5 | 22.1 | 20.7 | 22.0 | 18.2 | 20.5 | 20.1 | 22.5 | 16.0 | 23.8 | 18.8 | 19.5 | 22.2 | 18.7 | 58.7 |
| szl | OOM | 3.8 | 15.4 | 14.6 | 16.9 | 16.4 | 15.3 | 17.1 | 13.9 | 15.0 | 15.2 | 16.8 | 11.9 | 17.4 | 13.8 | 15.1 | 16.7 | 15.7 | 56.7 |
| lij | OOM | 11.1 | 20.7 | 19.1 | 22.5 | 22.1 | 22.7 | 22.3 | 18.6 | 22.6 | 20.1 | 22.6 | 16.2 | 23.7 | 18.9 | 20.3 | 22.3 | 18.6 | 56.1 |
| bjn | OOM | 9.2 | 15.1 | 14.2 | 16.7 | 16.4 | 16.4 | 16.9 | 13.6 | 14.3 | 14.9 | 16.9 | 11.9 | 17.2 | 13.9 | 14.7 | 16.7 | 14.7 | 51.8 |
| sc | OOM | 11.1 | 19.8 | 18.5 | 22.0 | 21.4 | 21.7 | 21.2 | 17.6 | 20.9 | 19.3 | 22.0 | 16.4 | 22.7 | 18.8 | 19.3 | 21.7 | 19.4 | 58.1 |
| ss | OOM | 8.8 | 15.4 | 13.7 | 16.5 | 16.2 | 16.6 | 15.6 | 14.2 | 16.0 | 15.3 | 16.8 | 11.1 | 17.8 | 13.5 | 15.5 | 17.1 | 10.5 | 49.2 |
| bem | OOM | 8.5 | 14.7 | 13.0 | 15.8 | 15.5 | 14.2 | 14.9 | 13.5 | 15.2 | 14.4 | 16.2 | 10.7 | 16.9 | 12.6 | 10.3 | 16.5 | 12.6 | 42.3 |
| lua | OOM | 9.1 | 15.1 | 14.2 | 16.6 | 16.8 | 17.3 | 16.2 | 13.6 | 15.3 | 14.9 | 16.5 | 12.9 | 17.1 | 13.7 | 15.2 | 16.3 | 14.4 | 39.4 |
| kam | OOM | 3.0 | 15.1 | 5.9 | 13.2 | 11.5 | 8.5 | 11.3 | 4.2 | 6.5 | 10.0 | 10.6 | 4.2 | 14.6 | 6.2 | 11.4 | 15.6 | 18.3 | 28.8 |
| min-Arab | OOM | 0.7 | 1.0 | 0.3 | 1.2 | 0.8 | 4.4 | 1.4 | 0.6 | 0.9 | 3.5 | 4.8 | 0.4 | 6.3 | 1.0 | 0.3 | 0.3 | 0.3 | - |
| ace-Arab | OOM | 0.7 | 1.3 | 0.3 | 0.6 | 0.9 | 1.3 | 1.2 | 3.0 | 1.0 | 3.7 | 2.0 | 0.8 | 6.5 | 1.9 | 0.4 | 0.4 | 0.5 | 21.1 |
| mos | OOM | 2.7 | 14.7 | 6.5 | 12.7 | 10.2 | 9.5 | 11.5 | 4.1 | 6.9 | 10.0 | 10.7 | 3.5 | 13.8 | 6.7 | 11.1 | 15.6 | 17.7 | 26.4 |
| ast | OOM | 11.9 | 21.2 | 19.8 | 23.1 | 23.7 | 24.0 | 22.4 | 18.5 | 21.3 | 20.5 | 24.6 | 16.7 | 24.2 | 19.6 | 20.2 | 23.0 | 20.9 | 59.6 |
| ars | OOM | 0.3 | 11.6 | 13.3 | 13.7 | 12.9 | 5.8 | 17.5 | 2.1 | 15.6 | 12.8 | 19.6 | 5.3 | 12.1 | 10.2 | 0.4 | 4.9 | 11.7 | 54.2 |
| kea | OOM | 3.3 | 18.8 | 7.1 | 15.5 | 8.6 | 10.6 | 13.5 | 4.6 | 8.8 | 13.1 | 12.4 | 4.1 | 17.4 | 9.3 | 14.1 | 18.2 | 21.2 | 44.9 |
| taq-Tfng | OOM | 2.0 | 4.6 | 5.5 | 1.7 | 5.2 | 3.9 | 5.3 | 3.7 | 5.2 | 5.1 | 5.9 | 4.7 | 6.6 | 6.3 | 0.8 | 0.6 | 8.1 | 19.3 |
| bjn-Arab | OOM | 5.9 | 5.5 | 2.1 | 4.7 | 4.7 | 12.4 | 9.6 | 11.9 | 9.5 | 9.8 | 12.6 | 5.4 | 12.7 | 9.6 | 0.4 | 4.1 | 8.0 | 20.4 |

| lang | cat. | B | T | $T_{75}$ | $T_{50}$ | $T_{25}$ | $T_{Gatitos}$ | $C_M$ | $C_P$ | $C_{MP}$ | $C_M$ T | $G_M$ | $G_P$ | $G_{MP}$ | $G_M$ T | $B_{BIG}$ | $T_{BIG}$ | $(C_M$ T$)_{BIG}$ | NLLB |
|---|---|---|---|---|---|---|---|---|---|---|---|---|---|---|---|---|---|---|---|
| umb | OOM | 2.2 | 13.7 | 5.7 | 12.0 | 7.2 | 7.6 | 11.0 | 3.4 | 8.9 | 10.9 | 12.5 | 5.2 | 11.0 | 9.6 | 9.8 | 14.9 | 16.3 | 31.2 |
| crh-Latn | OOM | 8.8 | 14.8 | 14.0 | 16.1 | 15.8 | 16.1 | 16.3 | 13.3 | 15.0 | 14.4 | 16.0 | 11.6 | 16.7 | 13.6 | 14.0 | 16.1 | 13.5 | 51.1 |
| kr-Arab | OOM | 4.5 | 2.4 | 0.5 | 1.0 | 0.9 | 8.7 | 6.9 | 8.3 | 7.1 | 4.0 | 8.9 | 4.4 | 8.9 | 2.6 | 1.0 | 0.5 | 1.7 | 12.7 |
| zh-Hant | OOM | 8.3 | 7.8 | 7.2 | 7.2 | 7.3 | 9.6 | 9.1 | 7.3 | 7.9 | 8.9 | 8.5 | 6.9 | 8.0 | 8.4 | 13.9 | 13.9 | 10.4 | 17.6 |
| ajp | OOM | 0.4 | 13.1 | 14.2 | 14.3 | 13.9 | 6.2 | 19.1 | 3.4 | 17.5 | 12.9 | 17.6 | 8.6 | 13.3 | 9.8 | 1.1 | 5.6 | 19.5 | 56.0 |
| ks-Deva | OOM | 3.6 | 3.5 | 3.9 | 4.4 | 4.4 | 3.5 | 3.6 | 3.3 | 3.5 | 5.3 | 4.0 | 3.5 | 3.7 | 5.6 | 3.3 | 3.1 | 3.7 | 20.4 |
| kab | OOM | 1.5 | 4.4 | 2.7 | 8.0 | 1.6 | 4.7 | 2.3 | 2.1 | 1.7 | 2.3 | 1.8 | 1.5 | 3.7 | 0.9 | 4.7 | 3.4 | 4.5 | 37.9 |
| azb | OOM | 0.2 | 0.6 | 14.6 | 2.3 | 5.4 | 0.4 | 0.5 | 0.8 | 0.4 | 0.4 | 0.5 | 0.2 | 0.4 | 0.4 | 0.5 | 0.7 | 0.6 | 27.8 |

**Table 24:** Full Results of CHRF on FLORES for the models that we trained, in the en→xx direction. See Table 23 to demystify the model abbreviations.

| lang | cat. | B | T | $T_{75}$ | $T_{50}$ | $T_{25}$ | $T_{GAT}$ | $C_M$ | $C_P$ | $C_{MP}$ | $C_M$ T | $G_M$ | $G_P$ | $G_{MP}$ | $G_M$ T | $B_{BIG}$ | $T_{BIG}$ | $(C_M$ T$)_{BIG}$ | NLLB |
|---|---|---|---|---|---|---|---|---|---|---|---|---|---|---|---|---|---|---|---|
| $\mu$ | mean | 47.2 | 47.2 | 47.1 | 46.7 | 46.8 | 46.4 | 47.4 | 46.0 | 46.4 | 47.3 | 47.9 | 45.9 | 47.3 | 47.8 | 53.3 | 53.0 | 52.9 | 59.9 |
| HRL | mean | 57.2 | 56.9 | 56.9 | 56.8 | 56.8 | 56.5 | 57.7 | 55.6 | 56.5 | 57.5 | 58.4 | 55.5 | 57.2 | 58.1 | 62.6 | 62.0 | 62.6 | 65.7 |
| MRL | mean | 52.1 | 51.8 | 51.9 | 51.5 | 51.8 | 51.3 | 52.5 | 50.6 | 51.2 | 52.2 | 53.2 | 50.5 | 52.1 | 52.9 | 58.1 | 57.7 | 58.0 | 63.3 |
| LRL | mean | 43.6 | 43.8 | 43.5 | 43.0 | 43.5 | 42.8 | 43.8 | 42.4 | 42.8 | 43.9 | 44.3 | 42.4 | 43.6 | 44.2 | 50.2 | 50.0 | 49.7 | 60.4 |
| URL | mean | 37.0 | 37.5 | 37.2 | 36.7 | 36.6 | 36.4 | 36.9 | 36.7 | 36.2 | 36.9 | 37.0 | 36.4 | 37.5 | 37.1 | 43.4 | 43.3 | 42.3 | 49.2 |
| OOM | mean | 32.5 | 32.5 | 32.5 | 31.8 | 32.1 | 31.5 | 33.0 | 31.5 | 31.4 | 32.7 | 33.0 | 31.4 | 32.8 | 32.8 | 37.9 | 37.5 | 37.3 | - |
| $\Delta_{UL}$ | URL | 39.0 | 39.4 | 39.1 | 38.6 | 38.6 | 38.3 | 38.9 | 38.4 | 38.2 | 39.0 | 39.2 | 38.2 | 39.3 | 39.3 | 45.4 | 45.3 | 44.5 | 52.6 |
| — | — | — | — | — | — | — | — | — | — | — | — | — | — | — | — | — | — | — | — |
| ace | URL | 29.6 | 30.0 | 30.2 | 29.0 | 29.0 | 29.0 | 29.1 | 30.8 | 29.3 | 28.8 | 27.8 | 30.2 | 31.4 | 29.4 | 36.2 | 35.1 | 34.1 | 41.2 |
| acm | URL | 51.2 | 50.7 | 50.9 | 50.5 | 50.5 | 50.2 | 51.5 | 49.0 | 50.0 | 51.5 | 52.2 | 48.9 | 50.8 | 52.2 | 58.1 | 57.7 | 57.4 | 61.4 |
| acq | URL | 52.5 | 51.8 | 52.2 | 51.9 | 52.1 | 51.7 | 52.9 | 50.5 | 51.2 | 52.8 | 54.0 | 50.3 | 52.5 | 53.6 | 59.7 | 58.8 | 58.9 | 32.9 |
| aeb | URL | 46.1 | 45.9 | 46.0 | 45.7 | 45.8 | 45.4 | 46.6 | 44.4 | 45.1 | 46.4 | 47.2 | 44.4 | 46.1 | 47.2 | 53.2 | 52.3 | 52.3 | 73.7 |
| apc | URL | 50.8 | 50.6 | 50.9 | 50.4 | 50.5 | 50.4 | 51.4 | 49.2 | 50.0 | 51.2 | 52.1 | 49.0 | 51.0 | 52.2 | 59.5 | 58.3 | 58.4 | 62.2 |
| ar-MA | URL | 41.6 | 41.6 | 41.6 | 41.2 | 41.5 | 41.0 | 41.9 | 40.1 | 41.3 | 42.1 | 42.6 | 40.1 | 42.1 | 42.7 | 49.0 | 48.2 | 47.7 | 35.8 |
| arz | URL | 48.7 | 48.5 | 48.8 | 48.6 | 48.4 | 48.0 | 49.2 | 47.1 | 48.2 | 48.9 | 49.9 | 47.0 | 48.9 | 49.8 | 55.9 | 54.9 | 54.9 | 64.5 |
| awa | URL | 54.3 | 53.8 | 53.8 | 53.5 | 53.8 | 53.5 | 53.7 | 52.6 | 53.2 | 53.9 | 55.2 | 52.7 | 54.1 | 54.9 | 60.5 | 60.5 | 60.5 | 35.4 |
| ber | URL | 23.7 | 23.1 | 22.3 | 21.9 | 22.5 | 17.5 | 21.4 | 22.3 | 21.1 | 20.5 | 22.0 | 23.4 | 22.5 | 20.9 | 34.3 | 32.4 | 28.3 | 43.0 |
| bm | URL | 22.1 | 25.7 | 24.9 | 23.8 | 22.8 | 23.9 | 23.4 | 22.2 | 23.3 | 25.4 | 21.6 | 22.6 | 22.6 | 23.4 | 24.2 | 29.7 | 26.2 | 53.5 |
| bug | URL | 24.8 | 25.5 | 25.5 | 24.4 | 25.0 | 24.5 | 26.1 | 25.4 | 24.5 | 25.1 | 26.2 | 24.4 | 26.6 | 25.2 | 29.3 | 29.0 | 27.8 | 35.7 |
| cjk | URL | 20.2 | 20.7 | 20.6 | 19.6 | 19.8 | 19.1 | 20.5 | 19.9 | 20.3 | 20.9 | 20.1 | 19.8 | 20.6 | 20.2 | 24.1 | 23.7 | 23.7 | 51.3 |
| din | URL | 21.8 | 21.3 | 21.7 | 20.2 | 20.1 | 20.3 | 21.1 | 20.9 | 20.1 | 20.7 | 20.5 | 21.2 | 20.8 | 20.5 | 22.3 | 22.2 | 22.1 | 30.0 |
| dyu | URL | 18.2 | 19.1 | 19.4 | 18.2 | 17.7 | 17.8 | 19.2 | 17.4 | 18.0 | 19.1 | 18.6 | 17.6 | 18.5 | 18.3 | 20.3 | 20.4 | 19.3 | 67.4 |
| fon | URL | 19.7 | 21.5 | 20.3 | 19.4 | 19.0 | 19.4 | 19.8 | 20.9 | 19.3 | 19.5 | 19.8 | 20.6 | 21.2 | 19.8 | 22.3 | 23.1 | 20.6 | 36.6 |
| hne | URL | 55.4 | 54.4 | 55.0 | 54.6 | 55.2 | 54.8 | 54.3 | 54.5 | 53.7 | 54.7 | 56.0 | 54.0 | 56.0 | 56.4 | 64.0 | 63.9 | 63.9 | 44.9 |
| ilo | URL | 44.7 | 46.6 | 46.0 | 45.8 | 44.5 | 45.2 | 44.7 | 45.1 | 44.1 | 45.7 | 43.5 | 44.4 | 44.4 | 45.8 | 55.2 | 55.1 | 55.6 | 42.2 |
| kac | URL | 18.7 | 19.1 | 17.5 | 17.7 | 17.3 | 18.4 | 17.8 | 18.6 | 17.4 | 18.9 | 18.3 | 20.2 | 19.4 | 18.0 | 21.8 | 20.7 | 18.9 | 66.7 |
| kbp | URL | 23.5 | 25.0 | 23.4 | 23.1 | 22.8 | 22.9 | 22.9 | 24.9 | 23.3 | 22.9 | 23.4 | 25.0 | 23.6 | 23.6 | 24.5 | 25.8 | 23.2 | 62.3 |
| kmb | URL | 20.5 | 20.5 | 20.2 | 18.8 | 19.7 | 19.0 | 20.4 | 20.0 | 20.1 | 20.4 | 20.3 | 20.4 | 20.7 | 20.3 | 23.6 | 23.4 | 23.2 | 53.9 |
| ltg | URL | 43.9 | 44.7 | 44.7 | 44.5 | 44.8 | 44.0 | 43.7 | 44.3 | 43.9 | 44.3 | 43.8 | 45.0 | 44.5 | 55.9 | 58.0 | 57.3 | 33.7 |
| lus | URL | 20.3 | 22.9 | 22.9 | 19.9 | 20.9 | 23.4 | 22.7 | 21.7 | 22.2 | 25.2 | 21.5 | 19.8 | 23.2 | 21.4 | 23.5 | 23.7 | 22.6 | 73.8 |
| mag | URL | 56.6 | 55.2 | 55.5 | 56.0 | 55.7 | 55.3 | 56.0 | 55.2 | 54.7 | 55.9 | 57.5 | 54.9 | 57.0 | 57.6 | 63.8 | 63.3 | 63.5 | 35.4 |
| mai | URL | 54.8 | 54.4 | 54.7 | 54.7 | 54.5 | 54.5 | 54.6 | 54.0 | 53.7 | 54.9 | 56.8 | 53.8 | 55.7 | 56.1 | 63.2 | 62.0 | 63.4 | 44.6 |
| min | URL | 39.8 | 40.8 | 41.0 | 40.8 | 40.9 | 40.2 | 39.6 | 41.8 | 40.3 | 38.9 | 40.2 | 42.4 | 42.4 | 39.3 | 48.2 | 49.2 | 47.5 | 43.7 |
| mni | URL | 32.8 | 31.6 | 32.1 | 31.8 | 28.6 | 31.7 | 28.6 | 34.9 | 27.7 | 28.4 | 29.7 | 29.8 | 32.8 | 28.2 | 37.9 | 44.1 | 40.2 | 36.7 |
| nus | URL | 18.0 | 19.0 | 18.2 | 17.8 | 17.3 | 17.4 | 17.9 | 18.2 | 17.2 | 16.4 | 17.3 | 18.0 | 18.1 | 16.5 | 20.1 | 20.4 | 17.1 | 32.7 |
| pag | URL | 38.2 | 39.8 | 39.0 | 39.5 | 37.3 | 37.2 | 37.7 | 39.5 | 37.5 | 37.5 | 38.5 | 38.5 | 39.7 | 38.2 | 47.3 | 46.8 | 44.9 | 64.0 |
| pap | URL | 57.3 | 56.9 | 56.9 | 56.0 | 56.2 | 55.4 | 57.3 | 54.7 | 55.9 | 56.2 | 55.3 | 55.6 | 56.5 | 68.0 | 66.2 | 66.3 | 51.5 |
| quy | URL | 27.8 | 29.3 | 28.3 | 28.3 | 27.7 | 28.8 | 28.6 | 28.5 | 28.7 | 29.6 | 26.8 | 28.2 | 27.9 | 28.7 | 33.3 | 33.2 | 33.5 | 52.3 |
| scn | URL | 51.2 | 51.6 | 51.4 | 50.9 | 51.2 | 50.5 | 51.6 | 50.3 | 50.2 | 51.2 | 51.8 | 50.3 | 51.3 | 51.6 | 59.2 | 58.9 | 57.8 | 60.7 |
| sg | URL | 21.4 | 23.3 | 22.0 | 21.4 | 21.6 | 21.0 | 21.6 | 22.4 | 21.8 | 22.3 | 22.3 | 22.6 | 22.8 | 21.8 | 27.3 | 25.6 | 24.8 | 65.3 |

| lang | cat. | B | T | $T_{75}$ | $T_{50}$ | $T_{25}$ | $T_{GAT}$ | $C_M$ | $C_P$ | $C_{MP}$ | $C_M$ T | $G_M$ | $G_P$ | $G_{MP}$ | $G_M$ T | $B_{BIG}$ | $T_{BIG}$ | $(C_M$ T$)_{BIG}$ | NLLB |
|---|---|---|---|---|---|---|---|---|---|---|---|---|---|---|---|---|---|---|---|
| shn | URL | 33.1 | 34.4 | 32.4 | 32.7 | 31.6 | 29.7 | 30.6 | 32.7 | 32.7 | 30.2 | 31.6 | 32.1 | 33.6 | 32.8 | 42.7 | 40.7 | 38.0 | 48.4 |
| vec | URL | 53.4 | 54.6 | 54.0 | 53.9 | 54.5 | 53.9 | 54.1 | 52.9 | 52.9 | 53.5 | 54.1 | 52.5 | 53.8 | 53.8 | 63.0 | 62.7 | 61.4 | 42.0 |
| war | URL | 58.1 | 58.9 | 58.5 | 58.6 | 58.6 | 58.7 | 57.3 | 57.6 | 56.8 | 57.0 | 58.1 | 57.4 | 58.6 | 57.3 | 66.0 | 65.8 | 65.2 | 42.9 |
| am | LRL | 46.8 | 46.5 | 46.9 | 46.2 | 47.1 | 46.1 | 47.4 | 45.2 | 46.0 | 47.0 | 48.8 | 44.7 | 47.3 | 48.5 | 54.7 | 53.8 | 54.2 | 71.1 |
| as | LRL | 41.1 | 41.0 | 38.5 | 37.4 | 39.7 | 37.0 | 38.8 | 36.3 | 36.2 | 41.9 | 41.8 | 37.4 | 39.3 | 41.2 | 49.5 | 47.3 | 46.3 | 45.4 |
| ay | LRL | 22.2 | 23.8 | 24.3 | 23.0 | 22.4 | 23.2 | 23.5 | 22.9 | 23.6 | 24.6 | 21.8 | 22.5 | 23.1 | 23.4 | 25.7 | 28.8 | 28.7 | 70.2 |
| ba | LRL | 37.7 | 36.0 | 32.9 | 35.1 | 35.9 | 31.0 | 37.0 | 34.1 | 33.5 | 40.2 | 39.4 | 35.6 | 35.2 | 38.7 | 41.4 | 39.8 | 42.1 | 52.1 |
| ban | LRL | 44.9 | 44.8 | 45.1 | 43.6 | 44.1 | 44.0 | 44.3 | 45.1 | 43.5 | 43.6 | 44.3 | 44.9 | 45.2 | 45.7 | 48.9 | 49.9 | 48.5 | 54.9 |
| be | LRL | 47.9 | 47.6 | 47.6 | 47.6 | 47.3 | 47.4 | 47.1 | 48.2 | 46.5 | 47.2 | 48.0 | 48.8 | 46.2 | 47.7 | 48.5 | 52.2 | 51.8 | 48.2 |
| bho | LRL | 47.1 | 46.7 | 46.9 | 46.6 | 47.1 | 46.8 | 47.1 | 46.5 | 46.4 | 47.1 | 48.2 | 46.3 | 47.8 | 48.2 | 54.5 | 54.1 | 54.1 | 53.4 |
| bo | LRL | 13.7 | 15.2 | 12.4 | 14.4 | 13.7 | 14.0 | 16.2 | 13.0 | 16.0 | 17.2 | 16.2 | 14.5 | 15.6 | 16.3 | 22.2 | 20.7 | 19.5 | 63.6 |
| cy | LRL | 68.5 | 68.2 | 68.5 | 67.7 | 68.1 | 67.5 | 69.2 | 66.2 | 67.3 | 69.0 | 70.0 | 65.8 | 68.5 | 69.6 | 75.3 | 74.9 | 75.6 | 44.0 |
| dz | LRL | 21.4 | 23.3 | 21.5 | 18.2 | 22.2 | 18.5 | 21.3 | 20.3 | 22.0 | 22.4 | 21.6 | 20.0 | 20.5 | 20.9 | 29.9 | 29.2 | 28.8 | 63.0 |
| ee | LRL | 27.4 | 28.9 | 28.8 | 27.6 | 26.8 | 28.5 | 27.4 | 28.4 | 27.7 | 28.7 | 26.7 | 29.3 | 27.6 | 28.8 | 28.0 | 30.0 | 26.7 | 52.0 |
| eo | LRL | 61.7 | 61.3 | 61.4 | 61.2 | 61.3 | 61.1 | 62.0 | 60.4 | 61.1 | 61.9 | 62.4 | 60.3 | 61.6 | 62.4 | 65.8 | 65.2 | 66.0 | 52.4 |
| eu | LRL | 50.7 | 50.2 | 50.2 | 50.1 | 50.1 | 49.7 | 51.1 | 48.6 | 49.9 | 50.7 | 51.8 | 48.8 | 50.5 | 51.7 | 56.9 | 56.3 | 56.9 | 68.0 |
| fa-AF | LRL | 54.6 | 54.1 | 54.3 | 54.0 | 54.4 | 53.8 | 55.0 | 52.9 | 53.4 | 54.7 | 55.9 | 53.1 | 54.6 | 55.7 | 61.2 | 60.2 | 60.9 | 61.7 |
| ff | LRL | 20.5 | 21.9 | 21.6 | 20.9 | 20.4 | 20.7 | 21.3 | 20.5 | 21.5 | 22.0 | 20.4 | 20.9 | 21.0 | 20.9 | 22.8 | 23.8 | 23.6 | 61.2 |
| fj | LRL | 32.5 | 31.6 | 31.3 | 29.7 | 31.2 | 29.4 | 29.7 | 32.8 | 28.9 | 28.8 | 31.7 | 32.4 | 32.1 | 29.3 | 36.5 | 36.7 | 32.5 | 54.6 |
| fo | LRL | 50.6 | 50.5 | 50.3 | 50.4 | 50.3 | 49.8 | 51.3 | 49.7 | 50.3 | 50.6 | 51.4 | 49.3 | 50.4 | 50.9 | 58.9 | 58.2 | 58.6 | 52.1 |
| ga | LRL | 57.0 | 56.6 | 56.3 | 56.3 | 56.2 | 56.2 | 57.5 | 55.0 | 56.1 | 57.5 | 58.5 | 54.8 | 57.1 | 58.0 | 65.2 | 64.4 | 65.6 | 61.8 |
| gn | LRL | 31.6 | 34.3 | 33.8 | 32.6 | 32.5 | 32.8 | 32.4 | 31.2 | 31.4 | 34.4 | 30.6 | 31.6 | 31.9 | 33.6 | 42.2 | 42.4 | 42.3 | 52.9 |
| gu | LRL | 56.6 | 56.0 | 56.1 | 55.9 | 56.3 | 55.8 | 57.1 | 54.7 | 55.6 | 56.8 | 57.8 | 54.7 | 56.6 | 57.3 | 63.0 | 62.3 | 63.1 | 60.1 |
| ht | LRL | 55.3 | 54.8 | 55.0 | 55.1 | 55.0 | 54.4 | 55.7 | 54.0 | 54.5 | 55.5 | 56.5 | 54.0 | 55.2 | 56.1 | 61.3 | 60.7 | 61.2 | 63.6 |
| jv | LRL | 52.2 | 51.5 | 52.0 | 51.4 | 51.6 | 51.3 | 52.5 | 51.1 | 51.1 | 52.3 | 53.1 | 50.8 | 52.3 | 53.1 | 58.5 | 57.9 | 57.9 | 54.6 |
| kg | LRL | 30.9 | 31.8 | 31.6 | 31.4 | 31.0 | 31.0 | 30.4 | 32.6 | 30.9 | 30.5 | 30.5 | 32.5 | 31.3 | 31.3 | 34.2 | 35.1 | 31.1 | 77.2 |
| kn | LRL | 53.1 | 52.7 | 52.7 | 52.7 | 52.9 | 52.4 | 53.6 | 51.4 | 52.0 | 53.1 | 54.4 | 51.2 | 53.2 | 53.9 | 59.0 | 58.4 | 59.2 | 54.6 |
| ks | LRL | 31.6 | 31.5 | 32.6 | 30.5 | 31.6 | 30.0 | 32.2 | 31.1 | 30.9 | 32.4 | 33.7 | 32.0 | 33.2 | 34.1 | 43.0 | 43.2 | 40.1 | 43.7 |
| ku | LRL | 43.5 | 43.1 | 43.4 | 42.8 | 42.9 | 42.3 | 43.4 | 42.2 | 42.4 | 43.6 | 44.3 | 42.1 | 43.5 | 44.1 | 51.2 | 49.8 | 49.9 | 65.0 |
| ky | LRL | 45.3 | 45.0 | 44.9 | 44.9 | 45.2 | 44.3 | 45.8 | 43.6 | 44.6 | 45.7 | 46.5 | 43.6 | 45.5 | 46.2 | 50.3 | 50.2 | 50.2 | 64.0 |
| lb | LRL | 59.8 | 59.2 | 59.3 | 59.0 | 59.3 | 59.0 | 60.3 | 58.2 | 58.6 | 60.1 | 61.1 | 58.1 | 59.9 | 60.6 | 66.5 | 66.2 | 66.6 | 65.4 |
| lg | LRL | 29.6 | 31.0 | 30.5 | 29.5 | 29.2 | 29.9 | 29.9 | 30.2 | 30.0 | 31.0 | 29.1 | 29.9 | 29.3 | 30.3 | 38.7 | 38.1 | 36.2 | 73.4 |
| ln | LRL | 36.5 | 37.5 | 37.3 | 36.2 | 36.3 | 36.7 | 36.6 | 35.6 | 36.4 | 37.7 | 35.7 | 35.5 | 36.0 | 37.9 | 44.9 | 44.4 | 44.3 | 59.5 |
| mn | LRL | 47.8 | 47.5 | 47.4 | 47.1 | 47.2 | 46.9 | 48.2 | 46.0 | 46.8 | 47.8 | 49.2 | 45.9 | 48.0 | 48.9 | 54.3 | 54.1 | 54.2 | 62.7 |
| mt | LRL | 69.7 | 69.1 | 69.2 | 69.2 | 69.3 | 68.9 | 70.2 | 68.0 | 68.6 | 69.8 | 70.9 | 67.8 | 69.5 | 70.6 | 75.5 | 75.0 | 75.4 | 68.1 |
| my | LRL | 44.7 | 43.9 | 44.0 | 43.7 | 44.5 | 43.0 | 45.5 | 42.4 | 44.0 | 45.0 | 45.6 | 42.6 | 44.3 | 45.8 | 50.5 | 49.9 | 50.4 | 66.7 |
| nso | LRL | 47.8 | 47.9 | 47.9 | 47.2 | 47.1 | 47.1 | 47.6 | 46.2 | 46.9 | 47.5 | 48.4 | 46.5 | 47.8 | 48.7 | 57.1 | 56.6 | 56.2 | 52.5 |
| oc | LRL | 65.8 | 65.8 | 65.6 | 65.1 | 65.3 | 65.2 | 66.6 | 64.6 | 66.0 | 66.5 | 66.3 | 64.8 | 66.0 | 66.4 | 71.2 | 70.4 | 70.3 | 67.9 |
| om | LRL | 30.3 | 31.3 | 30.5 | 28.5 | 30.3 | 29.7 | 29.4 | 29.9 | 29.0 | 30.6 | 28.4 | 29.8 | 28.9 | 30.0 | 43.5 | 42.3 | 42.1 | 61.8 |
| pa | LRL | 55.6 | 55.1 | 55.5 | 55.4 | 55.4 | 54.9 | 56.2 | 54.0 | 54.9 | 56.3 | 57.2 | 53.8 | 56.1 | 56.9 | 63.6 | 62.5 | 63.2 | 59.8 |
| ps | LRL | 47.8 | 46.8 | 47.2 | 46.8 | 46.9 | 46.7 | 47.8 | 46.3 | 46.5 | 47.8 | 48.5 | 45.8 | 47.7 | 48.1 | 54.0 | 53.2 | 52.8 | 71.2 |
| rn | LRL | 36.0 | 35.5 | 35.4 | 35.3 | 35.7 | 34.9 | 34.6 | 34.7 | 34.6 | 34.7 | 36.1 | 35.1 | 35.5 | 35.8 | 44.4 | 43.1 | 40.1 | 57.4 |
| rw | LRL | 41.1 | 40.4 | 40.1 | 39.4 | 40.2 | 39.3 | 39.5 | 39.5 | 39.2 | 39.3 | 40.8 | 39.6 | 40.6 | 41.0 | 49.9 | 48.7 | 49.0 | 65.2 |
| sa | LRL | 39.3 | 40.0 | 39.8 | 39.8 | 39.8 | 40.1 | 40.6 | 39.5 | 40.1 | 40.5 | 40.2 | 39.1 | 39.7 | 40.6 | 46.6 | 46.7 | 47.5 | 77.1 |
| si | LRL | 50.7 | 50.4 | 50.0 | 50.3 | 50.2 | 49.9 | 51.6 | 48.6 | 49.8 | 51.1 | 52.7 | 48.4 | 50.7 | 52.2 | 58.3 | 57.2 | 58.3 | 77.6 |
| su | LRL | 52.9 | 52.2 | 52.7 | 52.4 | 52.6 | 52.2 | 53.1 | 51.7 | 51.8 | 52.7 | 53.6 | 51.4 | 53.1 | 53.3 | 57.7 | 57.4 | 57.0 | 61.1 |
| tg | LRL | 50.3 | 50.1 | 49.6 | 49.8 | 49.9 | 49.2 | 50.6 | 48.2 | 49.4 | 50.5 | 51.7 | 48.2 | 50.3 | 51.3 | 57.8 | 56.8 | 57.4 | 70.2 |
| ti | LRL | 30.2 | 31.3 | 30.7 | 27.3 | 33.2 | 28.5 | 31.5 | 26.0 | 29.4 | 31.1 | 32.3 | 24.5 | 30.4 | 31.3 | 42.1 | 40.5 | 40.1 | 56.3 |
| tk | LRL | 50.6 | 50.0 | 50.1 | 50.0 | 49.8 | 49.3 | 51.0 | 48.0 | 49.2 | 50.4 | 52.2 | 48.4 | 50.6 | 51.5 | 59.0 | 58.2 | 58.6 | 40.9 |
| tn | LRL | 43.9 | 43.8 | 43.6 | 43.5 | 43.9 | 43.1 | 43.6 | 42.4 | 42.6 | 43.6 | 42.4 | 42.4 | 43.8 | 44.2 | 50.0 | 49.7 | 49.0 | 63.7 |
| tpi | LRL | 44.1 | 46.0 | 45.3 | 46.3 | 45.7 | 44.7 | 43.0 | 43.8 | 42.6 | 42.2 | 41.6 | 43.9 | 44.2 | 43.4 | 41.0 | 50.1 | 46.1 | 68.4 |
| ts | LRL | 35.7 | 38.8 | 38.4 | 37.6 | 37.8 | 38.7 | 35.7 | 36.3 | 36.4 | 37.2 | 36.0 | 37.0 | 36.4 | 37.2 | 41.1 | 44.6 | 41.2 | 58.4 |
| tum | LRL | 32.1 | 34.1 | 33.1 | 31.9 | 32.7 | 32.7 | 31.8 | 33.2 | 32.5 | 32.3 | 32.7 | 33.2 | 32.6 | 32.8 | 34.7 | 34.4 | 34.3 | 55.4 |
| ug | LRL | 43.2 | 43.2 | 43.2 | 42.8 | 43.0 | 42.3 | 43.6 | 41.7 | 42.6 | 43.6 | 45.1 | 41.4 | 43.2 | 44.5 | 49.9 | 49.1 | 49.2 | 63.1 |
| wo | LRL | 31.3 | 31.6 | 31.0 | 31.0 | 31.7 | 31.1 | 29.6 | 28.8 | 29.7 | 29.0 | 29.5 | 29.6 | 30.3 | 29.3 | 40.8 | 40.3 | 37.7 | 61.2 |
| xh | LRL | 49.4 | 48.9 | 48.9 | 48.8 | 49.0 | 48.6 | 49.5 | 47.8 | 48.2 | 48.9 | 50.1 | 47.7 | 49.4 | 49.4 | 55.8 | 55.4 | 54.9 | 51.9 |
| af | MRL | 70.4 | 69.9 | 70.2 | 69.9 | 69.9 | 69.7 | 70.7 | 69.2 | 69.6 | 70.4 | 71.7 | 68.8 | 70.4 | 71.2 | 75.1 | 74.5 | 74.9 | 60.6 |
| az | MRL | 48.0 | 47.6 | 47.9 | 47.6 | 47.9 | 47.2 | 48.5 | 46.5 | 47.2 | 48.2 | 49.2 | 46.4 | 48.0 | 48.7 | 52.9 | 52.5 | 52.5 | 66.2 |
| bg | MRL | 61.4 | 60.8 | 61.0 | 61.0 | 61.2 | 60.5 | 61.8 | 59.8 | 60.5 | 61.4 | 62.3 | 59.8 | 61.3 | 62.1 | 66.3 | 65.9 | 66.5 | 68.4 |
| bn | MRL | 53.2 | 52.7 | 53.1 | 52.6 | 52.7 | 52.2 | 54.2 | 51.3 | 52.5 | 53.3 | 54.5 | 51.1 | 53.3 | 54.3 | 59.8 | 59.4 | 60.0 | 58.3 |
| bs | MRL | 60.8 | 60.5 | 60.6 | 60.5 | 60.3 | 60.4 | 61.5 | 59.6 | 60.2 | 61.2 | 62.0 | 59.3 | 60.9 | 61.7 | 66.0 | 65.6 | 65.8 | 62.9 |
| ca | MRL | 64.3 | 63.9 | 64.2 | 64.0 | 63.9 | 63.8 | 64.7 | 63.2 | 63.8 | 64.6 | 65.2 | 63.1 | 64.4 | 65.1 | 68.6 | 68.1 | 68.5 | 61.6 |
| ceb | MRL | 58.5 | 58.2 | 58.2 | 58.0 | 58.3 | 57.7 | 58.1 | 57.8 | 56.9 | 57.8 | 59.3 | 57.2 | 58.4 | 58.7 | 65.3 | 64.7 | 64.6 | 58.3 |
| ckb | MRL | 42.8 | 44.0 | 43.8 | 43.2 | 42.3 | 43.3 | 43.4 | 41.9 | 42.0 | 44.2 | 43.2 | 41.3 | 42.2 | 44.5 | 55.1 | 54.0 | 54.2 | 67.4 |
| et | MRL | 56.7 | 56.0 | 56.0 | 56.0 | 56.1 | 55.6 | 57.0 | 54.9 | 55.3 | 56.6 | 57.8 | 54.6 | 56.5 | 57.3 | 63.1 | 62.4 | 63.0 | 59.5 |
| fil | MRL | 60.9 | 60.8 | 60.8 | 60.6 | 60.5 | 60.3 | 61.6 | 59.6 | 59.9 | 61.1 | 62.4 | 59.5 | 61.0 | 61.9 | 67.5 | 66.8 | 67.3 | 61.6 |
| gd | MRL | 51.8 | 51.2 | 50.9 | 50.6 | 51.1 | 50.7 | 52.0 | 50.0 | 51.1 | 51.7 | 52.8 | 49.7 | 51.3 | 52.2 | 57.8 | 57.5 | 58.1 | 64.9 |
| gl | MRL | 61.5 | 61.4 | 61.3 | 61.2 | 61.3 | 61.2 | 61.8 | 60.5 | 61.1 | 61.8 | 62.2 | 60.5 | 61.6 | 62.3 | 65.1 | 64.7 | 65.0 | 68.7 |
| ha | MRL | 41.2 | 40.7 | 40.8 | 39.8 | 41.6 | 39.9 | 41.4 | 40.7 | 39.8 | 40.5 | 41.5 | 40.4 | 41.9 | 41.5 | 45.8 | 45.6 | 45.6 | 61.9 |
| hr | MRL | 57.8 | 57.4 | 57.7 | 57.4 | 57.6 | 57.3 | 58.2 | 56.7 | 57.2 | 58.1 | 58.9 | 56.5 | 58.0 | 58.6 | 62.6 | 62.1 | 62.6 | 62.7 |
| hy | MRL | 54.6 | 54.6 | 54.7 | 54.5 | 54.3 | 53.8 | 55.7 | 53.1 | 54.3 | 55.3 | 55.9 | 52.8 | 54.7 | 55.8 | 61.2 | 60.7 | 61.7 | 55.9 |
| ig | MRL | 38.6 | 38.4 | 38.6 | 37.2 | 38.4 | 37.6 | 38.6 | 37.8 | 37.8 | 38.6 | 39.7 | 37.6 | 38.9 | 38.8 | 43.9 | 43.3 | 43.1 | 67.3 |
| is | MRL | 51.4 | 51.0 | 50.9 | 50.9 | 50.8 | 50.7 | 52.0 | 49.5 | 50.3 | 51.7 | 52.7 | 49.4 | 51.2 | 52.1 | 58.2 | 57.5 | 58.4 | 72.4 |
| ka | MRL | 49.6 | 49.6 | 49.6 | 49.5 | 49.5 | 49.0 | 50.6 | 48.0 | 49.4 | 50.5 | 51.0 | 48.1 | 49.6 | 50.7 | 55.1 | 54.8 | 55.6 | 67.0 |
| kk | MRL | 52.5 | 52.1 | 52.1 | 52.4 | 52.0 | 51.5 | 53.2 | 50.7 | 51.6 | 53.3 | 53.8 | 50.6 | 52.6 | 53.9 | 59.1 | 58.4 | 59.1 | 68.8 |
| km | MRL | 47.7 | 47.3 | 47.5 | 46.5 | 48.1 | 45.5 | 48.3 | 45.9 | 47.3 | 47.9 | 49.0 | 46.2 | 47.0 | 48.3 | 52.2 | 52.1 | 52.9 | 77.2 |
| lo | MRL | 52.1 | 51.6 | 52.1 | 51.8 | 51.9 | 51.3 | 52.5 | 50.7 | 51.6 | 52.7 | 53.2 | 51.0 | 52.0 | 53.6 | 59.2 | 58.4 | 59.1 | 68.9 |
| lt | MRL | 52.9 | 52.8 | 52.6 | 52.7 | 52.5 | 52.2 | 53.5 | 51.3 | 52.1 | 53.1 | 54.5 | 51.2 | 53.0 | 53.8 | 58.6 | 58.3 | 58.7 | 65.8 |
| lv | MRL | 56.6 | 56.2 | 56.2 | 56.2 | 56.2 | 55.9 | 57.1 | 55.1 | 55.6 | 56.7 | 57.5 | 55.0 | 56.6 | 57.5 | 62.5 | 62.1 | 62.4 | 52.9 |
| mg | MRL | 42.3 | 41.5 | 41.8 | 40.4 | 41.3 | 40.9 | 41.8 | 41.0 | 40.9 | 41.2 | 42.9 | 40.6 | 42.0 | 42.4 | 46.0 | 46.9 | 46.7 | 57.8 |
| mi | MRL | 41.7 | 41.7 | 41.7 | 41.4 | 41.3 | 41.0 | 42.5 | 40.2 | 41.8 | 42.6 | 42.7 | 40.3 | 41.8 | 43.0 | 48.2 | 47.5 | 48.3 | 64.4 |
| mk | MRL | 61.3 | 60.9 | 60.8 | 60.7 | 60.9 | 60.6 | 61.5 | 59.6 | 60.5 | 61.2 | 62.1 | 59.6 | 61.3 | 61.8 | 66.1 | 65.6 | 66.1 | 64.8 |
| ml | MRL | 53.6 | 53.2 | 53.0 | 53.0 | 53.3 | 52.3 | 54.1 | 51.4 | 52.8 | 54.1 | 54.9 | 51.3 | 54.0 | 54.5 | 60.6 | 59.5 | 60.7 | 62.8 |
| mr | MRL | 53.5 | 53.0 | 53.1 | 53.1 | 53.1 | 52.6 | 54.3 | 52.0 | 52.7 | 54.5 | 54.9 | 51.6 | 53.8 | 54.5 | 59.9 | 59.3 | 59.7 | 59.3 |
| ne | MRL | 57.2 | 56.6 | 56.6 | 56.8 | 56.9 | 56.5 | 57.4 | 55.4 | 56.1 | 57.3 | 58.8 | 55.3 | 57.4 | 58.2 | 63.7 | 62.9 | 63.6 | 47.9 |
| ny | MRL | 41.1 | 41.1 | 40.6 | 40.5 | 41.1 | 40.2 | 41.1 | 40.5 | 40.2 | 40.7 | 41.8 | 40.4 | 41.5 | 41.3 | 45.2 | 45.4 | 44.9 | 67.9 |

| lang | cat. | B | T | $T_{75}$ | $T_{50}$ | $T_{25}$ | $T_{GAT}$ | $C_M$ | $C_P$ | $C_{MP}$ | $C_M$ T | $G_M$ | $G_P$ | $G_{MP}$ | $G_M$ T | $B_{BIG}$ | $T_{BIG}$ | $(C_M$ T$)_{BIG}$ | NLLB |
|---|---|---|---|---|---|---|---|---|---|---|---|---|---|---|---|---|---|---|---|
| or | MRL | 50.3 | 49.5 | 49.5 | 49.2 | 49.3 | 48.7 | 50.5 | 48.2 | 49.3 | 50.0 | 52.0 | 47.9 | 50.2 | 51.2 | 60.3 | 59.1 | 59.9 | 68.8 |
| sd | MRL | 53.1 | 52.5 | 52.7 | 52.3 | 52.3 | 52.1 | 53.3 | 51.7 | 52.1 | 53.2 | 54.1 | 51.2 | 53.3 | 53.6 | 60.0 | 59.1 | 59.3 | 59.1 |
| sl | MRL | 56.0 | 55.8 | 55.7 | 55.6 | 55.6 | 55.2 | 56.2 | 54.5 | 55.1 | 56.0 | 57.0 | 54.5 | 55.9 | 56.6 | 61.1 | 60.7 | 61.1 | 56.5 |
| sm | MRL | 46.0 | 46.5 | 46.6 | 45.5 | 46.5 | 45.4 | 46.1 | 45.3 | 45.7 | 45.8 | 47.3 | 45.3 | 46.3 | 46.7 | 51.6 | 51.6 | 49.6 | 64.2 |
| sn | MRL | 42.3 | 42.0 | 42.0 | 41.7 | 42.0 | 41.2 | 42.1 | 41.5 | 41.4 | 42.0 | 43.1 | 41.4 | 42.2 | 42.4 | 47.3 | 46.7 | 46.4 | 58.7 |
| so | MRL | 41.1 | 40.8 | 41.1 | 40.0 | 41.2 | 40.1 | 41.3 | 40.5 | 40.3 | 40.7 | 42.1 | 40.6 | 41.7 | 41.2 | 46.7 | 46.6 | 46.1 | 71.1 |
| sq | MRL | 59.9 | 59.6 | 59.5 | 59.4 | 59.7 | 59.2 | 60.3 | 58.5 | 59.0 | 59.8 | 61.0 | 58.0 | 59.9 | 60.5 | 65.3 | 64.8 | 65.2 | 54.4 |
| sr | MRL | 60.2 | 59.9 | 59.7 | 60.1 | 59.8 | 59.6 | 60.7 | 58.6 | 59.1 | 60.5 | 61.6 | 58.3 | 60.3 | 61.0 | 66.3 | 66.1 | 66.3 | 71.3 |
| st | MRL | 50.3 | 49.7 | 49.9 | 49.6 | 49.7 | 49.3 | 50.2 | 48.7 | 48.9 | 49.9 | 51.2 | 48.7 | 49.9 | 50.8 | 58.1 | 57.1 | 57.4 | 63.0 |
| sw | MRL | 56.3 | 55.8 | 55.9 | 55.6 | 55.9 | 55.5 | 56.6 | 54.4 | 54.8 | 56.0 | 57.7 | 54.3 | 56.1 | 57.1 | 63.2 | 62.2 | 62.6 | 63.4 |
| ta | MRL | 51.3 | 50.6 | 50.9 | 50.4 | 50.7 | 50.1 | 51.8 | 49.0 | 50.2 | 51.2 | 52.3 | 49.2 | 51.1 | 51.9 | 57.8 | 56.8 | 57.6 | 66.8 |
| te | MRL | 56.6 | 55.3 | 55.7 | 55.6 | 55.6 | 55.2 | 56.7 | 53.6 | 55.0 | 56.1 | 57.6 | 53.9 | 56.1 | 57.3 | 62.8 | 61.9 | 62.7 | 60.9 |
| tt | MRL | 48.3 | 48.2 | 48.0 | 48.2 | 48.3 | 47.7 | 48.5 | 46.7 | 47.6 | 48.3 | 49.0 | 46.6 | 47.9 | 48.9 | 55.9 | 55.5 | 55.5 | 64.6 |
| ur | MRL | 53.0 | 52.5 | 52.7 | 52.4 | 52.8 | 52.3 | 53.7 | 51.5 | 52.2 | 53.4 | 54.6 | 51.4 | 53.2 | 54.2 | 59.5 | 58.7 | 59.5 | 69.4 |
| uz | MRL | 53.3 | 52.7 | 52.9 | 52.6 | 52.8 | 52.1 | 53.9 | 51.4 | 52.3 | 53.2 | 54.7 | 51.3 | 53.4 | 54.2 | 59.6 | 59.4 | 59.6 | 54.1 |
| yi | MRL | 48.2 | 47.5 | 49.0 | 45.9 | 46.7 | 47.9 | 48.7 | 45.3 | 45.9 | 48.7 | 48.0 | 45.2 | 46.3 | 48.4 | 52.1 | 53.9 | 51.6 | 61.2 |
| yo | MRL | 35.4 | 35.5 | 35.4 | 35.0 | 35.2 | 34.8 | 35.8 | 34.0 | 35.2 | 35.8 | 36.5 | 34.1 | 35.2 | 36.0 | 40.6 | 40.1 | 40.4 | 62.9 |
| yue | MRL | 46.1 | 46.6 | 46.4 | 46.6 | 46.5 | 45.9 | 46.6 | 44.9 | 45.3 | 46.6 | 47.3 | 44.2 | 46.2 | 47.4 | 54.4 | 53.9 | 54.3 | 58.5 |
| zu | MRL | 49.9 | 49.2 | 49.3 | 48.9 | 49.5 | 48.7 | 49.7 | 48.3 | 48.2 | 50.2 | 50.7 | 48.1 | 49.9 | 51.1 | 56.3 | 56.4 | 56.3 | 68.3 |
| ar | HRL | 55.7 | 55.2 | 55.5 | 55.2 | 55.1 | 55.1 | 56.5 | 53.5 | 54.4 | 56.0 | 57.3 | 53.4 | 55.5 | 57.2 | 63.1 | 62.4 | 62.9 | 63.0 |
| cs | HRL | 59.0 | 58.7 | 58.9 | 58.7 | 58.6 | 58.5 | 59.5 | 57.6 | 58.5 | 59.0 | 60.2 | 57.4 | 59.0 | 60.0 | 64.0 | 63.4 | 63.9 | 65.7 |
| da | HRL | 66.2 | 65.6 | 65.8 | 65.7 | 65.5 | 65.5 | 66.6 | 64.8 | 65.3 | 66.0 | 66.8 | 64.7 | 66.0 | 66.5 | 70.1 | 69.7 | 70.3 | 70.4 |
| de | HRL | 61.4 | 61.0 | 61.2 | 60.9 | 61.0 | 60.7 | 62.1 | 59.8 | 60.7 | 61.7 | 62.6 | 59.6 | 61.4 | 62.2 | 66.6 | 66.1 | 66.7 | 69.0 |
| el | HRL | 54.9 | 54.6 | 54.6 | 54.8 | 54.8 | 54.5 | 55.5 | 53.5 | 54.6 | 55.4 | 56.3 | 53.4 | 55.0 | 56.0 | 60.0 | 59.5 | 60.5 | 64.9 |
| es | HRL | 55.5 | 55.5 | 55.4 | 55.3 | 55.4 | 55.3 | 55.8 | 54.5 | 55.5 | 55.5 | 56.1 | 54.7 | 55.5 | 56.1 | 59.4 | 59.1 | 59.3 | 66.1 |
| fa | HRL | 54.6 | 54.2 | 54.3 | 54.3 | 54.5 | 53.9 | 55.0 | 52.8 | 53.6 | 54.9 | 56.1 | 52.6 | 54.8 | 55.7 | 60.7 | 59.8 | 60.7 | 66.7 |
| fi | HRL | 53.0 | 52.7 | 52.6 | 52.5 | 52.2 | 52.1 | 53.7 | 50.9 | 52.2 | 53.5 | 54.6 | 50.9 | 53.0 | 54.0 | 60.0 | 59.2 | 60.0 | 69.7 |
| fr | HRL | 63.5 | 63.2 | 63.2 | 63.2 | 63.3 | 63.1 | 63.8 | 62.3 | 62.8 | 63.6 | 64.5 | 62.0 | 63.7 | 64.1 | 67.7 | 67.2 | 67.6 | 68.3 |
| hi | HRL | 59.2 | 58.5 | 58.8 | 58.6 | 58.8 | 58.5 | 59.5 | 57.8 | 58.1 | 59.1 | 60.3 | 57.3 | 59.3 | 60.0 | 64.8 | 64.0 | 64.6 | 71.3 |
| hu | HRL | 54.0 | 53.7 | 53.8 | 53.9 | 53.5 | 53.2 | 54.4 | 52.2 | 53.1 | 54.3 | 55.4 | 51.9 | 53.8 | 55.0 | 60.4 | 59.8 | 60.4 | 63.7 |
| id | HRL | 61.8 | 61.4 | 61.7 | 61.4 | 61.4 | 61.2 | 62.2 | 60.6 | 61.0 | 61.8 | 62.7 | 60.3 | 61.9 | 62.6 | 66.4 | 66.0 | 66.3 | 60.0 |
| it | HRL | 57.5 | 57.2 | 57.1 | 57.3 | 57.3 | 57.0 | 57.7 | 56.3 | 56.9 | 57.7 | 58.3 | 56.3 | 57.5 | 57.9 | 61.5 | 61.1 | 61.4 | 68.0 |
| iw | HRL | 56.4 | 55.9 | 55.9 | 55.5 | 55.8 | 55.5 | 57.6 | 53.8 | 55.6 | 57.0 | 58.4 | 53.7 | 56.3 | 57.7 | 64.3 | 63.6 | 64.8 | 64.2 |
| ja | HRL | 47.9 | 47.7 | 47.6 | 47.4 | 47.6 | 47.2 | 48.6 | 46.1 | 47.3 | 48.5 | 49.3 | 45.9 | 48.1 | 49.1 | 54.5 | 53.7 | 54.5 | 66.4 |
| ko | HRL | 48.2 | 48.3 | 48.1 | 47.9 | 48.1 | 47.6 | 49.6 | 46.5 | 48.1 | 49.2 | 49.9 | 46.1 | 48.7 | 49.8 | 55.2 | 54.8 | 55.5 | 68.6 |
| ms | HRL | 62.1 | 61.6 | 61.8 | 61.7 | 61.8 | 61.3 | 62.4 | 60.7 | 61.2 | 62.1 | 63.3 | 60.5 | 62.2 | 63.0 | 67.2 | 66.4 | 66.9 | 69.0 |
| nl | HRL | 55.2 | 55.1 | 54.9 | 54.8 | 54.8 | 54.7 | 55.4 | 54.1 | 54.6 | 55.2 | 56.0 | 53.9 | 55.0 | 55.6 | 58.6 | 58.5 | 58.8 | 71.2 |
| no | HRL | 62.6 | 62.4 | 62.2 | 62.3 | 62.3 | 62.1 | 63.0 | 61.3 | 61.7 | 62.2 | 63.2 | 61.1 | 62.5 | 62.3 | 66.6 | 65.8 | 66.5 | 59.0 |
| pl | HRL | 51.5 | 51.5 | 51.4 | 51.0 | 51.1 | 50.9 | 52.1 | 50.2 | 50.9 | 51.8 | 52.6 | 50.3 | 51.6 | 52.4 | 55.7 | 55.0 | 55.8 | 66.1 |
| pt | HRL | 66.3 | 66.0 | 66.0 | 66.0 | 66.0 | 65.8 | 66.6 | 65.2 | 65.9 | 66.5 | 67.4 | 65.2 | 66.5 | 67.0 | 70.3 | 69.7 | 70.1 | 59.5 |
| ro | HRL | 62.8 | 62.7 | 62.5 | 62.4 | 62.4 | 62.4 | 63.5 | 61.5 | 62.2 | 63.0 | 64.1 | 61.3 | 62.8 | 63.6 | 67.6 | 67.3 | 67.4 | 59.5 |
| ru | HRL | 56.1 | 56.0 | 56.0 | 55.8 | 55.9 | 55.5 | 56.6 | 54.8 | 55.4 | 56.3 | 57.2 | 54.5 | 56.1 | 56.6 | 60.8 | 60.1 | 60.8 | 72.7 |
| sk | HRL | 58.5 | 58.5 | 58.3 | 58.5 | 58.5 | 58.2 | 59.3 | 57.3 | 58.0 | 58.9 | 60.2 | 57.2 | 58.9 | 59.7 | 64.4 | 63.8 | 64.2 | 58.2 |
| sv | HRL | 65.2 | 64.8 | 64.9 | 64.7 | 64.9 | 64.5 | 65.5 | 63.7 | 64.4 | 65.2 | 66.1 | 63.4 | 65.0 | 65.6 | 69.7 | 69.4 | 69.6 | 63.3 |
| th | HRL | 49.2 | 48.5 | 48.3 | 48.1 | 48.2 | 47.3 | 49.4 | 46.8 | 48.2 | 49.4 | 50.4 | 47.2 | 48.8 | 49.7 | 56.1 | 55.4 | 56.3 | 61.2 |
| tr | HRL | 57.1 | 56.4 | 56.5 | 56.3 | 56.1 | 56.0 | 57.3 | 55.1 | 55.8 | 57.2 | 58.4 | 54.5 | 57.0 | 57.7 | 63.1 | 62.2 | 63.0 | 63.7 |
| uk | HRL | 57.9 | 57.4 | 57.3 | 57.3 | 57.4 | 57.3 | 58.2 | 56.4 | 57.0 | 58.0 | 59.1 | 56.1 | 57.8 | 58.5 | 63.3 | 62.6 | 63.3 | 69.2 |
| vi | HRL | 54.1 | 53.5 | 53.2 | 53.4 | 53.3 | 53.3 | 54.5 | 52.2 | 53.4 | 54.3 | 55.5 | 52.2 | 54.3 | 54.7 | 59.7 | 59.0 | 59.5 | 70.1 |
| zh | HRL | 48.5 | 48.7 | 48.4 | 48.6 | 48.5 | 48.1 | 49.3 | 46.7 | 48.0 | 49.1 | 50.2 | 46.8 | 49.0 | 50.0 | 55.0 | 54.2 | 55.1 | 61.4 |

**Table 25:** Full Results of CHRF on FLORES for the models that we trained, in the xx→en direction. See Table 23 to demystify the model abbreviations.