# OpenReview forum: "GATITOS: Using a New Multilingual Lexicon for Low-resource Machine Translation"
_EMNLP/2023/Conference — EMNLP 2023 Main_

### Official Review · Reviewer_TqkJ · 2023-08-04

**Soundness:** 3

**Excitement:**

4: Strong: This paper deepens the understanding of some phenomenon or lowers the barriers to an existing research direction.

**Missing References:**

There is an important strand of research in (statistical and neural) MT on the use of bilingual terminology lists. This is not mentioned in the related work. A seminal paper for neural MT on this topic is
Dinu, Georgiana, Prashant Mathur, Marcello Federico, and Yaser Al-Onaizan. "Training Neural Machine Translation to Apply Terminology Constraints." In Proceedings of the 57th Annual Meeting of the Association for Computational Linguistics, pp. 3063-3068. 2019.

The use of bilingual terminology lists is clearly connected to this work and some of the approaches proposed in the past overlap with the methods mentioned in this paper.

**Paper Topic And Main Contributions:**

This paper presents an investigation of the importance and impact of using bilingual lexica in real-world setting machine translation (MT), which means 200 translation models trained on a huge quantity of web-crawl (parallel and monolingual) data. For this purpose, several approaches to using bilingual lexica for data augmentation are tested and evaluated in different conditions. In particular, emphasis has been placed on multilingual machine translation, the combination of supervised and self-supervised training, the quality and quantity of the lexical data, and the impact on the performance of bilingual lexica when increasing the quantity of training data. As the results of this analysis, a small and high-quality bilingual lexicon covering 184 low-resource languages is made available with this paper. The results show that: 1) the main impact on performance when using lexical data augmentation is visible in unsupervised MT, 2) the data augmentation approaches show to be effective and have high complementarity, 3) the gains given by using augmented data significantly reduces when scaling up the model, and 4) the quality of bilingual lexica is fundamental to have an impact on MT more than the size.

**Questions For The Authors:**

-) What is the reason for not using or combining large publicly available datasets? Even in a smaller preliminary study then validated on the in-house data?

-) Section 6.1.1, can you have a situation where two entries in the bilingual lexica overlap in the source but with very different translations? If you have this situation, how do you manage it?
For instance,
Monolingual sentence: A B C D G
Bilingual entries: B -> X and B C -> Y Z

**Reasons To Accept:**

Strenghtnesses:

-) This paper addresses an interesting problem and it is supported by a large set of analyses and experiments covering several language pairs. The evaluation tries to address different aspects of the problem with ad hoc experiments. I really appreciate the analysis of Section 8.4.

-) The findings are interesting, in particular for the unsupervised MT. This aspect can be more emphasized in the paper.

-) The discussion is grounded and I really appreciate the paragraph on the takeaway message at the end of each section in section 8.

-) The release of the dataset is an important contribution to fostering research activities in low-resourced languages.

**Reasons To Reject:**

Major weaknesses:
-) The introduction gives some emphasis on the impact of the proposed data augmentation methods on the combination of supervised and self-supervised training, but this aspect has been barely discussed in the paper.



-) Probably the main issue of this paper is the use of in-house data. This aspect has also been commented on in the limitation of the paper by the authors. The use of these data does not allow any other researcher to replicate the results of this paper. I believe that the main contributions of scientific papers should be replicable to foster innovation and improvement. However, if there is a need of using in-house data, please motivate it. It is not clear from the paper the reason because it is not possible to use huge quantities of available data (e.g. from the OPUS repository or the Paracrawl project).
The lack of data in some language directions can be a good point, but an initial study could have been done and presented in fewer language directions using public data. The findings could be then validated using the in-house data.

-) There is a high value in this work, but the overall message of the paper is a bit confusing. The paper starts by emphasizing the impact of bilingual lexica on the combination of supervised and self-supervised methods, but this aspect has not been further discussed in the paper. The paper presents some approaches for data augmentation with bilingual lexica, but it is stated (beginning of section 6) that these approaches are not the main contributions, but the conclusion mainly highlights the impact of different approaches. The final message is that the proposed techniques have an impact mainly on unsupervised NMT, reducing the scope of the paper. The appendix contains a lot of material that can be easily a new paper by itself. Overall, some work is needed to improve the paper's clarity and facilitate the reader's appreciation of the paper.
All these factors make me wonder if a clear and more direct message could be obtained by focusing the paper only on unsupervised NMT and then having a longer version of the paper (probably for a journal) with all the results.

**Reproducibility:**

2: Would be hard pressed to reproduce the results. The contribution depends on data that are simply not available outside the author's institution or consortium; not enough details are provided.

**Reviewer Confidence:**

4: Quite sure. I tried to check the important points carefully. It's unlikely, though conceivable, that I missed something that should affect my ratings.

**Typos Grammar Style And Presentation Improvements:**

Typos:


-) Structure of Section 3.3 seems to be wrong. Sections 3.4 and 3.5 should be subsubsections.

-) Page 2, line 122: “The tl;dr”

-) Related Work: NLI and MLM are not previously introduced.

-) Page 6, line 488: missing space between FLORES-200 and “and”

-) Page 7, line 530: missing space “wins.” and “For”

Presentation Improvements:

-) The structure of the paper can be improved by moving Sections 3, 4, and 5 after Section 6. In case, there is an interest in emphasizing the data conditions (monolingual vs bilingual), only Sections 4 and 5 can be moved.


-) To facilitate the reader, please move Table 4 to the previous page.

---

> ### Author Rebuttal · Authors · 2023-08-26
>
> Thank you for your review and constructive criticism. We will address your concerns in a pointwise fashion.
>
> - **During revision, we can certainly elaborate on how the combination of supervised and self-supervised training impacted our experiments.** The brief explanation is this: we experimented with two data augmentation approaches (Codeswitch and GLOWUP) on both monolingual data (used for self-supervised training) and parallel data (used for supervised training). We found that augmenting monolingual data was much more successful than augmenting parallel data. That is, **our methods worked much better when incorporated in the self-supervised part of training than in the supervised part.** We can add a discussion of why this might be the case, but our hypothesis is that the data-augmented supervised task is simply much more difficult for the model than the data-augmented self-supervised task, so much so that the model doesn’t learn as effectively from the former.
>
> - Another reviewer also commented on our use of in-house data, so we will give the same response here:
>
> We acknowledge that one of the weaknesses of our paper is the use of in-house training data, and
> that the research community would be benefited by open-sourcing this data. However, due to the
> policies of our organization, we are not able to release the training data at this time. Though not ideal, we defend our use of in-house data for the following reasons:
>
> 1. **This project aimed to examine the effect of multilingual lexicon data on large, massively multilingual models trained on lots of data. This is the setting in which SOTA production MT models are trained, and we wanted to replicate that.** Other works have already examined similar data augmentation approaches in settings with smaller models and less data (which is why the “initial study” you suggested would be pretty redundant), but we wanted to see how these methods scale to production-level scenarios. Our in-house dataset presented the best opportunity for doing these experiments.
> 2. **Many of the low-resource languages in our paper do not have publicly available data, or the data available for them is of dubious quality.** Since we were mainly focused on improvements in very low-resource languages, including this data in our training was crucial. Again, we would like to open-source our training data for these languages, but unfortunately this is out of our control.
>
> - We can add clarification about the contributions of our paper. Essentially, **there are two broad contributions of our work to the research community: the open-sourcing of GATITOS, and the experiments we did to demonstrate how multilingual lexicons could be applied to multilingual MT at scale.** We acknowledge that the first of these contributions is applicable to a much broader range of researchers and labs, since anyone can use our publicly available lexicon but only a small number of industry labs can train massively multilingual NMT models from scratch. And while it is true that we could have done additional experiments at a much smaller scale using publicly available data, we believe the existing literature already covers this area very well already and that repeating these efforts would be mostly redundant.
>
> - **It is true that our methods show the greatest improvement on unsupervised languages, followed by low-resource ones. These languages were the explicit focus of our paper**,  thus the title “Using a New Multilingual Lexicon for Low-resource Machine Translation”. This does reduce the scope of the paper, but we strongly believe that these languages deserve close attention and that our work is useful even if it only significantly improves the performance of these languages.
>
> - You make a good point about the length of our paper and its suitability for a journal rather than a conference. We performed many experiments and analyses, and not all of them fit within 8-9 pages. That being said, **we believe our work would still be useful when presented in the EMNLP format, since one needn't look at every experiment we did to appreciate the contribution of our paper and the open-source dataset associated with it.**
>
> - We gave the problem of polysemy a lot of consideration when deciding how to implement the data augmentation. Ultimately, we ruled out using something like Word Sense Disambiguation for the sake of simplicity.
> As stated in the Introduction, we wanted our data augmentation techniques to be “simple, generalizable, and easy to implement.” When working with hundreds of languages, we couldn’t guarantee that we could do WSD accurately for the entire dataset (especially the low-resource languages we were focused on).
> Furthermore, WSD is something MT models learn implicitly during training, so **even though some of our lexical substitutions may have been incorrect in context, this is not fundamentally a problem during training: it simply makes the task a denoising task, in which the model can learn implicitly when a word in not translated in the correct sense.** As has been demonstrated by works like BART, source-side noise is often *helpful* in training MT models, as long as the target side is clean.
> In light of this, we simply sample lexicon translations at random during the data augmentation process.
>
> - During revision, we will add the citation you mentioned. Thank you for bringing this work to our attention.

---

### Official Review · Reviewer_GrpG · 2023-08-11

**Soundness:** 3

**Excitement:**

2: Mediocre: This paper makes marginal contributions (vs non-contemporaneous work), so I would rather not see it in the conference.

**Paper Topic And Main Contributions:**

The author claims that cross-lingual vocabulary alignment is still highly imperfect in the approach combining supervised and self-supervised methods, especially for low-resource and unsupervised languages. To address these issues, the authors propose to use multilingual lexica for the translation tasks and design a dataset name GATITOS.

**Questions For The Authors:**

A. The writing is not clear, and the work of the paper seems to be unfinished. Have you done the rest of the work, as mentioned in Line 227?

B. The paper is titled with the dataset GATITOS, could you provide more details about this dataset?

**Reasons To Accept:**

1. The authors conduct experiments for many existing multilingual lexica methods on low-resource machine translation tasks.

2. A publicly available dataset named GATITOS is proposed.

**Reasons To Reject:**

1. The paper is difficult to follow, there are some writing issues, such as Line 122.

2. The motivation is not clear, which refers to other papers in Line 89. It should be directly and logically illustrated in the paper.

3. The work is not completed, as indicated in Line 227. The GATITOS dataset only covers 24 languages, which may be the main contribution to this paper.

**Reproducibility:**

2: Would be hard pressed to reproduce the results. The contribution depends on data that are simply not available outside the author's institution or consortium; not enough details are provided.

**Reviewer Confidence:**

4: Quite sure. I tried to check the important points carefully. It's unlikely, though conceivable, that I missed something that should affect my ratings.

---

> ### Author Rebuttal · Authors · 2023-08-26
>
> Thank you for your review and constructive criticism. We will address your concerns in a pointwise fashion.
>
> - We apologize for any confusion caused by typos or other writing errors, and will make sure to correct these errors during revision
>
> - We believe the motivation of our paper is clearly stated in the Introduction:
> 1. We provide a thorough comparison of several lexicon-based data augmentation variants for MT, all of which are simple, generalizable, and easy to implement
> 2. We test these approaches “in the wild”, i.e. in a highly multilingual, web-mined data regime such as production systems tend to use, with hundreds of languages and billions of monolingual and parallel sentences
> 3. We explore the effects of lexical data quality and quantity
> 4. We demonstrate the efficacy of bilingual lexicon-based approaches as models scale
> 5. We open-source the high-quality multilingual GATITOS lexicon for 184, mostly low-resource languages
>
> - At the time of writing the paper, GATITOS only covered 24 languages. Currently, it covers 184 languages and is publicly available. GATITOS is an ongoing project, and we hope to add more languages in the future.
>
> - During revision, we can certainly add more details about the GATITOS dataset to our paper. Of course, it would be helpful during the discussion period if you clarified what sort of dataset metadata you are interested in seeing so we can make sure it appears in the camera-ready version.

---

### Official Review · Reviewer_htr1 · 2023-08-11

**Soundness:** 4

**Excitement:**

4: Strong: This paper deepens the understanding of some phenomenon or lowers the barriers to an existing research direction.

**Paper Topic And Main Contributions:**

This work demonstrates that adding publicly available multilingual lexicon data and carefully curated bilingual data to low-resource languages and high-resource languages respectively can yield some significant gains over baseline models.

**Reasons To Accept:**

The strength of the paper is its experimentation on different augmentation strategies across hundreds of languages in a real-world setting.

**Reasons To Reject:**

none

**Reproducibility:**

N/A: Doesn't apply, since the paper does not include empirical results.

**Reviewer Confidence:**

3: Pretty sure, but there's a chance I missed something. Although I have a good feel for this area in general, I did not carefully check the paper's details, e.g., the math, experimental design, or novelty.

---

> ### Author Rebuttal · Authors · 2023-08-26
>
> Thank you for your review. We are happy you found our work valuable. Of course, we would greatly appreciate it if you elaborate on your reasons for supporting our paper during discussion with other reviewers.

---

### Official Review · Reviewer_zAXT · 2023-08-12

**Soundness:** 3

**Excitement:**

3: Ambivalent: It has merits (e.g., it reports state-of-the-art results, the idea is nice), but there are key weaknesses (e.g., it describes incremental work), and it can significantly benefit from another round of revision. However, I won't object to accepting it if my co-reviewers champion it.

**Missing References:**

A similar approach to GLOWUP is presented in “Prompting Neural Machine Translation with Translation Memories” (Rehemann et al., AAAI 2023).
However, the corresponding translation is the prefix of the decoded sentence which might yield better results also in your case.

**Paper Topic And Main Contributions:**

This paper introduces a new multilingual lexicon covering 184 languages.
The dataset will be open-sourced and targets low-resource languages while providing high quality.
To demonstrate the high quality of the lexicon, the authors provide experimental results on different translation tasks for different scenarios showing that leveraging these additional data yields higher translation quality.
The author use both established and newly developed methods for integrating lexicon data into the model training.

**Questions For The Authors:**

- While I appreciate that the lexicon was tested “in the wild”, the results are hard to reproduce. Even for reproducing the results based on Appendix A, a few details are missing (vocab. size, used word piece model, learning rate etc.).
- What is the intuition of introducing GLOWUP as a new task? What would happen if you do GLOWUP for the regular translation task (<2translation>)? It might work better during inference (Section 8.2) then.

**Reasons To Accept:**

- Curating a high-quality, multilingual lexicon and open-sourcing it
- Experimental results beyond the usual, lab-conditioned translation tasks

**Reasons To Reject:**

- Experimental results are hard to reproduce since data are mostly in-house, details on public dataset experiments are missing
- GLOWUP approach is not convincing (why not adding the corresponding translation as prefix to the decoder, see Missing References and Questions)
- No significance test, e.g. for Table 3: Seems simply adding the lexicon to the training data is the most efficient and effective approach; are differences to CodeSwitchMonoGatiPanlex significant? (Same for Table 4: Parallel vs. Parallel+GatiPanlex)

**Reproducibility:**

2: Would be hard pressed to reproduce the results. The contribution depends on data that are simply not available outside the author's institution or consortium; not enough details are provided.

**Reviewer Confidence:**

4: Quite sure. I tried to check the important points carefully. It's unlikely, though conceivable, that I missed something that should affect my ratings.

**Typos Grammar Style And Presentation Improvements:**

- Section 5: A short introduction of MASS would be great
- Section 5/6: Please introduce abbreviations first before using them (e.g. Glowup or MASS)
- Provide some examples in the methods section (not in the Appendix), in particular for newly introduced methods such as GLOWUP
- The naming in Section 7.1.3 is not very intuitive. I took me a while to understand the “GatiPanlex” suffix.
- Personally I prefer to avoid using casual formulations such as “tl;dr” or “Is this the Bitter Lessen […] getting us again? Perhaps— […]”

---

> ### Author Rebuttal · Authors · 2023-08-26
>
> Thank you for your review and constructive criticism.
>
> We acknowledge that one of the weaknesses of our paper is the use of in-house training data, and that the research community would be benefited by open-sourcing this data. However, due to the policies of our organization, we are not able to release the training data at this time. Though not ideal, we defend our use of in-house data for the following reasons:
>
> 1. **This project aimed to examine the effect of multilingual lexicon data on large, massively multilingual models trained on lots of data. This is the setting in which SOTA production MT models are trained, and we wanted to replicate that.** Other works have already examined similar data augmentation approaches in settings with smaller models and less data, but **we wanted to see how these methods scale.** Our in-house dataset presented the best opportunity for doing these experiments.
> 2. **The majority of the low-resource languages in our paper do not have publicly available data, or the data available for them is of dubious quality.** Since we were mainly focused on improvements in very low-resource languages, including this data in our training was crucial. Again, we would like to open-source our training data for these languages, but unfortunately this is out of our control.
>
> In regard to your criticism of the GLOWUP strategy: we considered the method you proposed as well, where the lexicon translations are added as prefixes to the decoder. **However, due to the large number of data augmentation variants we were already experimenting with, we opted to try just the simpler approach that involved only modification of the source text.** Future work should certainly explore the decoder-prefix strategy, as well as the use of the <2translation> tag rather than <2glowup>. **The latter experiment could be added to the paper during the revision period.** To answer your question, the intuition behind using a different tag was that it might help the model distinguish between the different tasks and avoid confusion.
>
> Other things we can add during the revision period to address your concerns include significance tests (p-values) for some of the tables, plus the training details you requested like learning rate, vocab size, etc.
>
> **Finally, we would like to stress that although our training data is not open-source, one of the main contributions of this paper is the release of GATITOS, a multilingual lexicon that now covers 184 low-resource languages, many of which do not have any other multilingual resources available on the web.** We believe this open-source dataset will prove highly useful in building systems for extremely low-resource languages, and we ask that you give this fact ample weight in considering the merits of our paper. **The experimental results should be considered secondary to the open-sourced dataset.**

---

### Meta-Review · Area_Chair_BMiW · 2023-09-14

**Recommendation:** 4

**Metareview:**

In this paper the authors present a curated bilingual lexicon covering 168 language pairs, and
make an experimental comparison of different ways of using this lexicon in data augmentation
for low-resource MT.

For soundness, reviewers zAXT, GrpG and TqkJ all agree on a score of 3. The main criticism being
that the experiments are impossible to reproduce because of the use of in-house data, and the lack
of detail on the experimental settings. To me, this is a valid criticism, the authors could have
used the Opus data set for example, and aspects such as the quality and domain of the dataset used
in these experiments is not available to the community. Given this, I think a score of 3 is reasonable.

There is a disagreement between the reviewers about excitement, ranging from 2 (GrpG) to 4 (TqkJ).
The latter reviewer emphasises the wide coverage of the experimental work, the interesting implications
of the results on unsupervised MT, and the usefulness of the data set (which is released as open-source).
To me, these all seem like worthy contributions.

Reviewer GrpG raised some concerns about the clarity of the paper, although did not back up their
concerns with any valid specific points. I note that reviewer zAXT also had several questions and
suggestions, and reviewer TqkJ raised issues about the structure of the paper. These comments
from reviewers suggest that there are clarity issues with the paper, and that the
the authors should take into account the
reviewers that provided constructive comments.

Lastly, reviewer GrpG raised a question about the completeness of the data set. In the discussion, it
was revealed that the data set was incomplete at submission time, but has now been completed with
184 languages. The small version (with 24 languages) was used in the experiments. The authors need to
clarify this is their update. I would also ask them to clarify the licence of the data set.

Overall, I expect that the data set is a worthwhile contribution, and the experiments shed light
on data augmentation for low-resource languages.

---

### Decision · Program_Chairs · 2023-10-07

**Decision:**

Accept-Main

**Comment:**

In this paper the authors present a curated bilingual lexicon covering 168 language pairs, and
make an experimental comparison of different ways of using this lexicon in data augmentation
for low-resource MT.

For soundness, reviewers zAXT, GrpG and TqkJ all agree on a score of 3. The main criticism being
that the experiments are impossible to reproduce because of the use of in-house data, and the lack
of detail on the experimental settings. To me, this is a valid criticism, the authors could have
used the Opus data set for example, and aspects such as the quality and domain of the dataset used
in these experiments is not available to the community. Given this, I think a score of 3 is reasonable.

There is a disagreement between the reviewers about excitement, ranging from 2 (GrpG) to 4 (TqkJ).
The latter reviewer emphasises the wide coverage of the experimental work, the interesting implications
of the results on unsupervised MT, and the usefulness of the data set (which is released as open-source).
To me, these all seem like worthy contributions.

Reviewer GrpG raised some concerns about the clarity of the paper, although did not back up their
concerns with any valid specific points. I note that reviewer zAXT also had several questions and
suggestions, and reviewer TqkJ raised issues about the structure of the paper. These comments
from reviewers suggest that there are clarity issues with the paper, and that the
the authors should take into account the
reviewers that provided constructive comments.

Lastly, reviewer GrpG raised a question about the completeness of the data set. In the discussion, it
was revealed that the data set was incomplete at submission time, but has now been completed with
184 languages. The small version (with 24 languages) was used in the experiments. The authors need to
clarify this is their update. I would also ask them to clarify the licence of the data set.

Overall, I expect that the data set is a worthwhile contribution, and the experiments shed light
on data augmentation for low-resource languages.